# Robust Knowledge Transfer in Tiered RL

**Jiawei Huang**
Department of Computer Science
ETH Zurich
jiawei.huang@inf.ethz.ch

**Niao He**
Department of Computer Science
ETH Zurich
jiawei.huang@inf.ethz.ch

## Abstract

In this paper, we study the Tiered Reinforcement Learning setting, a parallel transfer learning framework, where the goal is to transfer knowledge from the low-tier (source) task to the high-tier (target) task to reduce the exploration risk of the latter while solving the two tasks in parallel. Unlike previous work, we do not assume the low-tier and high-tier tasks share the same dynamics or reward functions, and focus on robust knowledge transfer without prior knowledge on the task similarity. We identify a natural and necessary condition called the "Optimal Value Dominance" for our objective. Under this condition, we propose novel online learning algorithms such that, for the high-tier task, it can achieve constant regret on partial states depending on the task similarity and retain near-optimal regret when the two tasks are dissimilar, while for the low-tier task, it can keep near-optimal without making sacrifice. Moreover, we further study the setting with multiple low-tier tasks, and propose a novel transfer source selection mechanism, which can ensemble the information from all low-tier tasks and allow provable benefits on a much larger state-action space.

## 1 Introduction

Comparing with individual learning from scratch, transferring knowledge from other similar tasks or side information has been proven to be an effective way to reduce the exploration risk and improve sample efficiency in Reinforcement Learning (RL). Multi-Task RL (MT-RL) [29] and Transfer RL [26, 18, 37] are two mainstream knowledge transfer frameworks; however, both are subject to limitations when dealing with real-world scenarios. MT-RL studies the setting where a set of similar tasks are solved concurrently, and the main objective is to accelerate the learning by sharing information of all tasks together. However, in practice, in many MT-RL scenarios, the tasks are not equally important and we are more interested in the performance of certain tasks. For example, in robot learning, a few robots are more valuable and hard to fix, while the others are cheaper or just simulators. Most existing works on MT-RL treat all tasks equally and focus primarily on the reduction of the total regret of all tasks as a whole [3, 8, 36, 11], with no guarantee of improving a particular task. In contrast, transfer RL distinguishes the priority of different tasks by categorizing them into source and target tasks and aims at transferring the knowledge from source tasks (or some side information like value predictors) to facilitate the learning of target tasks [21, 27, 9, 10]. However, a key assumption in transfer RL is that the source task is completely solved before the learning of the target task, and this is not always practical. For example, in some sim-to-real domain, the source task simulator may require a long time to solve [5], and in some user-interaction scenarios [13], the source and target tasks refer to different user groups and they have to be served simultaneously. In these cases, it's more reasonable to solve the source and target tasks in parallel and transfer the information immediately once available.

Recently, [13] proposed a new "parallel knowledge transfer" framework, called Tiered RL, which is promising to fill the gap. Tiered RL considers the case when a source task $M_{\text{Lo}}$ and a target task

$M_{\text{Hi}}$ are learned in parallel, by two separate algorithms $\text{Alg}^{\text{Lo}}$ and $\text{Alg}^{\text{Hi}}$, and its goal is to reduce the exploration risk and regret in learning $M_{\text{Hi}}$ by leveraging knowledge transfer from $M_{\text{Lo}}$ to $M_{\text{Hi}}$. [13] showed that under the strong assumption that $M_{\text{Lo}} = M_{\text{Hi}}$, it's possible to achieve constant regret in learning $M_{\text{Hi}}$ while keeping regret in $M_{\text{Lo}}$ optimal in gap-dependent setting. Yet, their algorithm based on Pessimistic Value Iteration (PVI) [16] can be hardly applied when the assumption $M_{\text{Lo}} = M_{\text{Hi}}$ breaks, given its pure exploitation nature, making their result very restrictive.

In this paper, we study the general Tiered RL setting *without prior knowledge* about how similar the tasks are [1]. The key question we would like to address is: **Can we design algorithms s.t.: (1) Regret in $M_{\text{Lo}}$ keeps near-optimal; (2) Regret in $M_{\text{Hi}}$ achieves provable benefits when $M_{\text{Lo}}$ and $M_{\text{Hi}}$ are similar while retaining near-optimal otherwise?** Note for $\text{Alg}^{\text{Lo}}$, we expect it to achieve near-optimal regret bounds, which is reasonable since the source task is often important and our results still hold if relaxing it. As for $\text{Alg}^{\text{Hi}}$, we expect it to be *robust*, i.e., it can adaptively exploit from $M_{\text{Lo}}$ if it is close to $M_{\text{Hi}}$, while avoiding negative transfer in other cases. Notably, our setting strictly generalizes [13] and is much more challenging, for balancing the exploitation from $M_{\text{Lo}}$ and exploration from $M_{\text{Hi}}$ without prior knowledge of task similarity. We give positive answers to the above question in this paper and demonstrate provable benefits with robust knowledge transfer for Tiered RL framework. Below, we summarize our main contributions in three aspects.

**Our first contribution** is to identify essential conditions and notions about when and how our objective is achievable. We first provide a mild condition called *Optimal Value Dominance* (OVD), and in Sec. 3, we justify its necessity to our objective by a lower bound result. Our lower bound holds even if $M_{\text{Lo}}$ is fully known to the learner, and therefore, it also justifies the necessity of similar assumptions in previous transfer RL literatures [9, 10]. Besides, we introduce the notion of *transferable states* to characterize states on which $M_{\text{Hi}}$ is expected to achieve benefits by transferring knowledge from $M_{\text{Lo}}$. We believe those findings also provide useful insights for further works.

As **our second contribution**, in Sec. 4, we propose novel algorithms for Tiered Multi-Armed Bandit (MAB) and Tiered RL, which can achieve robust parallel transfer by balancing between pessimism-based exploitation from $M_{\text{Lo}}$ and optimism-based online learning in $M_{\text{Hi}}$. Depending on the similarity between $M_{\text{Lo}}$ and $M_{\text{Hi}}$, our algorithms can enjoy constant regret on a proportion of state-action pairs or even on the entire $M_{\text{Hi}}$ by leveraging information from $M_{\text{Lo}}$, while timely interrupting negative transfer and retaining near-optimal regret on states dissimilar between two tasks. Moreover, in the bandit setting, our result implies a strictly improved regret bound when $M_{\text{Hi}} = M_{\text{Lo}}$, compared with previous results under the same setting [22, 13].

Beyond the single low-tier task setting, in many real-world scenarios, it's reasonable to assume there are multiple different low-tier tasks $\mathcal{M}_{\text{Lo}} = \{M_{\text{Lo},w}\}_{w=1}^{W}$ available. As **our third contribution**, in Sec. 5, we extend our algorithm to this setting with a new source task selection mechanism. By novel techniques, we show that, even if each $M_{\text{Lo},w}$ may only share similarity with $M_{\text{Hi}}$ on a small portion of states, we are able to ensemble the information from each source task together to achieve constant regret on a much larger state-action space "stitched" from the transferable state-action set in each individual $M_{\text{Lo},w}$, at the expense of an additional $\log W$ factor in regret. Besides, our algorithm is still robust to model difference and retains near-optimal regret in general. Although we only study the Tiered RL setting in this paper, we believe our task selection strategy can be applied to standard transfer RL setting [9, 10] when multiple (partially correct) side information or value predictors are provided, which is an interesting direction for future work. Finally, we conduct experiments in toy examples to verify our theoretical results.

## 1.1 Closely Related Work

For the lack of space, we only discuss closely related work here and defer the rest to Appx. A.2. The most related to us is the Tiered RL framework [13], which was originally motivated by Tiered structure in user-oriented applications. However, they only studied the case when $M_{\text{Hi}} = M_{\text{Lo}}$, which limits the practicability of their algorithms. Although there is a sequence of work studying parallel transfer learning in multi-agent system [25, 19], but they mainly focused on heuristic empirical algorithms and did not have theoretical guarantees.

Transfer RL [37], compared to learning from scratch, can reduces exploration risk of target task by leveraging information in similar source tasks or side information [21, 27, 9, 10]. Comparing with

---

[1]We defer the detailed framework to Appx. A.1 for completeness, which is the same as Fr. 1 in [13].

transfer RL setting, our parallel transfer setting has some additional challenges. Firstly, $M_{\text{Hi}}$ can only leverage estimated value/model/optimal policy from $M_{\text{Lo}}$ with uncertainty, which implies we need additional efforts to control failure events with non-zero probability comparing with normal transfer RL setting. Secondly, the constraints on the optimality of $\text{Regret}_K(M_{\text{Lo}})$, although reasonable, restrict the transferable information because in $M_{\text{Lo}}$ estimation uncertainty can only be controlled on those states frequently visited. Moreover, none of these previous work studies how to leverage multiple partially correct side information like what we did in Sec. 5. In MT-RL setting, the benefits of leveraging information gathered from other tasks has been observed from both empirical and theoretical works [31, 3, 8, 24, 36, 11]. But MTRL treats each task equally, and the reduction of total regret over all tasks does not directly imply benefits achieved in a particular task.

## 2 Preliminary and Problem Formulation

**Tiered Stochastic MAB and Tiered Episodic Tabular RL** In Tiered MAB setting, we consider a low-tier task $M_{\text{Lo}}$ and a high-tier task $M_{\text{Hi}}$ sharing the arm/action space $\mathcal{A} = \{1, 2, ..., A\}$. By pulling the arm $i \in [A]$ in $M_{\text{Lo}}$ or $M_{\text{Hi}}$, the agent can observe a random variable $r_{\text{Lo}}(i)$ or $r_{\text{Hi}}(i) \in [0, 1]$. We will use $\mu_{\text{Lo}}(i) = \mathbb{E}[r_{\text{Lo}}(i)]$ and $\mu_{\text{Hi}}(i) = \mathbb{E}[r_{\text{Hi}}(i)]$ to denote the expected return of the $i$-th arm in $M_{\text{Lo}}$ and $M_{\text{Hi}}$, respectively, and note that it's possible that $\mu_{\text{Lo}}(i) \neq \mu_{\text{Hi}}(i)$.

For Tiered RL, we assume that two tasks $M_{\text{Lo}} = \{\mathcal{S}, \mathcal{A}, H, \mathbb{P}_{\text{Lo}}, r_{\text{Lo}}\}$ and $M_{\text{Hi}} = \{\mathcal{S}, \mathcal{A}, H, \mathbb{P}_{\text{Hi}}, r_{\text{Hi}}\}$ share the finite state $\mathcal{S}$ and action space $\mathcal{A}$ across episode length $H$ (i.e. $\mathcal{S}_h = \mathcal{S}, \mathcal{A}_h = \mathcal{A}$ for any $h \in [H]$), but may have different time-dependent transition and reward functions $\mathbb{P}_{\text{Lo}} = \{\mathbb{P}_{\text{Lo},h}\}_{h=1}^H, r_{\text{Lo}} = \{r_{\text{Lo},h}\}_{h=1}^H$ and $\mathbb{P}_{\text{Hi}} = \{\mathbb{P}_{\text{Hi},h}\}_{h=1}^H, r_{\text{Hi}} = \{r_{\text{Hi},h}\}_{h=1}^H$. W.l.o.g., we assume the initial state $s_1$ is fixed, and the reward functions $r_{\text{Lo}}$ and $r_{\text{Hi}}$ are deterministic and bounded by $[0, 1]$. In episodic MDPs, we study the time-dependent policy specified as $\pi := \{\pi_1, ..., \pi_H\}$ with $\pi_h : \mathcal{S}_h \rightarrow \Delta(\mathcal{A}_h)$ for all $h \in [H]$, where $\Delta(\mathcal{A}_h)$ denotes the probability simplex over the action space. With a slight abuse of notation, when $\pi_h$ is a deterministic policy, we use $\pi_h : \mathcal{S}_h \rightarrow \mathcal{A}_h$ to denote the deterministic mapping. Besides, we use $Q_h^\pi(s, a) = \mathbb{E}[\sum_{h'=h}^H r_{h'}(s_{h'}, a_{h'})|s_h = s, a_h = a, \pi]$, $V_h^\pi(s) = \mathbb{E}_{a \sim \pi}[Q_h^\pi(s, a)]$ to denote the value function for $\pi$ at step $h \in [H]$, and denote $d_h^\pi(\cdot) := \Pr(s_h = \cdot|\pi)$ and $d_h^\pi(\cdot, \cdot) := \Pr(s_h = \cdot, a_h = \cdot|\pi)$ as the state and state-action occupancy w.r.t. policy $\pi$. We use $\pi^*$ to denote the optimal policy, and $V_h^*, Q_h^*, d_h^*$ as a short note when $\pi = \pi^*$. To avoid confusion, we will specify the policy and value functions in $M_{\text{Lo}}$ and $M_{\text{Hi}}$ by Lo and Hi in subscription, respectively. For example, in $M_{\text{Lo}}$ we have $\pi_{\text{Lo}} := \{\pi_{\text{Lo},1}, ..., \pi_{\text{Lo},H}\}$, $Q_{\text{Lo},h}^{\pi_{\text{Lo}}}/V_{\text{Lo},h}^{\pi_{\text{Lo}}}$ and $Q_{\text{Lo},h}^*/V_{\text{Lo},h}^*, d_{\text{Lo},h}^{\pi_{\text{Lo}}}, d_{\text{Lo},h}^*$, and similarly for $M_{\text{Hi}}$.

**Gap-Dependent Setting** Throughout, we focus on gap-dependent setting [17, 23, 34, 7]. Below we introduce the notion of gap for $M_{\text{Lo}}$ as an example, and those for $M_{\text{Hi}}$ follows similarly. In MAB case, the gap in $M_{\text{Lo}}$ w.r.t. arm $i$ is defined as $\Delta_{\text{Lo}}(i) := \max_{j \in [A]} \mu_{\text{Lo}}(j) - \mu_{\text{Lo}}(i), \forall i \in [A]$, and for tabular RL setting, we have $\Delta_{\text{Lo}}(s_h, a_h) := V_{\text{Lo},h}^*(s_h) - Q_{\text{Lo},h}^*(s_h, a_h), \forall h \in [H], s_h \in \mathcal{S}_h, a_h \in \mathcal{A}_h$. We use $\Delta_{\text{Lo,min}}$ to refer to the minimal gap such that $\Delta(s_h, a_h) \geq \Delta_{\text{Lo,min}}$ for all non-optimal actions $a_h$, and use $\Delta_{\min} := \min\{\Delta_{\text{Lo,min}}, \Delta_{\text{Hi,min}}\}$ to denote the minimal gap over two tasks. In the gap-dependent setting, we assume $\Delta_{\min} > 0$.

**Knowledge Transfer from Multiple Low-Tier Tasks** In this case, we assume there are $W > 1$ different source tasks $\mathcal{M}_{\text{Lo}} = \{M_{\text{Lo},w}\}_{w=1}^W$ and all the tasks share the same state and action spaces but may have different transition and reward function. We defer the extended framework for this setting to Appx. A. We specify the task index $w \in [W]$ in sub-scription to distinguish the notation for different source tasks (e.g. $\mathbb{P}_{\text{Lo},w,h}, Q_{\text{Lo},w,h}^{(\cdot)}$). Moreover, we define $\Delta_{\min} := \min\{\Delta_{\text{Lo,1,min}}, ..., \Delta_{\text{Lo,W,min}}, \Delta_{\text{Hi,min}}\}$. For convenience, in the rest of the paper, we use TRL-MST (Tiered RL with Multiple Source Task) as a short abbreviation for this setting.

**Performance Measure** We use Pseudo-Regret as performance measure: $\text{Regret}_K(M_{\text{Lo}}) := \mathbb{E}\left[\sum_{k=1}^K V_1^*(s_1) - V_1^{\pi_{\text{Lo}}^k}(s_1)\right]$; $\text{Regret}_K(M_{\text{Hi}}) := \mathbb{E}\left[\sum_{k=1}^K V_1^*(s_1) - V_1^{\pi_{\text{Hi}}^k}(s_1)\right]$, where $K$ is the number of iterations, $\{\pi_{\text{Lo}}^k\}_{k=1}^K$ and $\{\pi_{\text{Hi}}^k\}_{k=1}^K$ are generated by the algorithms.

**Frequently Used Notations** We denote $[n] = \{1, 2, ..., n\}$. Given a transition matrix $\mathbb{P} : \mathcal{S} \times \mathcal{A} \rightarrow \Delta(\mathcal{S})$, and a function $V : \mathcal{S} \rightarrow \mathbb{R}$, we use $\mathbb{P}V(s_h, a_h)$ as a short note of $\mathbb{E}_{s' \sim \mathbb{P}(\cdot|s,a)}[V(s')]$. We will use $i_{\text{Lo}}^*/i_{\text{Hi}}^*$ to denote the optimal arm in bandit setting, and $\pi_{\text{Lo}}^*/\pi_{\text{Hi}}^*$ denotes the optimal policy in RL setting. In TRL-MST setting, we use $i_{\text{Lo},w}^*/\pi_{\text{Lo},w}^*$ to distinguish different source tasks.

## 2.1 Assumptions and Characterization of Transferable States

Throughout the paper, we make several assumptions. The first one is the uniqueness of the optimal policy, which is common in the literature [22, 4].

**Assumption A.** Both $M_{\text{Lo}}$ (or $\{M_{\text{Lo},w}\}_{w=1}^{W}$) and $M_{\text{Hi}}$ have unique optimal arms/policies.

Next, we introduce a new concept called "Optimal Value Dominance" (OVD for short), which says that for each state (or at least those states reachable by optimal policy in $M_{\text{Lo}}$), the optimal value of $M_{\text{Lo}}$ is an approximate overestimation for the optimal value of $M_{\text{Hi}}$. In Sec. 3, we will use a lower bound to show such a condition is necessary to attain the robust transfer objective.

**Assumption B.** In single source task setting, we assume $M_{\text{Lo}}$ has Optimal Value Dominance (OVD) over $M_{\text{Hi}}$, s.t., $\forall h \in [H]$, for all $s_h \in \mathcal{S}_h$ (or only for those $s_h$ with $d_{\text{Lo},h}^*(s_h) > 0$), we have: $V_{\text{Lo},h}^*(s_h) \geq V_{\text{Hi},h}^*(s_h) - \frac{\Delta_{\min}}{2(H+1)}$. In TRL-MST setting, we assume each $M_{\text{Lo},w}$ has OVD over $M_{\text{Hi}}$.[2]

We remark that Assump. B is a rather mild condition that naturally holds with reward shaping. Note that since $V_{\text{Hi},h}^*(\cdot) \leq H - h$, by shifting the reward function of $M_{\text{Lo}}$ to $r_{\text{Lo},h}'(\cdot,\cdot) = r_{\text{Lo},h}(\cdot,\cdot) + 1$, we immediately obtain the OVD property. Even though, such a reward shift may impair the set of transferable states as we will introduce in Def. 2.2. We provide several reasonable settings in Appx. A.3 including identical model [13], small model difference, and known model difference, where Assump. B is satisfied and there exists a non-empty set of transferable states. We also point out that several existing work on transfer RL assumed something similar or even stronger [9, 10], which we defer a thorough discussion to Appx. A.2.

**Assumption C.** The learner has access to a quantity $\widetilde{\Delta}_{\min}$ satisfying $0 \leq \widetilde{\Delta}_{\min} \leq \Delta_{\min}$.

The final one is about the knowledge of a lower bound of $\Delta_{\min}$, which can always be satisfied by choosing $\widetilde{\Delta}_{\min} = 0$. Nevertheless, it would be more beneficial if the learner has access to some quantity $\widetilde{\Delta}_{\min}$ closer to $\Delta_{\min}$ than 0. As we introduce below, the magnitude of $\widetilde{\Delta}_{\min}$ is related to how we quantify the similarity between $M_{\text{Lo}}$ and $M_{\text{Hi}}$ and which states we expect to benefit from knowledge transfer. Below we focus on the single source task setting and defer the discussion for TRL-MST setting to Sec. 5.

**Definition 2.1** ($\varepsilon$-Close). Task $M_{\text{Hi}}$ is $\varepsilon$-close to task $M_{\text{Lo}}$ on $s_h$ at step $h$ for some $\varepsilon > 0$, if $V_{\text{Lo},h}^*(s_h) - V_{\text{Hi},h}^*(s_h) \leq \varepsilon$ and $\pi_{\text{Hi}}^*(s_h) = \pi_{\text{Lo}}^*(s_h)$.

**Definition 2.2** ($\lambda$-Transferable States). State $s_h$ is $\lambda$-transferable for some $\lambda > 0$, if $d_{\text{Lo}}^*(s_h) > \lambda$ and $M_{\text{Hi}}$ is $\frac{\widetilde{\Delta}_{\min}}{4(H+1)}$-close to $M_{\text{Lo}}$ on $s_h$. The set of $\lambda$-transferable states at $h \in [H]$ is denoted as $\mathcal{Z}_h^\lambda$.

We regard $s_h$ in $M_{\text{Lo}}$ has transferable knowledge to $M_{\text{Hi}}$, if it can be reached by optimal policy in $M_{\text{Lo}}$ and the optimal value and action at $s_h$ for two tasks are similar. Here the condition $d_{\text{Lo}}^*(s_h) > 0$ is necessary since in $M_{\text{Lo}}$, only the states reachable by $\pi_{\text{Lo}}^*$ can be explored sufficiently by $\text{Alg}^{\text{Lo}}$ due to its near optimal regret. Combining with Assump. B, one can observe that the value difference on transferable states are controlled by $|V_{\text{Lo},h}^*(s_h) - V_{\text{Hi},h}^*(s_h)| = O(\frac{\widetilde{\Delta}_{\min}}{H}) \leq O(\frac{\Delta_{\min}}{H})$. As we will show in Thm. 3.2 in Sec. 3, the term $O(\Delta_{\min})$ is indeed unimprovable if we expect robustness.

## 3 Lower Bound Results: Necessary Condition for Robust Transfer

Now we establish lower bounds that show Assump. B is necessary and how the magnitude of $\Delta_{\min}$ restricts the robust transfer objective. The results in this section are based on two-armed Bernoulli bandits for simplicity, and the proofs are deferred to Appx. B. By extending these hard instances to RL case, there is a gap caused by the additional $\frac{1}{H}$ in Assump. B and Def. 2.2, which comes from the requirement of our algorithm design, and we leave it to the future work.

**Justification for Assump. B** We show that if Assump. B is violated, it is impossible to have algorithms $(\text{Alg}^{\text{Lo}}, \text{Alg}^{\text{Hi}})$ to simultaneously achieve constant regret when $M_{\text{Lo}} = M_{\text{Hi}}$, while retaining sub-linear regret for all the cases regardless of the similarity between $M_{\text{Lo}}$ and $M_{\text{Hi}}$. Here we require constant regret on $M_{\text{Lo}} = M_{\text{Hi}}$ since we believe it is a minimal expectation to achieve benefits in transfer when

---

[2]For convenience, we include the bandit setting as a special case with $H = 0$; see also Def. 2.1, 2.2 and F.1.

the two tasks are identical. Intuitively, without Assump. B, even if we know $\mu_{\mathrm{Lo}}(i^*_{\mathrm{Lo}}) = \mu_{\mathrm{Hi}}(i^*_{\mathrm{Lo}})$, we cannot ensure $i^*_{\mathrm{Lo}}$ is the optimal arm in $M_{\mathrm{Hi}}$. Then, if $(\mathrm{Alg}^{\mathrm{Lo}}, \mathrm{Alg}^{\mathrm{Hi}})$ can achieve constant regret on $M_{\mathrm{Lo}} = M_{\mathrm{Hi}}$, the algorithm must stop exploration on the arm $i \neq i^*_{\mathrm{Lo}}$ after finite steps, and thus, it suffers from linear regret in another instance of $M_{\mathrm{Hi}}$ where $i^*_{\mathrm{Lo}} \neq i^*_{\mathrm{Hi}}$.

Moreover, Thm. 3.1 holds even if the learner has full information of $M_{\mathrm{Lo}}$, where the setting degenerates to normal transfer RL since there is no need to explore $M_{\mathrm{Lo}}$. This explains why similar assumptions to Assump. B are considered in previous transfer RL works [9, 10].

**Theorem 3.1.** *Under the violation of Assump. B, even regardless of the optimality of $\mathrm{Alg}^{\mathrm{Lo}}$, for each algorithm pair $(\mathrm{Alg}^{\mathrm{Lo}}, \mathrm{Alg}^{\mathrm{Hi}})$, it cannot simultaneously (1) achieve constant regret for the case when $M_{\mathrm{Lo}} = M_{\mathrm{Hi}}$ and (2) ensure sub-linear regret in all the other cases.*

$\Delta_{\min}$ **in Tolerance Error is Inevitable**   Next, we show that, if $M_{\mathrm{Hi}}$ and $M_{\mathrm{Lo}}$ are $\Delta$-close for some $\Delta \geq \frac{\Delta_{\min}}{2}$, in general we cannot expect to achieve constant regret on $M_{\mathrm{Hi}}$ by leveraging $M_{\mathrm{Lo}}$ without other loss. Similar to Thm. 3.1, the main idea is to construct different instances for $M_{\mathrm{Hi}}$ with different optimal arms and cannot be distinguished within finite number of trials.

**Theorem 3.2.** *[Transferable States are Restricted by $\Delta_{\min}$] Under Assump. B, regardless of the optimality of $\mathrm{Alg}^{\mathrm{Lo}}$, given arbitrary $\Delta_{\min}$ and arbitrary $\Delta \in [\frac{\Delta_{\min}}{2}, \Delta_{\min}]$, for each algorithm pair $(\mathrm{Alg}^{\mathrm{Lo}}, \mathrm{Alg}^{\mathrm{Hi}})$, it cannot simultaneously (1) achieve constant regret for the case when $M_{\mathrm{Lo}}$ and $M_{\mathrm{Hi}}$ with minimal gap $\Delta_{\min}$ are $\Delta$-close, and (2) ensure sub-linear regret in all other cases.*

## 4 Robust Tiered MAB/RL with Single Source Task

In this section, we study Tiered MAB and Tiered RL when a single low-tier task $M_{\mathrm{Lo}}$ is available. The key challenge compared with [13] is that we do not have knowledge about whether $M_{\mathrm{Lo}}$ and $M_{\mathrm{Hi}}$ are similar or not so the pure exploitation will not work. Instead, the algorithm should be able to identify whether $M_{\mathrm{Lo}}$ and $M_{\mathrm{Hi}}$ are close enough to transfer by data collected so far, and balance between the exploration by itself and the exploitation from $M_{\mathrm{Lo}}$ at the same time.

To overcome the challenge, we identify a state-wise checking event, such that, under Assump. B, if $s_h$ is transferable, the event is true almost all the time, and otherwise, every mistake will reduce the uncertainty so the chance the event holds is limited. By utilizing it, our algorithm can wisely switch between optimistic exploration and pessimistic exploitation and achieve robust transfer. In Sec. 4.1, we start with the MAB setting and illustrate the main idea, and in Sec. 4.2, we generalize our techniques to RL setting, and discuss how to overcome the challenges brought by state transition.

### 4.1 Robust Transfer in Tiered Multi-Armed Bandits

The algorithm is provided in Alg. 1. We choose $\mathrm{Alg}^{\mathrm{Lo}}$ as UCB, and $\mathrm{Alg}^{\mathrm{Hi}}$ as an exploitation-or-UCB style algorithm branched by a checking event in line 7, which is the key step to avoid negative transfer.

---
**Algorithm 1:** Robust Tiered MAB
---
1 **Initialize**: $\alpha > 2$; $N^1_{\mathrm{Lo}}(i), N^1_{\mathrm{Hi}}(i), \widehat{\mu}^1_{\mathrm{Lo}}(i), \widehat{\mu}^1_{\mathrm{Hi}}(i) \leftarrow 0, \ \forall i \in \mathcal{A}$; $f(k) := 1 + 16A^2(k+1)^2$
2 Pull each arm at the beginning $A$ iterations
3 **for** $k = A+1, A+2, ..., K$ **do**
4 $\quad \overline{\mu}^k_{\mathrm{Lo}}(i) \leftarrow \widehat{\mu}^k_{\mathrm{Lo}}(i) + \sqrt{\frac{2\alpha \log f(k)}{N^k_{\mathrm{Lo}}(i)}}, \quad \overline{\pi}^k_{\mathrm{Lo}} \leftarrow \arg\max_i \overline{\mu}^k_{\mathrm{Lo}}(i), \quad \pi^k_{\mathrm{Lo}} \leftarrow \overline{\pi}^k_{\mathrm{Lo}}.$
5 $\quad \underline{\mu}^k_{\mathrm{Lo}}(i) \leftarrow \widehat{\mu}^k_{\mathrm{Lo}}(i) - \sqrt{\frac{2\alpha \log f(k)}{N^k_{\mathrm{Lo}}(i)}}, \quad \underline{\pi}^k_{\mathrm{Lo}} \leftarrow \arg\max_i \underline{\mu}^k_{\mathrm{Lo}}(i).$
6 $\quad \overline{\mu}^k_{\mathrm{Hi}}(i) \leftarrow \widehat{\mu}^k_{\mathrm{Hi}}(i) + \sqrt{\frac{2\alpha \log f(k)}{N^k_{\mathrm{Hi}}(i)}}, \quad \overline{\pi}^k_{\mathrm{Hi}} \leftarrow \arg\max_i \overline{\mu}^k_{\mathrm{Hi}}(i).$
7 $\quad$ **if** $\underline{\mu}^k_{\mathrm{Lo}}(\underline{\pi}^k_{\mathrm{Lo}}) \leq \overline{\mu}^k_{\mathrm{Hi}}(\underline{\pi}^k_{\mathrm{Lo}}) + \varepsilon$ and $N^k_{\mathrm{Lo}}(\underline{\pi}^k_{\mathrm{Lo}}) > k/2$ **then** $\pi^k_{\mathrm{Hi}} \leftarrow \underline{\pi}^k_{\mathrm{Lo}}$ **else** $\pi^k_{\mathrm{Hi}} \leftarrow \overline{\pi}^k_{\mathrm{Hi}}$.
8 $\quad$ Interact $M_{\mathrm{Hi}}/M_{\mathrm{Lo}}$ by $\pi^k_{\mathrm{Hi}}/\pi^k_{\mathrm{Lo}}$; Update $N^{k+1}_{\mathrm{Lo}}/N^{k+1}_{\mathrm{Hi}}$ and empirical mean $\widehat{\mu}^{k+1}_{\mathrm{Lo}}/\widehat{\mu}^{k+1}_{\mathrm{Hi}}$.
9 **end**

---

**Key Insights: Separation between Transferable and Non-Transferable Cases**   To understand our checking event, we consider the following two cases: (1) $M_{\mathrm{Lo}}$ and $M_{\mathrm{Hi}}$ are $\varepsilon$-close, and (2) $i^*_{\mathrm{Hi}} \neq i^*_{\mathrm{Lo}}$

(in the rest cases we have $i_{\text{Hi}}^* = i_{\text{Lo}}^*$ but $\mu_{\text{Lo}}(i_{\text{Lo}}^*) > \mu_{\text{Hi}}(i_{\text{Lo}}^*) + \varepsilon$, so exploiting from $M_{\text{Lo}}$ is harmless). Recall Assump. B, in Case 1, we have $\mu_{\text{Lo}}(i_{\text{Lo}}^*) \leq \mu_{\text{Hi}}(i_{\text{Lo}}^*) + \varepsilon$, while in Case 2, with an appropriate choice of $\varepsilon$ (e.g. $\varepsilon = \frac{\widetilde{\Delta}_{\min}}{4} < \frac{\Delta_{\min}}{4}$), we have $\mu_{\text{Lo}}(i_{\text{Lo}}^*) \geq \mu_{\text{Hi}}(i_{\text{Hi}}^*) - \frac{\Delta_{\min}}{2} \geq \mu_{\text{Hi}}(i_{\text{Lo}}^*) + \varepsilon + \frac{\Delta_{\text{Hi}}(i_{\text{Lo}}^*)}{2}$, which reveals the separation between two cases. As a result, if we can construct an uncertainty-based upper bound $\overline{\mu}_{\text{Hi}}^k(i_{\text{Lo}}^*)$ for $\mu_{\text{Hi}}(i_{\text{Lo}}^*)$, we should expect the event $\mathcal{E} := \mu_{\text{Lo}}(i_{\text{Lo}}^*) < \overline{\mu}_{\text{Hi}}^k(i_{\text{Lo}}^*) + \varepsilon$ almost always be true in Case 1, while in Case 2, everytime $\mathcal{E}$ occurs and $\text{Alg}^{\text{Hi}}$ takes $i_{\text{Lo}}^*$, the "self-correction" is triggered: the uncertainty is reduced so the estimation $\overline{\mu}_{\text{Hi}}^k(i_{\text{Lo}}^*)$ gets closer to $\mu_{\text{Lo}}(i_{\text{Lo}}^*)$, and because of the separation between $\mu_{\text{Lo}}(i_{\text{Lo}}^*)$ and $\mu_{\text{Hi}}(i_{\text{Lo}}^*)$, the number of times that $\mathcal{E}$ is true is limited. The remaining issue is that we do not know $\mu_{\text{Lo}}(i_{\text{Lo}}^*)$ and $i_{\text{Lo}}^*$, and we approximate them with LCB value $\underline{\mu}_{\text{Lo}}^k(\cdot)$ and its greedy policy. The additional checking event $N_{\text{Lo}}^k(\underline{\pi}_{\text{Lo}}^k) > k/2$ is used to increase the confidence that $\underline{\pi}_{\text{Lo}}^k = i_{\text{Lo}}^*$ once transfer, which also contributes to reducing the regret. Finally, to achieve constant regret, we use $\alpha$ to control the total failure rate to be $\sum_{k=1}^{+\infty} k^{-\Theta(\alpha)} = C$ for some constant $C$. We summarize the main result in Thm. 4.1 and defer the proof to Appx. C.

**Theorem 4.1.** *[Tiered MAB with Single Source Tasks] Under Assump. A, B and C, by running Alg. 1 with $\varepsilon = \frac{\widetilde{\Delta}_{\min}}{4}$ and $\alpha > 2$, we always have $Regret_K(M_{Hi}) = O(\sum_{\Delta_{Hi}(i)>0} \frac{1}{\Delta_{Hi}(i)} \log K)$. Moreover, if $M_{Hi}$ and $M_{Lo}$ are $\frac{\widetilde{\Delta}_{\min}}{4}$-close, we have: $Regret_K(M_{Hi}) = O(\sum_{\Delta_{Hi}(i)>0} \frac{1}{\Delta_{Hi}(i)} \log \frac{A}{\Delta_{\min}})$.*

**Comparison with [22, 13]** As we can see, our algorithm can automatically achieve constant regret if tasks are similar while retaining near-optimal otherwise. Notably, even when $M_{\text{Hi}} = M_{\text{Lo}}$, our regret bound $\widetilde{O}(\sum_{\Delta_{\text{Hi}}(i)>0} \frac{1}{\Delta_{\text{Hi}}(i)})$ is strictly better than $\widetilde{O}(\sqrt{\frac{A}{\Delta_{\min}}} \sqrt{\sum_{\Delta_{\text{Hi}}(i)>0} \frac{1}{\Delta_{\text{Hi}}(i)}})$ in [22] and $\widetilde{O}(\sum_{\Delta_{\text{Hi}}(i)>0}(A-i)(\frac{1}{\Delta_{\text{Hi}}(i)} - \frac{\Delta_{\text{Hi}}(i)}{\Delta_{\text{Hi}}(i-1)^2}))$ in [13] under the same setting.

## 4.2 Robust Transfer in Tiered Tabular RL

In this section, we focus on RL setting. We provide the algorithm in Alg. 2, where we defer the details of **ModelLearning** function and the requirements for **Bonus** function to Appx. D.1. In the following, we first highlight our main result.

**Theorem 4.2.** *[Tiered RL with Single Source Tasks] Under Assump. A, B and C, Cond. D.1 for $Alg^{Lo}$ and Cond. D.3 for **Bonus** function, by running Alg. 2 with $\varepsilon = \frac{\widetilde{\Delta}_{\min}}{4(H+1)}$, $\alpha > 2$, an arbitrary $\lambda > 0$, we have*

$$Regret_K(M_{Hi}) = O\Big(SH \sum_{h=1}^{H} \sum_{(s_h,a_h) \in \mathcal{S}_h \times \mathcal{A}_h \setminus \mathcal{C}_h^\lambda} (\frac{H}{\Delta_{\min}} \wedge \frac{1}{\Delta_{Hi}(s_h,a_h)}) \log(SAHK)\Big).$$

Here the set $\mathcal{C}_h^\lambda \subset \mathcal{S}_h \times \mathcal{A}_h$ captures the benefitable state-action pairs to be introduced later. For simplicity, in Thm. 4.2 above, we omit all constant terms independent with $K$ that may include $\lambda^{-1}$, $\Delta_{\min}^{-1}$ or $\log 1/d_h^*(\cdot)$. The complete version of Thm. 4.2 can be found in Thm. D.16. As we can see, comparing with pure online learning algorithms [23, 34, 7], in our setting, $M_{\text{Hi}}$ only suffers non-constant regret on a subset of the state-action space. The $SH$ factor may be further improved by choosing better **Bonus** functions than our Example D.4 given in Appx. D.1.

Different from the bandit setting, the state transition causes more challenges. In the following, we first explain the algorithm design to highlight how we overcome the difficulties, and then provide the analysis and proof sketch. Detailed proofs can be found in Appx. D.

**Technical Challenges and Algorithm Design** Similar to MAB setting, for $M_{\text{Lo}}$ we choose an arbitrary near-optimal algorithm, and for $M_{\text{Hi}}$ we set up a state-wise checking event $\underline{V}_{\text{Lo},h}^k(\cdot) \leq \widetilde{Q}_{\text{Hi},h}^k(\cdot, \pi_{\text{Lo},h}^k) + \varepsilon$ in line 15 to determine whether to exploit from $M_{\text{Lo}}$ or not. Here $\underline{V}_{\text{Lo},h}^k$ and $\widetilde{Q}_{\text{Hi},h}^k$ serve as lower and upper bounds for $V_{\text{Lo}}^*$ and $Q_{\text{Hi}}^*$, and $\underline{V}_{\text{Lo},h}^k$ and $\pi_{\text{Lo},h}^k$ are constructed by Pessimistic Value Iteration [16], which can be shown to converge to $V_{\text{Lo},h}^*$ and $\pi_{\text{Lo},h}^*$, respectively. Similar to [13], for the choice of $\text{Alg}^{\text{Lo}}$ and the bonus term used to construct lower/upper confidence estimation, we consider general algorithm framework under Cond. D.1 and Cond. D.3 in Appx. D.1.

To overcome challenges resulting from state transition, we make two major modifications when moving from MAB to RL setting. First of all, because of the constraint on the optimality of $\text{Alg}^{\text{Lo}}$,

---

**Algorithm 2:** Robust Tiered RL

---

1 **Input**: Ratio $\lambda \in (0,1)$; $\alpha > 2$; Auxiliary functions **Bonus** and **ModelLearning**; Sequence of confidence level $(\delta_k)_{k\geq 1}$ with $\delta_k = 1/SAHk^{\alpha}$; $\varepsilon := \widetilde{\Delta}_{\min}/4(H+1)$ for some $\widetilde{\Delta}_{\min} \leq \Delta_{\min}$

2 **Initialize**: $D_{\mathrm{Lo}}^0/D_{\mathrm{Hi}}^0 \leftarrow \{\}$; $\forall k,\ \underline{V}_{(\cdot),H+1}^k, \underline{Q}_{(\cdot),H+1}^k, \widetilde{V}_{\mathrm{Hi},H+1}^k, \widetilde{Q}_{\mathrm{Hi},H+1}^k \leftarrow 0$.

3 **for** $k = 1, 2, \ldots$ **do**

4     $\pi_{\mathrm{Lo}} \leftarrow \mathrm{Alg}^{\mathrm{Lo}}(D_{\mathrm{Lo}}^{k-1})$;

5     $\{\widehat{\mathbb{P}}_{\mathrm{Lo},h}^k\}_{h=1}^H \leftarrow \mathbf{ModelLearning}(D_{\mathrm{Lo}}^{k-1})$, $\{b_{\mathrm{Lo},h}^k\}_{h=1}^H \leftarrow \mathbf{Bonus}(D_{\mathrm{Lo}}^{k-1}, \delta_k)$.

6     **for** $h = H, H-1 \ldots, 1$ **do**

7         $\underline{Q}_{\mathrm{Lo},h}^k(\cdot,\cdot) \leftarrow \max\{0, r_{\mathrm{Lo},h}(\cdot,\cdot) + \widehat{\mathbb{P}}_{\mathrm{Lo},h}^k \underline{V}_{\mathrm{Lo},h+1}^k(\cdot,\cdot) - b_{\mathrm{Lo},h}^k(\cdot,\cdot)\}$.

8         $\underline{V}_{\mathrm{Lo},h}^k(\cdot) = \max_a \underline{Q}_{\mathrm{Lo},h}^k(\cdot,a)$,    $\underline{\pi}_{\mathrm{Lo},h}^k(\cdot) \leftarrow \arg\max_a \underline{Q}_{\mathrm{Lo},h}^k(\cdot,a)$.

9     **end**

10     $\{\widehat{\mathbb{P}}_{\mathrm{Hi},h}^k\}_{h=1}^H \leftarrow \mathbf{ModelLearning}(D_{\mathrm{Hi}}^{k-1})$;    $\{b_{\mathrm{Hi},h}^k\}_{h=1}^H \leftarrow \mathbf{Bonus}(D_{\mathrm{Hi}}^{k-1}, \delta_k)$.

11     **for** $h = H, H-1 \ldots, 1$ **do**

12         $\underline{Q}_{\mathrm{Hi},h}^{\pi_{\mathrm{Hi}}^k}(\cdot,\cdot) \leftarrow \max\{0, r_{\mathrm{Hi},h}(\cdot,\cdot) + \widehat{\mathbb{P}}_{\mathrm{Hi},h}^k \underline{V}_{\mathrm{Hi},h+1}^k(\cdot,\cdot) - b_{\mathrm{Hi},h}^k(\cdot,\cdot)\}$

13         $\widetilde{Q}_{\mathrm{Hi},h}^k(\cdot,\cdot) \leftarrow \min\{H, r_{\mathrm{Hi},h}(\cdot,\cdot) + \widehat{\mathbb{P}}_{\mathrm{Hi},h}^k \widetilde{V}_{\mathrm{Hi},h+1}^k(\cdot,\cdot) + b_{\mathrm{Hi},h}^k(\cdot,\cdot)\}$.

14         **for** $s_h \in \mathcal{S}_h$ **do**

15             **if** $\underline{V}_{Lo,h}^k(s_h) \leq \widetilde{Q}_{Hi,h}^k(s_h, \underline{\pi}_{Lo,h}^k) + \varepsilon$ *and* $\max_a N_{Lo,h}^k(s_h, a) > \frac{\lambda}{3}k$ **then**

16                 $\pi_{\mathrm{Hi}}^k(s_h) \leftarrow \arg\max_a N_{\mathrm{Lo},h}^k(s_h, a)$. // "Trust and Exploit" Branch

17             **end**

18             **else** $\pi_{\mathrm{Hi}}^k(\cdot) \leftarrow \arg\max_a \widetilde{Q}_{\mathrm{Hi},h}^k(\cdot, a)$. // "Explore by itself" Branch ;

19             $\widetilde{V}_{\mathrm{Hi},h}^k(s_h) \leftarrow \min\{H, \widetilde{Q}_{\mathrm{Hi},h}^k(s_h, \pi_{\mathrm{Hi}}^k) + \frac{1}{H}(\widetilde{Q}_{\mathrm{Hi},h}^k(s_h, \pi_{\mathrm{Hi}}^k) - \underline{Q}_{\mathrm{Hi},h}^{\pi_{\mathrm{Hi}}^k}(s_h, \pi_{\mathrm{Hi}}^k))\}$

20             $\underline{V}_{\mathrm{Hi},h}^{\pi_{\mathrm{Hi}}^k}(s_h) = \underline{Q}_{\mathrm{Hi},h}^{\pi_{\mathrm{Hi}}^k}(s_h, \pi_{\mathrm{Hi}}^k)$.

21         **end**

22     **end**

23     Deploy $\pi_{\mathrm{Lo}}/\pi_{\mathrm{Hi}}$ to $M_{\mathrm{Lo}}/M_{\mathrm{Hi}}$ and get $\tau_{\mathrm{Lo}}^k/\tau_{\mathrm{Hi}}^k$; and update $D_{\mathrm{Lo}}^k, D_{\mathrm{Hi}}^k$.

24 **end**

---

we cannot expect $M_{\mathrm{Lo}}$ to provide useful information on those $(s_h, a_h)$ with $d_{\mathrm{Lo}}^*(s_h, a_h) = 0$ since they will not be explored sufficiently. Therefore, in the checking event in line 15, we include $\max_a N_{\mathrm{Lo},h}^k(s_h, a) > \Theta(\lambda k)$ as a criterion, where $\lambda$ is a hyper-parameter chosen and input to the algorithm. Intuitively, for all $s_h, a_h$, we should expect $N_{\mathrm{Lo},h}^k(s_h, a_h) \approx \widetilde{O}(d_{\mathrm{Lo}}^*(s_h, a_h) \cdot k)$ when $k$ is large enough. Therefore, by comparing $N_{\mathrm{Lo},h}^k$ with $\lambda k$, we can filter out those $s_h$ with $d_{\mathrm{Lo}}^*(s_h) < \lambda$ to avoid harm from inaccurate estimation.

Secondly and more importantly, different from MAB setting, besides the error occurred at a particular step, we also need to handle the error accumulated during the back-propagation process of value iteration. In our case, this is reflected by the loss of overestimation when we incorporate selective exploitation into the optimism-based exploration framework. To see this, suppose at some $s_h$, we have an overestimation on optimal value $Q_{\mathrm{Hi},h}^*$ denoted as $\widetilde{Q}_{\mathrm{Hi},h}^k$. When the checking criterion is satisfied, if we mimic the MAB setting, i.e., assign $\pi_{\mathrm{Lo},h}^k$ to $\pi_{\mathrm{Hi},h}^k$ and update value by $\widetilde{V}_{\mathrm{Hi},h}^k(s_h) \leftarrow \widetilde{Q}_{\mathrm{Hi},h}^k(s_h, \pi_{\mathrm{Hi},h}^k)$, when $\pi_{\mathrm{Lo}}^*(s_h) \neq \pi_{\mathrm{Hi},h}^*(s_h)$, $\widetilde{V}_{\mathrm{Hi},h}^k(s_h)$ is no longer guaranteed to be an overestimation for $V_{\mathrm{Hi},h}^*(s_h)$. As $\widetilde{V}_{\mathrm{Hi},h}^k(s_h)$ involves in back-propagation, it will pull down the estimation value for its ancestor states, thus reducing the chance to visit $s_h$ and slowing down the "self-correction process" which works well in MAB setting.

The key insight to overcome such difficulty is that, if the checking event holds yet $\pi_{\mathrm{Lo}}^*(s_h) \neq \pi_{\mathrm{Hi},h}^*(s_h)$, the gap between $\widetilde{Q}_{\mathrm{Hi},h}^k(s_h, \pi_{\mathrm{Lo}}^*)$ and $Q_{\mathrm{Hi},h}^*(s_h, \pi_{\mathrm{Lo}}^*)$ should not be small, and we can show that $\widetilde{Q}_{\mathrm{Hi},h}^k(s_h, \underline{\pi}_{\mathrm{Lo}}^k) \approx \widetilde{Q}_{\mathrm{Hi},h}^k(s_h, \pi_{\mathrm{Lo}}^*) \geq Q_{\mathrm{Hi},h}^*(s_h, \pi_{\mathrm{Lo}}^*) + \Theta(\frac{H}{H+1}\Delta_{\mathrm{Hi}}(s_h, \pi_{\mathrm{Lo}}^*))$ with the choice of

$\varepsilon = O(\widetilde{\Delta}_{\min}/H)$. Therefore, revising $\widetilde{Q}^k_{\text{Hi},h}(s_h, \underline{\pi}^k_{\text{Lo}})$ by adding $1/H$ of the gap $\widetilde{Q}^k_{\text{Hi},h}(s_h, \underline{\pi}^k_{\text{Lo}}) - Q^*_{\text{Hi},h}(s_h, \pi^*_{\text{Lo}})$ (line 19) is enough to guarantee the overestimation. Lastly, since $Q^*_{\text{Hi},h}(s_h, \pi^*_{\text{Lo}})$ in unknown, we construct an underestimation $\underline{Q}^{\pi^k_{\text{Hi}}}_{\text{Hi},h}(s_h, \pi^*_{\text{Lo}})$ and use it instead. As a result, we have the following theorem, where the clip function is defined by $\text{Clip}[x|w] := x \cdot \mathbb{I}[x \geq w]$.

**Theorem 4.3.** *There exists $k_{ost} = Poly(S, A, H, \lambda^{-1}, \Delta_{\min}^{-1})$, such that, for all $k \geq k_{ost}$, on some event $\mathcal{E}_k$ with $\mathbb{P}(\mathcal{E}_k) \leq 3\delta_k$, we have $Q^*_{Hi,h}(s_h, a_h) \leq \widetilde{Q}^k_{Hi,h}(s_h, a_h)$, $V^*_{Hi,h}(s_h) \leq \widetilde{V}^k_{Hi,h}(s_h), \forall h \in [H]$, $s_h \in \mathcal{S}_h$, $a_h \in \mathcal{A}_h$ and*

$$V^*_{Hi,1}(s_1) - V^{\pi^k_{Hi}}_{Hi,1}(s_1) \leq 2e\mathbb{E}_{\pi^k_{Hi}}\left[\sum_{h=1}^{H} Clip\left[\min\{H, 4b^k_{Hi,h}(s_h, a_h)\}\Big| \frac{\Delta_{\min}}{4eH} \vee \frac{\Delta_{Hi}(s_h, a_h)}{4e}\right]\right]. \quad (1)$$

**Benefits of Knowledge Transfer**   We first take a look at $k \geq k_{ost}$. As implied from Eq. (1), we can upper bound the regret on each $s_h, a_h$ by summing over the RHS of Eq. (1). Note that by Cond. D.3, $b^k_{\text{Hi},h}(s_h, a_h) = O\big(\frac{\text{Poly}(SAH)\log k}{\sqrt{N^k_{\text{Hi},h}(s_h, a_h)}}\big)$ and $\mathbb{E}[N^k_{\text{Hi},h}(s_h, a_h)] = \sum_{k'=1}^{k-1} d^{\pi^k_{\text{Hi}}}(s_h, a_h)$, we can establish the near-optimal regret bound with similar techniques in [23] regardless of the similarity between $M_{\text{Hi}}$ and $M_{\text{Lo}}$. Moreover, because of the knowledge transfer from $M_{\text{Lo}}$, we can achieve better regret bounds for $M_{\text{Hi}}$. In the following, we characterize three subclasses of state-action pairs, on which $\text{Alg}^{\text{Hi}}$ only suffers constant regret. *First of all*, for those $s_h \in \mathcal{Z}^\lambda_h$, we can expect the checking event almost always hold for arbitrary $k$. Hence, when $k$ is large enough, $\pi^k_{\text{Hi},h}(s_h) = \underline{\pi}^k_{\text{Lo},h}(s_h) \approx \pi^*_{\text{Hi},h}(s_h)$, implying $\text{Alg}^{\text{Hi}}$ will almost never take sub-optimal actions at $s_h$ since then. We denote this first subclass as $\mathcal{C}^{1,\lambda}_h := \{(s_h, a_h)|s_h \in \mathcal{Z}^\lambda_h, a_h \neq \pi^*_{\text{Hi},h}(s_h)\}$. *Secondly*, note that, given a state $s_h$, if all possible trajectories starting from $s_1$ to $s_h$ have overlap with $\mathcal{C}^{1,\lambda}_{h'}$ for some $h' \in [h-1]$, when $k$ is large enough, $\pi^k_{\text{Hi}}$ will almost have no chance to reach $s_h$ and will not suffer the regret at $s_h$. For convenience, we define function $\text{Block}(\{\mathcal{C}^{1,\lambda}_{h'}\}^{h-1}_{h'=1}, s_h)$ which takes $\texttt{True}$ for those states described above, and takes $\texttt{False}$ for the others. Then, we define the second subclass by $\mathcal{C}^{2,\lambda}_h := \{(s_h, a_h)|\text{Block}(\{\mathcal{C}^{1,\lambda}_{h'}\}^{h-1}_{h'=1}, s_h) = \texttt{True}, s_h \notin \mathcal{Z}^\lambda_h, a_h \in \mathcal{A}_h\}$. *Finally*, for those $s_h, a_h$ with $d^*_{\text{Hi}}(s_h, a_h) > 0$, we can show $N^k_{\text{Hi},h}(s_h, a_h) \approx d^*_{\text{Hi}}(s_h, a_h)k$. Therefore, $b^k_{\text{Hi},h}(s_h, a_h) \propto \log k/\sqrt{N^k_{\text{Hi},h}(s_h, a_h)}$ in Eq. (1) will decay and the clipping operator will take effect, which leads to constant regret. This third subclass is denoted by $\mathcal{C}^*_h := \{(s_h, a_h)|d^*_{\text{Hi}}(s_h, a_h) > 0\}$. Based on the above discussion, we define $\mathcal{C}^\lambda_h := \mathcal{C}^{\lambda,1}_h \cup \mathcal{C}^{\lambda,2}_h \cup \mathcal{C}^*_h$ to be the benefitable states set in Thm. 4.2.

For $k \leq k_{ost}$, for the lack of overestimation, we simply use $H$ to upper bound the value gap $V^* - V^{\pi^k_{\text{Hi}}}$. This results in a $Poly(S, A, H, \lambda^{-1}, \Delta_{\min}^{-1})$ burn-in term, which was omitted in Thm. 4.2 since it is independent with $K$. Besides, by the definition of $k_{ost}$ in Thm. 4.3, we can see the trade-off of choosing $\lambda$: a smaller $\lambda$ can enlarge $\mathcal{C}^\lambda_h$ so we have constant regret on more state-action pairs, while it also results in the delay of overestimation by the larger $k_{ost}$.

**Constant Regret in the Entire MDP**   We may expect constant regret in the entire $M_{\text{Hi}}$ in some special cases. Note that, if $\forall h \in [H], \forall s_h$ with $d^*_{\text{Hi}}(s_h) > 0$, $s_h \in \mathcal{Z}^\lambda_h$, we have $\mathcal{C}^\lambda_h = \mathcal{S}_h \times \mathcal{A}_h$, $\text{Regret}_K(M_{\text{Hi}})$ will be independent w.r.t. $K$. From this perspective, if $\lambda$ is chosen appropriately, e.g. $\lambda \leq \min_{s_h} d^*_{\text{Hi}}(s_h)$, we can recover the constant regret under the setting $M_{\text{Lo}} = M_{\text{Hi}}$ in [13].

**Choice of $\lambda$**   In this paper, we do not treat $\lambda$ as a parameter to optimize. In practice, without prior knowledge about $\max_{s_h} d^*_{\text{Lo}}(s_h)$, one may choose $\lambda = O(1/S)$ to ensure some chance that transferable states exist, since there exists at least some states satisfying $d^*_{\text{Lo}}(s_h) \geq 1/S$.

## 5   Robust Tiered MAB/RL with Multiple Low-Tier Tasks

Now, we focus on the case when a source task set $\mathcal{M}_{\text{Lo}} := \{M_{\text{Lo},w}\}^W_{w=1}$ is available (see Frw. 5 in Appx. A). Our objective is to achieve benefits on those states $s_h$ as long as there exists some task $w \in [W]$ such that $M_{\text{Lo},w}$ and $M_{\text{Hi}}$ are close on $s_h$, while retaining near-optimal regret in other cases under Assump. B. The key challenge comparing with single task case is that, $\text{Alg}^{\text{Hi}}$ has to identify for each state which task in $\mathcal{M}_{\text{Lo}}$ is the appropriate one to leverage. The main novelty and contribution

in this section is a task selection mechanism we call *"Trust till Failure"*, which can automatically adapt to the similar task if it exists. We first highlight the main results for MAB and RL setting.

**Theorem 5.1.** *[Tiered MAB with Multiple Source Tasks] Under Assump. A, B, and C, by running Alg. 3 with $\mathcal{M}_{Lo} = \{M_{Lo,w}\}_{w=1}^{W}$ and $M_{Lo}$, with $\varepsilon = \frac{\widetilde{\Delta}_{\min}}{4}$ and $\alpha > 2$, we always have: $Regret_K(M_{Hi}) = O(\sum_{\Delta_{Hi}(i) > 0} \frac{1}{\Delta_{Hi}(i)} \log(WK))$. Moreover, if at least one task in $\mathcal{M}_{Lo}$ is $\frac{\widetilde{\Delta}_{\min}}{4}$-close to $M_{Hi}$, we further have: $Regret_K(M_{Hi}) = O(\sum_{\Delta_{Hi}(i) > 0} \frac{1}{\Delta_{Hi}(i)} \log \frac{AW}{\Delta_{\min}})$.*

**Theorem 5.2.** *[Tiered RL with Multiple Source Tasks] Under Assump. A, B, C, and Cond. D.3, F.4, by running Alg. 7 in Appx. F.1 with $\varepsilon = \frac{\widetilde{\Delta}_{\min}}{4(H+1)}$, $\alpha > 2$ and any $\lambda > 0$, we have*

$$Regret_K(M_{Hi}) = O\Big(SH \sum_{h=1}^{H} \sum_{(s_h, a_h) \in \mathcal{S}_h \times \mathcal{A}_h \setminus \mathcal{C}_h^{\lambda, [W]}} (\frac{H}{\Delta_{\min}} \wedge \frac{1}{\Delta_{Hi}(s_h, a_h)}) \log(SAHWK)\Big).$$

---

**Algorithm 3:** Robust Tiered MAB with Multiple Source Tasks

---
1  **Initialize**: $\alpha > 2$; $N_{Lo}^1(i)$, $N_{Hi}^1(i)$, $\widehat{\mu}_{Lo}^1(i)$, $\widehat{\mu}_{Hi}^1(i) \leftarrow 0$, $\forall i \in \mathcal{A}$; $f(k) := 1 + 16TA^2(k+1)^2$
2  Pull each arm at the beginning $A$ iterations. Set $w^A \leftarrow$ Null.
3  **for** $k = A + 1, 2, ..., K$ **do**
4      **for** $w = 1, 2..., W$ **do**
5          $\overline{\mu}_{Lo,w}^k(i) \leftarrow \widehat{\mu}_{Lo,w}^k(i) + \sqrt{\frac{2\alpha \log f(k)}{N_{Lo,w}(i)}}$,    $\overline{\pi}_{Lo,w}^k \leftarrow \arg\max_i \overline{\mu}_{Lo,w}^k(i)$,    $\pi_{Lo,w}^k \leftarrow \overline{\pi}_{Lo,w}^k$.
6          $\underline{\mu}_{Lo,w}^k(i) \leftarrow \widehat{\mu}_{Lo,w}^k(i) - \sqrt{\frac{2\alpha \log f(k)}{N_{Lo,w}(i)}}$,    $\underline{\pi}_{Lo,w}^k \leftarrow \arg\max_i \underline{\mu}_{Lo,w}^k(i)$.
7      **end**
8      $\overline{\mu}_{Hi}^k(i) \leftarrow \widehat{\mu}_{Hi}^k(i) + \sqrt{\frac{2\alpha \log f(k)}{N_{Hi}^k(i)}}$,    $\overline{\pi}_{Hi}^k \leftarrow \arg\max_i \overline{\mu}_{Hi}^k(i)$.
9      $\mathcal{I}^k \leftarrow \{w \in [W] | \underline{\mu}_{Lo,w}^k(\pi_{Lo,w}^k) \leq \overline{\mu}_{Hi}^k(\pi_{Lo,w}^k) + \varepsilon$ and $N_{Lo,w}^k(\underline{\pi}_{Lo,w}^k) > k/2\}$
10     **if** $\mathcal{I}^k = \emptyset$ **then** $w^k \leftarrow$ Null,    $\pi_{Hi}^k \leftarrow \overline{\pi}_{Hi}^k$ ;
11     **else**
12         **if** $w^{k-1} \neq$ Null and $w^{k-1} \in \mathcal{I}^k$ **then** $w^k \leftarrow w^{k-1}$,    $\pi_{Hi}^k \leftarrow \underline{\pi}_{Lo,w^k}^k$ ;
13         **else if** $w^{k-1} \neq$ Null and $\exists w \in \mathcal{I}^k$ such that $\pi_{Hi}^{k-1} = \arg\max_i N_{Lo,w}^k(i)$ **then**
14             $w^k \leftarrow w$,    $\pi_{Hi}^k \leftarrow \underline{\pi}_{Lo,w}^k$
15         **end**
16         **else** $w^k \sim \text{Unif}(\mathcal{I}^k)$,    $\pi_{Hi}^k \leftarrow \underline{\pi}_{Lo,w^k}^k$ ;
17     **end**
18     Interact with $M_{Hi}/\{M_{Lo,}\}_{w=1}^{W}$ by $\pi_{Hi}^k/\{\pi_{Lo,w}^k\}_{w=1}^{W}$;
19     Update $\{N_{Lo,w}^{k+1}\}_{w=1}^{W}$, $N_{Hi}^{k+1}$ and empirical mean $\{\widehat{\mu}_{Lo,w}^{k+1}\}_{w=1}^{W}$, $\widehat{\mu}_{Hi}^{k+1}$ for each arm.
20 **end**

---

For the lack of space, in the following, we only analyze the bandit setting to explain the key idea of our task selection strategy. For the RL setting, we defer to Appx. F the algorithm Alg. 7, detailed version of Thm. 5.2 (Thm. D.16), defintion of transferable set $\mathcal{C}_h^{\lambda, [W]}$ (Def. F.2), and technical details.

**Algorithm Design and Proof Sketch for Bandit Setting** The algorithm is provided in Alg. 3. Comparing with Alg. 1 in single task setting, the main difference is the task selection strategy from line 9 to line 17. We first examine each source task with a checking event similar to single task setting, and collect those feasible tasks passing the test to $\mathcal{I}^k$. Intuitively, for those $M_{Lo,w^*}$ close to $M_{Lo}$, we expect $M_{Lo,w} \in \mathcal{I}^k$ holds almostly for arbitrary $k > 0$, while for the other $M_{Lo,w'}$, if it takes the position of $w^k$, following $M_{Lo,w'}$ will reduce the uncertainty and it will be ruled out from $\mathcal{I}^k$, eventually. So we expect $w^k$ can "escape" from dissimilar source tasks but be absorbed to the similar task if exists. Therefore, if $\mathcal{I}^k$ is empty, $\text{Alg}^{Hi}$ will do exploration by itself. Otherwise, we choose one from $\mathcal{I}^k$ to transfer the action until it fails on the checking event. However, for any $\varepsilon$ the algorithm chosen, those "marginally similar" source tasks (denoted as $M_{Lo,\widetilde{w}}$), which are $\varepsilon'$-close to $M_{Hi}$ for some $\varepsilon'$ only slightly larger than $\varepsilon$, may cause some trouble. Because the checking event

will finally eliminate $M_{\mathrm{Lo},\widetilde{w}}$ since they are not $\varepsilon$-close, but it may occupy the position $w^k$ for a long time before elimination, especially when $\varepsilon'$ is extremely close to $\varepsilon$. After eliminating $M_{\mathrm{Lo},\widetilde{w}}$, $\mathrm{Alg}^{\mathrm{Hi}}$ needs to re-select one from $\mathcal{I}^k$. Now since other sub-optimal arms in $M_{\mathrm{Hi}}$ haven't been chosen for a long time and the confidence level $\delta_k = O(1/k^\alpha)$ is decreasing, $\mathcal{I}^k$ will include those dissimilar tasks again, which causes difficulty to identify the true similar task. To solve this issue, once the previous trusted task fails, we give priority to the task recommending the same action as the previous one (line 14). As a result, since $M_{\mathrm{Lo},\widetilde{w}}$ and $M_{\mathrm{Lo},w^*}$ share the optimal action, after the elimination of $M_{\mathrm{Lo},\widetilde{w}}$, we can expect $w^k$ to only switch among those tasks $M_{\mathrm{Lo},w}$ with $\pi^*_{\mathrm{Lo},w} = \pi^*_{\mathrm{Hi}}$. We highlight this technical novelty to Lem. 5.3 below, and defer all the proofs to Appx. E.

**Lemma 5.3.** *[Absorbing to Similar Task] Under Assump. A, B and C, there exists a constant $c^*$, s.t., if there exists at least one $w^* \in [W]$ such that $M_{Lo,w^*}$ is $\frac{\widetilde{\Delta}_{\min}}{4}$-close to $M_{Hi}$, by running Alg. 3 with $\varepsilon = \frac{\widetilde{\Delta}_{\min}}{4}$ and $\alpha > 2$, for any $k \geq k^* := c^* \frac{\alpha A}{\Delta_{\min}^2} \log \frac{\alpha A W}{\Delta_{\min}}$, we have $\Pr(\pi^k_{Hi} \neq i^*_{Hi}) = O(\frac{A}{k^{2\alpha-2}})$.*

# 6 Experiments

In this section, we evaluate our most representative algorithm, Alg. 7, in multiple source tasks setting.

**Experiments Setting**[3] We set $S = A = 3$ and $H = 5$. The details for construction of source and target tasks are defered to Appx. G. We adapt StrongEuler in [23] as online learning algorithm to solve source tasks, and use the bonus function in [23] as the bonus function in our Alg. 7. We evaluate our algorithm when $W = 0, 1, 2, 5$, where $W = 0$ means the high-tier task $M_{\mathrm{Hi}}$ is simply solved by normal online learning method (StrongEuler) without any parallel knowledge transfer. We choose $\lambda = 0.3 \approx 1/S$ in Alg. 7, and in the MDP instance we test, across all $S \cdot H = 15$ states, for $W = 1, 2, 5$, the number of transferable states would be 6, 9 and 13, respectively.

We choose iteration number $K = 1e7$, where we start the transfer since $k = 5e5$ to avoid large "burn-in" terms. As we can see, after the transfer starts, the regret in target task will suddenly increase for a while, because the target task has to make some mistakes and learn from it as a result of the model uncertainty. However, because of our algorithm design, the negative transfer will terminate after a very short period. As predicted by our theory, by adding more and more source tasks which can introduce new transferable states, the target task will suffer less and less regret.

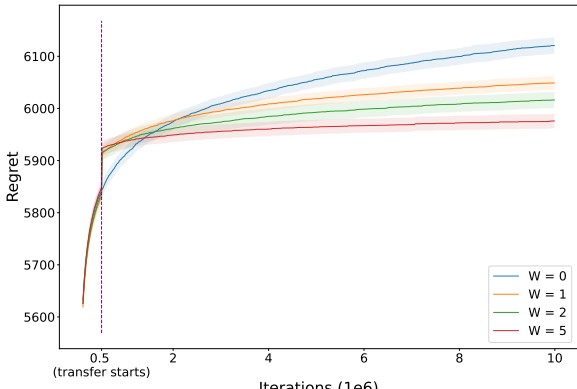

Figure 1: **Regret in the Target Task given Multiple Source Tasks** We report the result when $W$ source tasks are available with $W = 0, 1, 2, 5$. The shadows indicate 96% confidence interval.

# 7 Conclusion and Future Work

In this paper, we study how to do robust parallel transfer RL when single or multiple source tasks are avilable, without knowledge on models' similarity. The possible future directions include relaxing assumptions, better strategies to leveraging multiple source tasks, and identifying mild structural assumptions allowing for more aggressive transfer, and we defer to Appx. A.4 for more details.

---

[3]Code is available at `https://github.com/jiaweihhuang/Robust-Tiered-RL`

## Acknowledgments and Disclosure of Funding

The authors would like to thank Andreas Krause for valuable discussion. The work is supported by ETH research grant and Swiss National Science Foundation (SNSF) Project Funding No. 200021-207343.

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
