}}^*)$, we cannot ensure $i_{\text{Lo}}^*$ is the optimal arm in $M_{\text{Hi}}$. Then, if $(\text{Alg}^{\text{Lo}}, \text{Alg}^{\text{Hi}})$ can achieve constant regret on $M_{\text{Lo}} = M_{\text{Hi}}$, the algorithm must stop exploration on the arm $i \neq i_{\text{Lo}}^*$ after finite steps, and thus, it suffers from linear regret in another instance of $M_{\text{Hi}}$ where $i_{\text{Lo}}^* \neq i_{\text{Hi}}^*$.

Moreover, Thm. 3.1 holds even if the learner has full information of $M_{\text{Lo}}$, where the setting degenerates to normal transfer RL since there is no need to explore $M_{\text{Lo}}$. This explains why similar assumptions to Assump. B are considered in previous transfer RL works [9, 10].

**Theorem 3.1.** *Under the violation of Assump. B, even regardless of the optimality of $Alg^{Lo}$, for each algorithm pair $(Alg^{Lo}, Alg^{Hi})$, it cannot simultaneously (1) achieve constant regret for the case when $M_{Lo} = M_{Hi}$ and (2) ensure sub-linear regret in all the other cases.*

$\Delta_{\min}$ **in Tolerance Error is Inevitable**   Next, we show that, if $M_{\text{Hi}}$ and $M_{\text{Lo}}$ are $\Delta$-close for some $\Delta \geq \frac{\Delta_{\min}}{2}$, in general we cannot expect to achieve constant regret on $M_{\text{Hi}}$ by leveraging $M_{\text{Lo}}$ without other loss. Similar to Thm. 3.1, the main idea is to construct different instances for $M_{\text{Hi}}$ with different optimal arms and cannot be distinguished within finite number of trials.

**Theorem 3.2.** *[Transferable States are Restricted by $\Delta_{\min}$] Under Assump. B, regardless of the optimality of $Alg^{Lo}$, given arbitrary $\Delta_{\min}$ and arbitrary $\Delta \in [\frac{\Delta_{\min}}{2}, \Delta_{\min}]$, for each algorithm pair $(Alg^{Lo}, Alg^{Hi})$, it cannot simultaneously (1) achieve constant regret for the case when $M_{Lo}$ and $M_{Hi}$ with minimal gap $\Delta_{\min}$ are $\Delta$-close, and (2) ensure sub-linear regret in all other cases.*

## 4   Robust Tiered MAB/RL with Single Source Task

In this section, we study Tiered MAB and Tiered RL when a single low-tier task $M_{\text{Lo}}$ is available. The key challenge compared with [13] is that we do not have knowledge about whether $M_{\text{Lo}}$ and $M_{\text{Hi}}$ are similar or not so the pure exploitation will not work. Instead, the algorithm should be able to identify whether $M_{\text{Lo}}$ and $M_{\text{Hi}}$ are close enough to transfer by data collected so far, and balance between the exploration by itself and the exploitation from $M_{\text{Lo}}$ at the same time.

To overcome the challenge, we identify a state-wise checking event, such that, under Assump. B, if $s_h$ is transferable, the event is true almost all the time, and otherwise, every mistake will reduce the uncertainty so the chance the event holds is limited. By utilizing it, our algorithm can wisely switch between optimistic exploration and pessimistic exploitation and achieve robust transfer. In Sec. 4.1, we start with the MAB setting and illustrate the main idea, and in Sec. 4.2, we generalize our techniques to RL setting, and discuss how to overcome the challenges brought by state transition.

### 4.1   Robust Transfer in Tiered Multi-Armed Bandits

The algorithm is provided in Alg. 1. We choose $\text{Alg}^{\text{Lo}}$ as UCB, and $\text{Alg}^{\text{Hi}}$ as an exploitation-or-UCB style algorithm branched by a checking event in line 7, which is the key step to avoid negative transfer.

---

**Algorithm 1:** Robust Tiered MAB

---

1 **Initialize:** $\alpha > 2$; $N_{\text{Lo}}^1(i), N_{\text{Hi}}^1(i), \widehat{\mu}_{\text{Lo}}^1(i), \widehat{\mu}_{\text{Hi}}^1(i) \leftarrow 0, \forall i \in \mathcal{A}$; $f(k) := 1 + 16A^2(k+1)^2$

2 Pull each arm at the beginning $A$ iterations

3 **for** $k = A+1, A+2, ..., K$ **do**

4   $\quad \overline{\mu}_{\text{Lo}}^k(i) \leftarrow \widehat{\mu}_{\text{Lo}}^k(i) + \sqrt{\frac{2\alpha \log f(k)}{N_{\text{Lo}}^k(i)}}, \quad \overline{\pi}_{\text{Lo}}^k \leftarrow \arg\max_i \overline{\mu}_{\text{Lo}}^k(i), \quad \pi_{\text{Lo}}^k \leftarrow \overline{\pi}_{\text{Lo}}^k.$

5   $\quad \underline{\mu}_{\text{Lo}}^k(i) \leftarrow \widehat{\mu}_{\text{Lo}}^k(i) - \sqrt{\frac{2\alpha \log f(k)}{N_{\text{Lo}}^k(i)}}, \quad \underline{\pi}_{\text{Lo}}^k \leftarrow \arg\max_i \underline{\mu}_{\text{Lo}}^k(i).$

6   $\quad \overline{\mu}_{\text{Hi}}^k(i) \leftarrow \widehat{\mu}_{\text{Hi}}^k(i) + \sqrt{\frac{2\alpha \log f(k)}{N_{\text{Hi}}^k(i)}}, \quad \overline{\pi}_{\text{Hi}}^k \leftarrow \arg\max_i \overline{\mu}_{\text{Hi}}^k(i).$

7   $\quad$**if** $\underline{\mu}_{\text{Lo}}^k(\underline{\pi}_{\text{Lo}}^k) \leq \overline{\mu}_{\text{Hi}}^k(\underline{\pi}_{\text{Lo}}^k) + \varepsilon$ and $N_{\text{Lo}}^k(\underline{\pi}_{\text{Lo}}^k) > k/2$ **then** $\pi_{\text{Hi}}^k \leftarrow \underline{\pi}_{\text{Lo}}^k$ **else** $\pi_{\text{Hi}}^k \leftarrow \overline{\pi}_{\text{Hi}}^k.$

8   $\quad$Interact $M_{\text{Hi}}/M_{\text{Lo}}$ by $\pi_{\text{Hi}}^k/\pi_{\text{Lo}}^k$; Update $N_{\text{Lo}}^{k+1}/N_{\text{Hi}}^{k+1}$ and empirical mean $\widehat{\mu}_{\text{Lo}}^{k+1}/\widehat{\mu}_{\text{Hi}}^{k+1}.$

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

_{\mathrm{Hi},h}^k(s_h, a_h) = O(\frac{\mathrm{Poly}(SAH)\log k}{\sqrt{N_{\mathrm{Hi},h}^k(s_h, a_h)}})$ and $\mathbb{E}[N_{\mathrm{Hi},h}^k(s_h, a_h)] = \sum_{k'=1}^{k-1} d^{\pi_{\mathrm{Hi}}^k}(s_h, a_h)$, we can establish the near-optimal regret bound with similar techniques in [23] regardless of the similarity between $M_{\mathrm{Hi}}$ and $M_{\mathrm{Lo}}$. Moreover, because of the knowledge transfer from $M_{\mathrm{Lo}}$, we can achieve better regret bounds for $M_{\mathrm{Hi}}$. In the following, we characterize three subclasses of state-action pairs, on which $\mathrm{Alg}^{\mathrm{Hi}}$ only suffers constant regret. *First of all*, for those $s_h \in \mathcal{Z}_h^\lambda$, we can expect the checking event almost always hold for arbitrary $k$. Hence, when $k$ is large enough, $\pi_{\mathrm{Hi},h}^k(s_h) = \underline{\pi}_{\mathrm{Lo},h}^k(s_h) \approx \pi_{\mathrm{Hi},h}^*(s_h)$, implying $\mathrm{Alg}^{\mathrm{Hi}}$ will almost never take sub-optimal actions at $s_h$ since then. We denote this first subclass as $\mathcal{C}_h^{1,\lambda} := \{(s_h, a_h)|s_h \in \mathcal{Z}_h^\lambda, a_h \neq \pi_{\mathrm{Hi},h}^*(s_h)\}$. *Secondly*, note that, given a state $s_h$, if all possible trajectories starting from $s_1$ to $s_h$ have overlap with $\mathcal{C}_{h'}^{1,\lambda}$ for some $h' \in [h-1]$, when $k$ is large enough, $\pi_{\mathrm{Hi}}^k$ will almost have no chance to reach $s_h$ and will not suffer the regret at $s_h$. For convenience, we define function $\mathrm{Block}(\{\mathcal{C}_{h'}^{1,\lambda}\}_{h'=1}^{h-1}, s_h)$ which takes $\mathrm{True}$ for those states described above, and takes $\mathrm{False}$ for the others. Then, we define the second subclass by $\mathcal{C}_h^{2,\lambda} := \{(s_h, a_h)|\mathrm{Block}(\{\mathcal{C}_{h'}^{1,\lambda}\}_{h'=1}^{h-1}, s_h) = \mathrm{True}, s_h \notin \mathcal{Z}_h^\lambda, a_h \in \mathcal{A}_h\}$. *Finally*, for those $s_h, a_h$ with $d_{\mathrm{Hi}}^*(s_h, a_h) > 0$, we can show $N_{\mathrm{Hi},h}^k(s_h, a_h) \approx d_{\mathrm{Hi}}^*(s_h, a_h)k$. Therefore, $b_{\mathrm{Hi},h}^k(s_h, a_h) \propto \log k/\sqrt{N_{\mathrm{Hi},h}^k(s_h, a_h)}$ in Eq. (1) will decay and the clipping operator will take effect, which leads to constant regret. This third subclass is denoted by $\mathcal{C}_h^* := \{(s_h, a_h)|d_{\mathrm{Hi}}^*(s_h, a_h) > 0\}$. Based on the above discussion, we define $\mathcal{C}_h^\lambda := \mathcal{C}_h^{\lambda,1} \cup \mathcal{C}_h^{\lambda,2} \cup \mathcal{C}_h^*$ to be the benefitable states set in Thm. 4.2.

For $k \leq k_{ost}$, for the lack of overestimation, we simply use $H$ to upper bound the value gap $V^* - V^{\pi_{\mathrm{Hi}}^k}$. This results in a $Poly(S, A, H, \lambda^{-1}, \Delta_{\min}^{-1})$ burn-in term, which was omitted in Thm. 4.2 since it is independent with $K$. Besides, by the definition of $k_{ost}$ in Thm. 4.3, we can see the trade-off of choosing $\lambda$: a smaller $\lambda$ can enlarge $\mathcal{C}_h^\lambda$ so we have constant regret on more state-action pairs, while it also results in the delay of overestimation by the larger $k_{ost}$.

**Constant Regret in the Entire MDP**   We may expect constant regret in the entire $M_{\mathrm{Hi}}$ in some special cases. Note that, if $\forall h \in [H], \forall s_h$ with $d_{\mathrm{Hi}}^*(s_h) > 0$, $s_h \in \mathcal{Z}_h^\lambda$, we have $\mathcal{C}_h^\lambda = \mathcal{S}_h \times \mathcal{A}_h$, $\mathrm{Regret}_K(M_{\mathrm{

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

# Contents

# A  Extended Introduction

## A.1  Tiered-RL Framework

---
**Algorithm 4:** The Tiered RL Framework with Single Low-Tier Task

---
1 Initialize $D_{\mathrm{Lo}}^1, D_{\mathrm{Hi}}^1 \leftarrow \{\}$.
2 **for** $k = 1, 2, ..., K$ **do**
3     $\pi_{\mathrm{Lo}}^k \leftarrow \mathrm{Alg}^{\mathrm{Lo}}(D_{\mathrm{Lo}}^k)$;    $\pi_{\mathrm{Lo}}^k$ interacts with $M_{\mathrm{Lo}}$, and collect data $\tau_{\mathrm{Lo}}^k$;    $D_{\mathrm{Lo}}^{k+1} = D_{\mathrm{Lo}}^k \cup \{\tau_{\mathrm{Lo}}^k\}$
4     $\pi_{\mathrm{Hi}}^k \leftarrow \mathrm{Alg}^{\mathrm{Hi}}(D_{\mathrm{Hi}}^k)$;     $\pi_{\mathrm{Hi}}^k$ interacts with $M_{\mathrm{Hi}}$, and collect data $\tau_{\mathrm{Hi}}^k$;    $D_{\mathrm{Hi}}^{k+1} = D_{\mathrm{Hi}}^k \cup \{\tau_{\mathrm{Hi}}^k\}$.
5 **end**

---

---
**Algorithm 5:** The Tiered RL Framework with Multiple Low-Tier Tasks

---
1 Initialize $D_{\mathrm{Lo}}^1 \leftarrow \{\}$; $D_{\mathrm{Hi},w}^1 \leftarrow \{\}$, $\forall w \in [W]$.
2 **for** $k = 1, 2, ..., K$ **do**
3     **for** $w \in [W]$ **do**
4        $\pi_{\mathrm{Lo},w}^k \leftarrow \mathrm{Alg}^{\mathrm{Lo}}(D_{\mathrm{Hi},w}^k)$;    $\pi_{\mathrm{Lo},w}^k$ interacts with $M_{\mathrm{Lo},w}$, and collect data $\tau_{\mathrm{Lo},w}^k$.
5        $D_{\mathrm{Lo},w}^{k+1} = D_{\mathrm{Hi}}^k \cup \{\tau_{\mathrm{Lo},w}^k\}$.
6     **end**
7     $\pi_{\mathrm{Hi}}^k \leftarrow \mathrm{Alg}^{\mathrm{Hi}}(D_{\mathrm{Hi}}^k)$;    $\pi_{\mathrm{Hi}}^k$ interacts with $M_{\mathrm{Hi}}$, and collect data $\tau_{\mathrm{Hi}}^k$;    $D_{\mathrm{Hi}}^{k+1} = D_{\mathrm{Hi}}^k \cup \{\tau_{\mathrm{Hi}}^k\}$.
8 **end**

---

## A.2  Other Related Works

**Online and Offline RL**  In normal online RL/MAB setting, the learner targets at actively explore the environment while balancing the trade-off between the exploration and exploitation [17, 1, 23, 7, 2, 15] Differently, motivated by many real world scenarios where historical data are available, offline RL considers how to do pure exploitation given the pre-collected dataset without additional exploration and new information collection, and theoretical works mainly focus on methods for sufficiently exploitation [35, 28, 32, 16, 20, 14, 4].

Recently, there is also a line of work studying the settings lying between pure online and offline RL, such as hybrid setting where offline data is available for online exploration [33, 30], and efficient batched exploration with limited policy deployments [12]. Tiered RL framework can be regarded as another approach to bridging the online and offline setting, where we do online learning in the high-tier task with a gradually updated dataset from low-tier task for reference.

**Detailed Comparison with Previous Transfer RL Paper about Assumptions**  The most recent works in transfer RL are [9] and [10]. In general we are not comparable because of our different settings, but we can observe some similarity of our assumptions and the way to capture the transferable states.

[9] considered the case where a predicted Q-function $\{\widetilde{Q}_h(\cdot, \cdot)\}_{h=1}^H$ is provided for each state action pair, which can be regarded as $\{Q_{\mathrm{Lo},h}^*(\cdot, \cdot)\}_{h=1}^H$ in our setting. The key assumption in their paper is the "approximate distillation" condition (Def. 3.1 in [9]), which assumed that for each $s_h$, there exists $a_h \in \mathcal{A}_h$ such that $\Delta(s_h, a_h) + \max\{0, Q_h^*(s_h, a_h) - \widetilde{Q}_h(s_h, a_h)\} \le \varepsilon$. However, according to Eq. (2) in their Thm. 3.1, there is an $\varepsilon' TH = 4\varepsilon(H+1)TH$ term in the regret of their algorithm (where $T$ is the episode number). Therefore, in order to achieve regret sub-linear to $T$, they need $\varepsilon = T^{-\alpha}$ for some $\alpha > 0$. As $T \to +\infty$, we have $\varepsilon \to 0$, then their "approximate distillation" condition will reduce to $V_{\mathrm{Lo},h}^*(s_h) \ge V_{\mathrm{Hi},h}^*(s_h)$, which is a stronger version of our OVD condition in Assump. B.

As for [10], the authors assumed that there is a value function and parameter $\beta$ such that $\beta \widetilde{V}_h(s_h)$ (i.e. $V_{\mathrm{Lo}}^*(s_h)$ in our setting) forms an overestimation for $V_h^*(s_h)$ in target task, which is also similar to our Assump. B. Besides, although they didn't make it explicitly, to achieve provable benefits, they also require such a overestimation $\beta \widetilde{V}_h(s_h)$ should not deviate too far away from the true value $V_h^*(s_h)$. To see this, in Sec. 5.1 of [10], they use $V_h^*(s) \ge \Delta + \widetilde{Q}_h^u(s_h, a_h)$ to characterize state-action pairs

with regret reduction, where $\widetilde{Q}_h^u(s_h, a_h) := \mathbb{E}_{s'}[r(s_h, a_h) + \beta \widetilde{V}_{h+1}(s')]$, and $\beta \widetilde{V}_{h+1}$ has to stay close to $V_h^*(s)$ for such condition to be realizable.

Finally, both [9, 10] assumed the condition holds for each state action pair, while our Assump. B can only require the overestimation on states reachable by optimal policy.

### A.3 Examples for Assump. B

**Example A.1** (Identical Model [13]). For arbitrary $h \in [H], s_h \in \mathcal{S}_h, a_h \in \mathcal{A}_h$, $r_{\text{Lo},h}(s_h, a_h) = r_{\text{Hi},h}(s_h, a_h)$, $\mathbb{P}_{\text{Hi},h}(\cdot|s_h, a_h) = \mathbb{P}_{\text{Lo},h}(\cdot|s_h, a_h)$.

**Example A.2** (Small Model Error). For arbitrary $h \in [H], s_h \in \mathcal{S}_h, a_h \in \mathcal{A}_h$, $|r_{\text{Lo},h}(s_h, a_h) - r_{\text{Hi},h}(s_h, a_h)| \le \frac{\Delta_{\min}}{4H(H+1)}$, $\|\mathbb{P}_{\text{Hi},h}(\cdot|s_h, a_h) - \mathbb{P}_{\text{Lo},h}(\cdot|s_h, a_h)\|_1 \le \frac{\Delta_{\min}}{4H^2(H+1)}$.

**Example A.3** (Known Model Difference). Suppose there exists known quantities $\xi_r$ and $\xi_\mathbb{P}$ such that, for arbitrary $h \in [H], s_h \in \mathcal{S}_h, a_h \in \mathcal{A}_h$:

$$|r_{\text{Lo},h}(s_h, a_h) - r_{\text{Hi},h}(s_h, a_h)| \le \xi_r, \quad \|\mathbb{P}_{\text{Hi},h}(\cdot|s_h, a_h) - \mathbb{P}_{\text{Lo},h}(\cdot|s_h, a_h)\|_1 \le \xi_\mathbb{P}.$$

Then, one can revise the reward function of $M_{\text{Lo}}$ to $r'_{\text{Lo}}$ defined by $r'_{\text{Lo},h}(s_h, a_h) = r_{\text{Lo},h}(s_h, a_h) + \xi_r + (H - h)\xi_\mathbb{P}$, and the new MDP $M_{\text{Lo}'} = \{\mathcal{S}, \mathcal{A}, \mathbb{P}_{\text{Lo}}, r'_{\text{Lo}}, H\}$ has optimal value dominance on $M_{\text{Hi}}$.

**Proofs for Examples Above**   Ex. A.1 is obvious, we just prove the rest two. First of all, for arbitrary $h \in [H]$ and $s_h \in \mathcal{S}_h$, we should have:

$$V_{\text{Hi},h}^*(s_h) - V_{\text{Lo},h}^*(s_h) \le Q_{\text{Hi},h}^*(s_h, \pi_{\text{Hi}}^*) - Q_{\text{Lo},h}^*(s_h, \pi_{\text{Hi}}^*)$$
$$=r_{\text{Hi},h}(s_h, \pi_{\text{Hi}}^*) - r_{\text{Lo},h}(s_h, \pi_{\text{Hi}}^*) + (\mathbb{P}_{\text{Hi},h} - \mathbb{P}_{\text{Lo},h})V_{\text{Hi},h+1}^*(s_h, \pi_{\text{Hi}}^*) + \mathbb{P}_{\text{Lo},h}(V_{\text{Hi},h+1}^* - V_{\text{Lo},h+1}^*)(s_h, \pi_{\text{Hi}}^*)$$
$$\le r_{\text{Hi},h}(s_h, \pi_{\text{Hi}}^*) - r_{\text{Lo},h}(s_h, \pi_{\text{Hi}}^*) + (\mathbb{P}_{\text{Hi},h} - \mathbb{P}_{\text{Lo},h})V_{\text{Hi},h+1}^*(s_h, \pi_{\text{Hi}}^*) + \mathbb{P}_{\text{Lo},h}(V_{\text{Hi},h+1}^* - Q_{\text{Lo},h+1}^*(\cdot, \pi_{\text{Hi}}^*))(s_h, \pi_{\text{Hi}}^*)$$
$$\le \ldots$$
$$\le \mathbb{E}_{M_{\text{Lo}}, \pi_{\text{Hi}}^*}\Big[\sum_{h'=h}^H r_{\text{Hi},h}(s_{h'}, a_{h'}) - r_{\text{Lo},h'}(s_{h'}, a_{h'}) + (\mathbb{P}_{\text{Hi},h'} - \mathbb{P}_{\text{Lo},h'})V_{\text{Hi},h'+1}^*(s_{h'}, a_{h'})|s_h\Big]$$
$$\le \mathbb{E}_{M_{\text{Lo}}, \pi_{\text{Hi}}^*}\Big[\sum_{h'=h}^H |r_{\text{Hi},h}(s_{h'}, a_{h'}) - r_{\text{Lo},h'}(s_{h'}, a_{h'})| + (H - h) \cdot \|\mathbb{P}_{\text{Hi},h'}(\cdot|s_{h'}, a_{h'}) - \mathbb{P}_{\text{Lo},h'}(\cdot|s_{h'}, a_{h'})\|_1|s_h\Big],$$

Therefore, in Example A.2, we should expect:

$$V_{\text{Hi},h}^*(s_h) - V_{\text{Lo},h}^*(s_h) \le (H - h) \cdot \frac{\Delta_{\min}}{4H(H+1)} + (H - h) \cdot (H - h) \cdot \frac{\Delta_{\min}}{4H^2(H+1)} \le \frac{\Delta_{\min}}{2(H+1)}.$$

Besides, for Example A.3, we have:

$$V_{\text{Hi},h}^*(s_h) - V_{\text{Lo}',h}^*(s_h) \le \mathbb{E}_{M_{\text{Lo}}, \pi_{\text{Hi}}^*}\Big[\sum_{h'=h}^H r_{\text{Hi},h}(s_{h'}, a_{h'}) - r_{\text{Lo},h'}(s_{h'}, a_{h'}) + \xi_r$$
$$+ (H - h)\xi_\mathbb{P} + (H - h) \cdot \|\mathbb{P}_{\text{Hi},h'}(\cdot|s_{h'}, a_{h'}) - \mathbb{P}_{\text{Lo},h'}(\cdot|s_{h'}, a_{h'})\|_1|s_h\Big]$$
$$\le 0$$

Therefore, both Example A.2 and A.3 satisfy Assump. B.

### A.4 Detailed Discussion on Open Problems

We believe there are many interesting directions to follow in the future and highlight in three aspects:

First of all, we conjecture our unique optimal policy assumption can be relaxed and the $O(\frac{1}{H})$ factor in Def. 2.2 and Def. F.1 can be removed by advanced techniques. It's also important to study how to get rid of lower bound knowledge in $\Delta_{\min}$.

Secondly, in TRL-MST setting, for those $s_h$ such that there are multiple source tasks $M_{\text{Lo},w_1}, M_{\text{Lo},w_2}, ..., M_{\text{Lo},w_j} \in \mathcal{M}_{\text{Lo}}$ close to $M_{\text{Hi}}$, beyond the constant regret, one may consider to

integrate the information from those source tasks together to further accelerate the learning; Moreover, although we do not make it explicitly, it's possible to combine our techniques in Sec. 5 with existing MT-RL algorithms to develop algorithms with guarantees about the reduction on not only the total regret but also some specific tasks.

Finally, although we show in Sec. 3 that robust transfer objective requires OVD assumption, and the model difference tolerance is at most $O(\Delta_{\min})$, we conjecture that, there might exists milder assumptions about the structure of source and target tasks and the prior knowledge about it, which may eliminate out our hard instance. Additionally, in some cases, it may be reasonable to relax the objective by allowing some chance of negative transfer in part of target tasks. Then, we can do more aggressive transfer without too much concern on the algorithm's overall performance. These potential directions are left for future work.

## B   Proofs for Lower Bound

**Theorem 3.1.** *Under the violation of Assump. B, even regardless of the optimality of $Alg^{Lo}$, for each algorithm pair $(Alg^{Lo}, Alg^{Hi})$, it cannot simultaneously (1) achieve constant regret for the case when $M_{Lo} = M_{Hi}$ and (2) ensure sub-linear regret in all the other cases.*

*Proof.* Consider the two-armed bandit setting. Given arbitrary $\Delta, \mu \in (0, 1)$ satisfying $0 < \mu - \Delta < \mu < \mu + \Delta$, we can construct two two-armed Bernoulli bandit problem $M$ and $M'$ such that:

$$\mu_M(1) = \mu_{M'}(1) = \mu; \quad \mu_M(2) = \mu - \Delta; \quad \mu_{M'}(2) = \mu + \Delta.$$

We choose $M$ to be the low-tier task, i.e. $M_{Lo} = M$, and choose $M$ and $M'$ to be the high-tier task. Note that the minimal gap in $M$ and $M'$ is $\Delta$, and $\mu_M(1) = \mu_{M'}(2) - \Delta < \mu_{M'}(2) - \frac{\Delta}{2}$, which implies $M_{Lo}$ does not have optimal value dominance on $M_{Hi}$ when $M_{Lo} = M$ and $M_{Hi} = M'$.

Now, we consider the following learning process: the learner will get access to the low-tier task $M_{Lo} = M$, and the high-tier task $M_{Hi}$ will be uniformly randomly selected between $M$ and $M'$, while the learner does not know which it is. Without loss of generality, we consider deterministic algorithms $Alg^{Lo}$ and $Alg^{Hi}$ (since one can first generate the randomness before the learning process), i.e. for arbitrary step $k$, given the interaction history $\tau_k := (a_{Hi}^1, r_{Hi}^1, a_{Lo}^1, r_{Lo}^1, ..., a_{Hi}^{k-1}, r_{Hi}^{k-1}, a_{Lo}^{k-1}, r_{Lo}^{k-1})$, the policy $(\pi_{Lo}^k, \pi_{Hi}^k)$ produced by $Alg^{Lo}(\tau_k)$ and $Alg^{Hi}(\tau_k)$ is fixed, where $a_{Lo}^k, r_{Lo}^k$ (or $a_{Hi}^k, r_{Hi}^k$) denotes the arm pulled and the reward observed in task $M_{Lo}$ (or $M_{Hi}$) at iteration $k$.

In the following, we will use $Pr_{Alg^{Lo}, Alg^{Hi}}^{M_{Lo}, M_{Hi}}(\cdot)$ to denote the probability if the learner use algorithm pair $(Alg^{Lo}, Alg^{Hi})$ and solve task pair $(M_{Lo}, M_{Hi})$. Note that the pseudo-regret of $Alg^{Hi}$ when $M_{Hi} = M$ can be written as:

$$\text{Regret}_K(M_{Hi}; M_{Hi} = M) = \sum_{\tau_K : Pr_{Alg^{Lo}, Alg^{Hi}}^{M,M}(\tau_K) > 0} Pr_{Alg^{Lo}, Alg^{Hi}}^{M,M}(\tau_K) N_{Hi}(2; \tau_K) \Delta. \tag{2}$$

where $N_{Hi}(i; \tau_K)$ denotes the number of times the $i$-th arm is pulled in task $M_{Hi}$ in trajectory $\tau_K$.

Because both $M_{Lo}$ and $M_{Hi}$ are two-armed Bernoulli bandits, and each arm in those MDPs has non-zero probability mass on both value 0 and 1 and the algorithms are deterministic, for arbitrary $k \geq 1$ and $\tau_k$, $Pr_{Alg^{Lo}, Alg^{Hi}}^{M,M}(\tau_k) > 0$ if and only if $Pr_{Alg^{Lo}, Alg^{Hi}}^{M,M}(\tau_k) > 0$.

Now, we consider the following probability ratio, for arbitrary $\tau_k$ with $Pr_{Alg^{Lo}, Alg^{Hi}}^{M,M}(\tau_k) > 0$:

$$\frac{Pr_{Alg^{Lo}, Alg^{Hi}}^{M,M'}(\tau_k)}{Pr_{Alg^{Lo}, Alg^{Hi}}^{M,M}(\tau_k)} = \frac{Pr_{M'}(r_{Hi} = r_{Hi}^{k-1}|a_{Hi}^{k-1})}{Pr_M(r_{Hi} = r_{Hi}^{k-1}|a_{Hi}^{k-1})} \cdot \frac{Pr_{Alg^{Lo}, Alg^{Hi}}^{M,M'}(\tau_{k-1})}{Pr_{Alg^{Lo}, Alg^{Hi}}^{M,M}(\tau_{k-1})}$$

$$(Pr_M(\cdot) \text{ denotes the probability of event on model } M)$$

$$= \prod_{k'=1}^k \frac{Pr_{M'}(r_{Hi} = r_{Hi}^{k'}|a_{Hi}^{k'})}{Pr_M(r_{Hi} = r_{Hi}^{k'}|a_{Hi}^{k'})} \geq (\frac{1 - \mu - \Delta}{1 - \mu + \Delta})^{N_{Hi}(2; \tau_k)}. \tag{3}$$

where for the first equality, we use the fact that the algorithms are deterministic, and the randomness of $r_{Lo}^{k-1}$ only depends on $a_{Lo}^{k-1}$ so it cancels out; the last inequality is because that the ratio is 1 if

$a_{\text{Hi}}^{k'} = 1$ and the ratio can be lower bounded by $(1 - \mu - \Delta)/(1 - \mu + \Delta)$ otherwise, Therefore, combining with Eq. (2), we have:

$$\text{Regret}_K(M_{\text{Hi}}; M_{\text{Hi}} = M') = \sum_{\tau_K} \Pr_{\text{Alg}^{\text{Lo}}, \text{Alg}^{\text{Hi}}}^{M, M'}(\tau_K) N_{\text{Hi}}(1; \tau_K) \Delta$$

$$= \sum_{\tau_K : \Pr_{\text{Alg}^{\text{Lo}}, \text{Alg}^{\text{Hi}}}^{M, M}(\tau_K) > 0} \Pr_{\text{Alg}^{\text{Lo}}, \text{Alg}^{\text{Hi}}}^{M, M}(\tau_K) \frac{\Pr_{\text{Alg}^{\text{Lo}}, \text{Alg}^{\text{Hi}}}^{M, M'}(\tau_K)}{\Pr_{\text{Alg}^{\text{Lo}}, \text{Alg}^{\text{Hi}}}^{M, M}(\tau_K)} N_{\text{Hi}}(1; \tau_K) \Delta$$

$$\geq \sum_{\tau_K : \Pr_{\text{Alg}^{\text{Lo}}, \text{Alg}^{\text{Hi}}}^{M, M}(\tau_K) > 0} \Pr_{\text{Alg}^{\text{Lo}}, \text{Alg}^{\text{Hi}}}^{M, M}(\tau_K) \left(\frac{1 - \mu - \Delta}{1 - \mu + \Delta}\right)^{N_{\text{Hi}}(2; \tau_K)} N_{\text{Hi}}(1; \tau_K) \Delta$$

(4)

Suppose the algorithm pair $(\text{Alg}^{\text{Lo}}, \text{Alg}^{\text{Hi}})$ can achieve constant regret $C$ when $(M_{\text{Lo}}, M_{\text{Hi}}) = (M, M)$, i.e.

$$\mathbb{E}_{\text{Alg}^{\text{Lo}}, \text{Alg}^{\text{Hi}}, M, M}[N_{\text{Hi}}(2; \tau_K) \Delta] \leq C, \quad \forall K \geq 1.$$

then, according to Markov inequality, for arbitrary constant $\delta \in (0, 1)$ we have:

$$\Pr_{\text{Alg}^{\text{Lo}}, \text{Alg}^{\text{Hi}}}^{M, M}\left(N_{\text{Hi}}(2; \tau_K) \leq \frac{C}{\Delta\delta}\right) \geq 1 - \delta, \quad \forall K \geq 1.$$

(5)

which is equivalent to (note that $\tau_K$ is the random variable)

$$\sum_{\tau_K : N_{\text{Hi}}(2; \tau_K) \leq \frac{C}{\Delta\delta}} \Pr_{\text{Alg}^{\text{Lo}}, \text{Alg}^{\text{Hi}}}^{M, M}(\tau_K) \geq 1 - \delta.$$

Combining with Eq. (4), by choosing an arbitrary constant $\delta \in (0, 1)$, for arbitrary $K \geq 1$, we have:

$$\text{Regret}_K(M_{\text{Hi}}; M_{\text{Hi}} = M') \geq \sum_{\tau_K : N_{\text{Hi}}(2; \tau_K) \leq \frac{C}{\Delta\delta}} \Pr_{\text{Alg}^{\text{Lo}}, \text{Alg}^{\text{Hi}}}^{M, M}(\tau_K) \left(\frac{1 - \mu - \Delta}{1 - \mu + \Delta}\right)^{N_{\text{Hi}}(2; \tau_K)} N_{\text{Hi}}(1; \tau_K) \Delta$$

$$\geq \sum_{\tau_K : N_{\text{Hi}}(2; \tau_K) \leq \frac{C}{\Delta\delta}} \Pr_{\text{Alg}^{\text{Lo}}, \text{Alg}^{\text{Hi}}}^{M, M}(\tau_K) \left(\frac{1 - \mu - \Delta}{1 - \mu + \Delta}\right)^{\frac{C}{\Delta\delta}} \left(K - \frac{C}{\Delta\delta}\right) \Delta$$

$$\geq (1 - \delta) \cdot \left(\frac{1 - \mu - \Delta}{1 - \mu + \Delta}\right)^{\frac{C}{\Delta\delta}} \left(K - \frac{C}{\Delta\delta}\right) \Delta$$

$$= O(K).$$

which finishes the proof. □

**Theorem 3.2.** *[Transferable States are Restricted by $\Delta_{\min}$] Under Assump. B, regardless of the optimality of $Alg^{Lo}$, given arbitrary $\Delta_{\min}$ and arbitrary $\Delta \in [\frac{\Delta_{\min}}{2}, \Delta_{\min}]$, for each algorithm pair $(Alg^{Lo}, Alg^{Hi})$, it cannot simultaneously (1) achieve constant regret for the case when $M_{Lo}$ and $M_{Hi}$ with minimal gap $\Delta_{\min}$ are $\Delta$-close, and (2) ensure sub-linear regret in all other cases.*

*Proof.* We can construct three two-armed Bernoullis bandit problem $M, M'$ and $M''$ such that:

$$\mu_M(1) = \mu, \quad \mu_M(2) = \mu - \Delta;$$
$$\mu_{M'}(1) = \mu - \Delta', \quad \mu_{M'}(2) = \mu - \Delta - \Delta';$$
$$\mu_{M''}(1) = \mu - \Delta', \quad \mu_{M''}(2) = \mu + \Delta - \Delta';$$

where $\Delta$ and $\mu$ are chosen to satisfy $0 < \mu - 2\Delta < \mu < \mu + \Delta$, and $\Delta' \in [\frac{\Delta}{2}, \Delta]$. Note that by construction, $\Delta$ is $\Delta_{\min}$. Now, consider the following learning process, the learner will be provided $M$ as the low-tier task $M_{\text{Lo}}$, and the high-tier task $M_{\text{Hi}}$ will be uniformly sampling from $\{M', M''\}$. Easy to check that, $M_{\text{Lo}} = M$, has optimal value dominance on $M_{\text{Hi}}$ when $M_{\text{Hi}} = M'$ or $M_{\text{Hi}} = M''$. Next, we want to show that, for arbitrary algorithm pair $(\text{Alg}^{\text{Lo}}, \text{Alg}^{\text{Hi}})$, if the learner can achieve constant regret when $M_{\text{Hi}} = M'$, it must achieve linear regret when $M_{\text{Hi}} = M''$.

The remaining proof is similar to the proof for Thm. 3.1. First of all, we have:

$$\text{Regret}_K(M_{\text{Hi}}; M_{\text{Hi}} = M') = \sum_{\tau_K : \Pr^{M,M'}_{\text{Alg}^{\text{Lo}},\text{Alg}^{\text{Hi}}}(\tau_K) > 0} \Pr^{M,M'}_{\text{Alg}^{\text{Lo}},\text{Alg}^{\text{Hi}}}(\tau_K) N_{\text{Hi}}(2; \tau_K)\Delta. \qquad (6)$$

As an analogue of Eq. (3), we have:

$$\frac{\Pr^{M,M'}_{\text{Alg}^{\text{Lo}},\text{Alg}^{\text{Hi}}}(\tau_k)}{\Pr^{M,M''}_{\text{Alg}^{\text{Lo}},\text{Alg}^{\text{Hi}}}(\tau_k)} \geq \left(\frac{\mu - \Delta - \Delta'}{\mu + \Delta - \Delta'}\right)^{N_{\text{Hi}}(2;\tau_k)}.$$

Combining with Eq. (6), we have:

$$\text{Regret}_K(M_{\text{Hi}}; M_{\text{Hi}} = M'') \geq \sum_{\tau_K : \Pr^{M,M'}_{\text{Alg}^{\text{Lo}},\text{Alg}^{\text{Hi}}}(\tau_K) > 0} \Pr^{M,M'}_{\text{Alg}^{\text{Lo}},\text{Alg}^{\text{Hi}}}(\tau_K)\left(\frac{\mu - \Delta - \Delta'}{\mu + \Delta - \Delta'}\right)^{N_{\text{Hi}}(2;\tau_K)} N_{\text{Hi}}(1; \tau_K)\Delta$$

$$(7)$$

Suppose the algorithm pair $(\text{Alg}^{\text{Lo}}, \text{Alg}^{\text{Hi}})$ can achieve constant regret $C$ when $(M_{\text{Lo}}, M_{\text{Hi}}) = (M, M')$, we must have:

$$\sum_{\tau_K : N_{\text{Hi}}(2;\tau_K) \leq \frac{C}{\Delta\delta}} \Pr^{M,M'}_{\text{Alg}^{\text{Lo}},\text{Alg}^{\text{Hi}}}(\tau_K) \geq 1 - \delta.$$

By choosing an arbitrary fixed constant $\delta \in (0, 1)$, for arbitrary $K$, we have:

$$\text{Regret}_K(M_{\text{Hi}}; M_{\text{Hi}} = M'') \geq \sum_{\tau_K : N_{\text{Hi}}(2;\tau_K) \leq \frac{C}{\Delta\delta}} \Pr^{M,M'}_{\text{Alg}^{\text{Lo}},\text{Alg}^{\text{Hi}}}(\tau_K)\left(\frac{1 - \mu - \Delta}{1 - \mu + \Delta}\right)^{N_{\text{Hi}}(2;\tau_K)} N_{\text{Hi}}(1; \tau_K)\Delta$$

$$\geq \sum_{\tau_K : N_{\text{Hi}}(2;\tau_K) \leq \frac{C}{\Delta\delta}} \Pr^{M,M'}_{\text{Alg}^{\text{Lo}},\text{Alg}^{\text{Hi}}}(\tau_K)\left(\frac{1 - \mu - \Delta}{1 - \mu + \Delta}\right)^{\frac{C}{\Delta\delta}} (K - \frac{C}{\Delta\delta})\Delta$$

$$\geq (1 - \delta) \cdot \left(\frac{\mu - \Delta - \Delta'}{\mu + \Delta - \Delta'}\right)^{\frac{C}{\Delta\delta}} (K - \frac{C}{\Delta\delta})\Delta$$

$$= O(K).$$

which finishes the proof. $\qquad\qquad\square$

## C   Proofs for Tiered MAB with Single Source/Low-Tier Task

**Lemma C.1** (Concentration Inequality). *In Alg. 1, at each iteration $k$, we have:*

$$\Pr(|\mu_{Lo}(i) - \widehat{\mu}^k_{Lo}(i)| \geq \sqrt{\frac{2\alpha \log f(k)}{N^k_{Lo}(i)}}) \leq \frac{2}{f(k)^\alpha} \leq \frac{1}{8Ak^{2\alpha}}, \quad \forall i \in [A]$$

$$\Pr(|\mu_{Hi}(i) - \widehat{\mu}^k_{Hi}(i)| \geq \sqrt{\frac{2\alpha \log f(k)}{N^k_{Hi}(i)}}) \leq \frac{2}{f(k)^\alpha} \leq \frac{1}{8Ak^{2\alpha}}, \quad \forall i \in [A]$$

As a direct result, we have the following lemma:

**Lemma C.2.** *[Valid Under Estimation] For arbitrary $i \in [A]$, if $\mu_{Lo}(i) \leq \mu_{Hi}(i) + \varepsilon$, for arbitrary iteration $k$ in Alg. 1, we have:*

$$\Pr(\underline{\mu}_{Lo}(i) \leq \overline{\mu}_{Hi}(i) + \varepsilon) \geq 1 - \frac{4}{f(k)^\alpha} \geq 1 - \frac{1}{4Ak^{2\alpha}}$$

*Proof.* According to Lem. C.1, w.p. at least $1 - \frac{4}{f(k)^\alpha}$ we have:

$$\Pr(\underline{\mu}_{\text{Lo}}(i) \leq \overline{\mu}_{\text{Hi}}(i) + \varepsilon) \geq \Pr(\{\underline{\mu}_{\text{Lo}}(i) \leq \mu_{\text{Lo}}(i)\} \cap \{\mu_{\text{Hi}}(i) + \varepsilon \leq \overline{\mu}_{\text{Hi}}(i) + \varepsilon\}) \geq 1 - \frac{4}{f(k)^\alpha} \geq 1 - \frac{1}{4Ak^{2\alpha}}.$$

$$\square$$

Next, we recall two useful lemma: Lemma 4.2 and Lemma D.1 from [13].

**Lemma C.3** (Property of UCB; Lem 4.2 in [13])**.** *With the choice that $f(k) = 1 + 16A^2(k+1)^2$, there exists a constant c, for arbitrary i with $\Delta_{Lo}(i) > 0$ and arbitrary $\nu \in [1, 4A]$, in UCB algorithm, we have:*

$$\Pr(N_{Lo}^k(i) \geq \frac{k}{\nu}) \leq \frac{2}{k^{2\alpha-1}}, \quad \forall k \geq \nu + c \cdot \frac{\alpha\nu}{\Delta_{Lo}^2(i)} \log(1 + \frac{\alpha A}{\Delta_{\min}}).$$

**Lemma C.4** (Lemma D.1 in [13])**.** *Given an arm i, we separate all the arms into two parts depending on whether its gap is larger than $\Delta_{Lo}(i)$ and define $G_i^{lower} := \{\iota | \Delta_{Lo}(\iota) > \Delta_{Lo}(i)/2\}$ and $G_i^{upper} := \{\iota | \Delta_{Lo}(\iota) \leq \Delta_{Lo}(i)/2\}$. With the choice that $f(k) = 1 + 16A^2(k+1)^2$, there is a constant c, such that for arbitrary i with $\Delta_{Lo}(i) > 0$, for $\underline{\pi}_{Lo}^k$ in Alg 1, there exists a constant c, such that:*

$$\Pr(i = \underline{\pi}_{Lo}^k) \leq 2/k^{2\alpha} + 2A/k^{2\alpha-1}, \quad \forall k \geq k_i := 8\alpha c\Big( \sum_{\iota \in G_i^{lower}} \frac{1}{\Delta_{Lo}^2(\iota)} + \frac{4|G_i^{upper}|}{\Delta_{Lo}^2(i)} \Big) \log(1 + \frac{\alpha A}{\Delta_{\min}}) \tag{8}$$

**Lemma C.5.** *We denote $k_i' := 3A + c \cdot \frac{3\alpha A}{\Delta_{Lo}^2(i)} \log(1 + \frac{\alpha A}{\Delta_{\min}})$ and $\widetilde{k}_i := 3 + c \cdot \frac{3\alpha}{\Delta_{Lo}^2(i)} \log(1 + \frac{\alpha A}{\Delta_{\min}})$, where c is specified in Lem. C.3, and denote $k_{\max} := \max_{i \neq i^*} \max\{k_i, k_i'\}$, where $k_i$ is defined in Lemma C.4, we have:*

$$\Pr(i^* = \underline{\pi}_{Lo}^k) = 1 - \sum_{i \neq i^*} \Pr(i = \underline{\pi}_{Lo}^k) \geq 1 - \frac{2A}{k^{2\alpha}} - \frac{2A^2}{k^{2\alpha-1}}, \quad \forall k \geq k_{\max}. \tag{9}$$

$$\Pr(N_{Lo}^k(i^*) > \frac{k}{2}) \geq \Pr(N_{Lo}^k(i^*) \geq \frac{2k}{3}) \geq 1 - \sum_{i \neq i^*} \Pr(N_{Lo}^k(i) \leq \frac{k}{3A}) \geq 1 - \frac{2A}{k^{2\alpha-1}}, \quad \forall k \geq k_{\max}. \tag{10}$$

$$\Pr(N_{Lo}^k(i) > \frac{k}{2}) \leq \Pr(N_{Lo}^k(i) \geq \frac{k}{3}) \leq \frac{2}{k^{2\alpha-1}}, \quad \forall k \geq \widetilde{k}_i. \tag{11}$$

*Proof.* By applying Lem. C.4 and Lem. C.3 we can obtain the results. $\square$

**Lemma C.6.** *For arbitrary $K \geq A + 1$ and arbitrary $k_0 \leq K$, and $i \neq i_{Hi}^*$, we have:*

$$N_{Hi}^K(i) \leq k_0 + \sum_{k=k_0+1}^{K} \mathbb{I}[\{\underline{\mu}_{Lo}^k(\underline{\pi}_{Lo}^k) \leq \overline{\mu}_{Hi}^k(\underline{\pi}_{Lo}^k) + \varepsilon\} \cap \{N_{Lo}^k(\underline{\pi}_{Lo}^k) > k/2\} \cap \{i = \underline{\pi}_{Lo}^k\} \cap \{\pi_{Hi}^k = i\}]$$

$$+ \sum_{k=k_0+1}^{K} \mathbb{I}[0 \geq \widehat{\mu}_{Hi}^k(i_{Hi}^*) + \sqrt{\frac{2\alpha \log f(k)}{N_{Hi}^k(i_{Hi}^*)}} - \mu_{Hi}(i_{Hi}^*)]$$

$$+ \sum_{k=k_0+1}^{K} \mathbb{I}[\widetilde{\mu}_{Hi}^k(i) + \sqrt{\frac{2\alpha \log f(K)}{k}} - \mu_{Hi}(i) - \Delta_{Hi}(i) \geq 0] \tag{12}$$

*Proof.*

$$N_{Hi}^K(i) = \sum_{k=1}^{K} \mathbb{I}[\pi_{Hi}^k = i] \leq k_0 + \sum_{k_0+1}^{K} \mathbb{I}[\pi_{Hi}^k = i]$$

$$\leq k_0 + \sum_{k=k_0+1}^{K} \underbrace{\mathbb{I}[\{\underline{\mu}_{Lo}^k(\underline{\pi}_{Lo}^k) \leq \overline{\mu}_{Hi}^k(\underline{\pi}_{Lo}^k) + \varepsilon\} \cap \{N_{Lo}^k(\underline{\pi}_{Lo}^k) > k/2\} \cap \{i = \underline{\pi}_{Lo}^k\}}_{e_1} \cap \{\pi_{Hi}^k = i\}]$$

$$+ \sum_{k=k_0+1}^{K} \underbrace{\mathbb{I}[\{\widehat{\mu}_{Hi}^k(i) + \sqrt{\frac{2\alpha \log f(k)}{N_{Hi}^k(i)}} - \mu_{Hi}(i) - \Delta_{Hi}(i) \geq \widehat{\mu}_{Hi}^k(i_{Hi}^*) + \sqrt{\frac{2\alpha \log f(k)}{N_{Hi}^k(i_{Hi}^*)}} - \mu_{Hi}(i_{Hi}^*)\}}_{e_2} \cap \{\pi_{Hi}^k = i\}].$$

(If $\pi_{Hi}^k = i$ happens, one of $e_1$ and $e_2$ must hold)

For the second term, we have:

$$\sum_{k=k_0+1}^{K} \mathbb{I}[\{\widehat{\mu}_{\mathrm{Hi}}^k(i) + \sqrt{\frac{2\alpha \log f(k)}{N_{\mathrm{Hi}}^k(i)}} - \mu_{\mathrm{Hi}}(i) - \Delta_{\mathrm{Hi}}(i) \geq \widehat{\mu}_{\mathrm{Hi}}^k(i_{\mathrm{Hi}}^*) + \sqrt{\frac{2\alpha \log f(k)}{N_{\mathrm{Hi}}^k(i_{\mathrm{Hi}}^*)}} - \mu_{\mathrm{Hi}}(i_{\mathrm{Hi}}^*)\} \cap \{\pi_{\mathrm{Hi}}^k = i\}]$$

$$\leq \sum_{k=k_0+1}^{K} \mathbb{I}[0 \geq \widehat{\mu}_{\mathrm{Hi}}^k(i_{\mathrm{Hi}}^*) + \sqrt{\frac{2\alpha \log f(k)}{N_{\mathrm{Hi}}^k(i_{\mathrm{Hi}}^*)}} - \mu_{\mathrm{Hi}}(i_{\mathrm{Hi}}^*)]$$

$$+ \sum_{k=k_0+1}^{K} \mathbb{I}[\{\widehat{\mu}_{\mathrm{Hi}}^k(i) + \sqrt{\frac{2\alpha \log f(k)}{N_{\mathrm{Hi}}^k(i)}} - \mu_{\mathrm{Hi}}(i) - \Delta_{\mathrm{Hi}}(i) \geq 0\} \cap \{\pi_{\mathrm{Hi}}^k = i\}]$$

$$(\mathbb{I}[a \geq b] \leq \mathbb{I}[a \geq c] + \mathbb{I}[c \geq b]; \mathbb{I}[a \cap b] \leq \mathbb{I}[a])$$

$$\leq \sum_{k=k_0+1}^{K} \mathbb{I}[0 \geq \widehat{\mu}_{\mathrm{Hi}}^k(i_{\mathrm{Hi}}^*) + \sqrt{\frac{2\alpha \log f(k)}{N_{\mathrm{Hi}}^k(i_{\mathrm{Hi}}^*)}} - \mu_{\mathrm{Hi}}(i_{\mathrm{Hi}}^*)] + \sum_{k=k_0+1}^{K} \mathbb{I}[\widetilde{\mu}_{\mathrm{Hi}}^k(i) + \sqrt{\frac{2\alpha \log f(K)}{k}} - \mu_{\mathrm{Hi}}(i) - \Delta_{\mathrm{Hi}}(i) \geq 0]$$

where in the last step, $\widetilde{\mu}_{\mathrm{Hi}}^k(i)$ is defined to be the average of $k$ random samples from reward distribution of arm $i$ in $M_{\mathrm{Hi}}$, and we replace $N_{\mathrm{Hi}}^k(i)$ in the denominator with increasing $k$ since the indicator function equals 1 only when $\{\pi_{\mathrm{Hi}}^k = i\}$, which implies that $N_{\mathrm{Hi}}^k(i)$ should increase by 1. $\qquad \square$

**Theorem 4.1.** *[Tiered MAB with Single Source Tasks] Under Assump. A, B and C, by running Alg. 1 with $\varepsilon = \frac{\widetilde{\Delta}_{\min}}{4}$ and $\alpha > 2$, we always have $Regret_K(M_{Hi}) = O(\sum_{\Delta_{Hi}(i)>0} \frac{1}{\Delta_{Hi}(i)} \log K)$. Moreover, if $M_{Hi}$ and $M_{Lo}$ are $\frac{\widetilde{\Delta}_{\min}}{4}$-close, we have: $Regret_K(M_{Hi}) = O(\sum_{\Delta_{Hi}(i)>0} \frac{1}{\Delta_{Hi}(i)} \log \frac{A}{\Delta_{\min}})$.*

*Proof.* We first study the case when $M_{\mathrm{Hi}}$ and $M_{\mathrm{Lo}}$ satisfy Def. 2.1.

**Case 1: $M_{\mathbf{Hi}}$ and $M_{\mathbf{Lo}}$ are $\varepsilon$-close** In this case, since $i_{\mathrm{Lo}}^* = i_{\mathrm{Hi}}^*$, we use $i^*$ to denote the common optimal arm. As a result of Lem. C.2, we have:

$$\Pr(\underline{\mu}_{\mathrm{Lo}}^k(i^*) \leq \overline{\mu}_{\mathrm{Hi}}^k(i^*) + \varepsilon) \geq 1 - \frac{1}{4Ak^{2\alpha}}.$$

We consider the same $k_{\max}$ defined in Lem. C.5, as a result of Lem. C.5, for arbitrary $K \geq k_{\max} + 1$,

$$\sum_{k=k_{\max}+1}^{K} \Pr(\pi_{\mathrm{Hi}}^k \neq i_{\mathrm{Hi}}^*) \leq \sum_{k=k_{\max}+1}^{K} \Pr(\underline{\mu}_{\mathrm{Lo}}^k(i^*) > \overline{\mu}_{\mathrm{Hi}}^k(i^*) + \varepsilon) + \Pr(N_{\mathrm{Lo}}^k(i^*) \leq \frac{k}{2}) + \Pr(i^* \neq \pi_{\mathrm{Lo}}^k)$$

$$\leq \sum_{k=k_{\max}+1}^{K} \frac{1}{4Ak^{2\alpha}} + \frac{2A}{k^{2\alpha}} + \frac{2A^2}{k^{2\alpha-1}} + \frac{2A}{k^{2\alpha-1}}$$

$$\leq \sum_{k=k_{\max}+1}^{\infty} \frac{8A^2}{k^{2\alpha-1}} \leq \frac{8A^2}{(2\alpha-2)k_{\max}^{2\alpha-2}}.$$

Therefore, all we need to do is to upper bound the regret up to step $k_{\max}$. In the following, we separately upper bound $\mathbb{E}[N_{\mathrm{Hi}}^{k_{\max}}(i)]$ for $i \neq i^*$ for two cases depending on the comparison between $\Delta_{\mathrm{Hi}}(i)$ and $\Delta_{\mathrm{Lo}}(i)$.

**Case 1-(a) $0 < \Delta_{\mathbf{Hi}}(i) \leq 4\Delta_{\mathbf{Lo}}(i)$** Recall $\widetilde{k}_i$ in Lem. C.5. In this case, since $\Delta_{\mathrm{Lo}}^{-1}(i) \leq \Delta_{\mathrm{Hi}}^{-1}(i)$, we have $\widetilde{k}_i = O(\frac{1}{\Delta_{\mathrm{Hi}}^2(i)} \log \frac{A}{\Delta_{\min}})$, and by taking expectation over Eq. (12):

$$\mathbb{E}[N_{\mathrm{Hi}}^K(i)] \leq \widetilde{k}_i + \sum_{k=\widetilde{k}_i+1}^{K} \Pr(\{N_{\mathrm{Lo}}^k(\pi_{\mathrm{Lo}}^k) \geq \frac{k}{2}\}) + \sum_{k=1}^{K} \Pr(0 \geq \widehat{\mu}_{\mathrm{Hi}}^k(i_{\mathrm{Hi}}^*) + \sqrt{\frac{2\alpha \log f(k)}{N_{\mathrm{Hi}}^k(i_{\mathrm{Hi}}^*)}} - \mu_{\mathrm{Hi}}(i_{\mathrm{Hi}}^*))$$

$$+ \mathbb{E}[\sum_{k=1}^{K} \mathbb{I}[\widetilde{\mu}_{\mathrm{Hi}}^k(i) + \sqrt{\frac{2\alpha \log f(k)}{k}} - \mu_{\mathrm{Hi}}(i) - \Delta_{\mathrm{Hi}}(i) \geq 0]]$$

$$\leq \widetilde{k}_i + \sum_{k=\widetilde{k}_i+1}^{K} \frac{2}{k^{2\alpha-1}} + \sum_{k=1}^{K} \frac{1}{8Ak^{2\alpha}} + 1 + \frac{2}{\Delta_{\mathrm{Hi}}^2(i)}(\alpha \log f(K) + \sqrt{\pi\alpha \log f(K)} + 1)$$

$$\text{(Lem. 8.2 in [17])}$$

$$= O(\frac{1}{\Delta_{\mathrm{Hi}}^2(i)} \log \frac{AK}{\Delta_{\min}})$$

**Case 1-(b)** $\Delta_{\mathbf{Hi}}(i) > 4\Delta_{\mathbf{Lo}}(i) > 0$  We introduce $\bar{k}_i := \frac{c_{\mathrm{Hi},i}\alpha}{\Delta_{\mathrm{Hi}}^2(i)} \log \frac{A}{\Delta_{\min}}$, where $c_{\mathrm{Hi},i}$ is the minimal constant, such that when $k \geq \frac{c_{\mathrm{Hi},i}\alpha}{\Delta_{\mathrm{Hi}}^2(i)} \log \frac{\alpha A}{\Delta_{\min}}$, we always have $k \geq \frac{256\alpha \log f(k)}{\Delta_{\mathrm{Hi}}^2(i)}$. Therefore, for all $k \geq \bar{k}_i$, $N_{\mathrm{Lo}}^k(i) > \frac{k}{2}$ implies $N_{\mathrm{Lo}}^k(i) \geq \frac{128\alpha \log f(k)}{\Delta_{\mathrm{Hi}}^2(i)}$ and we have:

$$\mathbb{I}[\{\underline{\mu}_{\mathrm{Lo}}^k(i) \leq \overline{\mu}_{\mathrm{Hi}}^k(i) + \varepsilon\} \cap \{N_{\mathrm{Lo}}^k(i) \geq \frac{k}{2}\} \cap \{i = \underline{\pi}_{\mathrm{Lo}}^k\} \cap \{\pi_{\mathrm{Hi}}^k = i\}]$$

$$= \mathbb{I}[\{\underline{\mu}_{\mathrm{Lo}}^k(i) - \mu_{\mathrm{Lo}}(i) + \frac{\Delta_{\mathrm{Hi}}(i)}{4} \leq \overline{\mu}_{\mathrm{Hi}}^k(i) \pm \mu_{\mathrm{Hi}}(i) \pm \mu_{\mathrm{Hi}}(i^*) \pm \mu_{\mathrm{Lo}}(i^*) - \mu_{\mathrm{Lo}}(i) + \varepsilon + \frac{\Delta_{\mathrm{Hi}}(i)}{4}\}$$

$$\cap \{N_{\mathrm{Lo}}^k(i) \geq \frac{128\alpha \log f(k)}{\Delta_{\mathrm{Hi}}^2(i)}\} \cap \{i = \underline{\pi}_{\mathrm{Lo}}^k\} \cap \{\pi_{\mathrm{Hi}}^k = i\}]$$

$$\leq \mathbb{I}[\{\underline{\mu}_{\mathrm{Lo}}^k(i) - \mu_{\mathrm{Lo}}(i) + \frac{\Delta_{\mathrm{Hi}}(i)}{4} \leq \overline{\mu}_{\mathrm{Hi}}^k(i) - \mu_{\mathrm{Hi}}(i) - \Delta_{\mathrm{Hi}}(i) + \frac{\Delta_{\mathrm{Hi}}(i)}{4} + \frac{\Delta_{\mathrm{Hi}}(i)}{8} + \frac{\Delta_{\mathrm{Hi}}(i)}{8} + \frac{\Delta_{\mathrm{Hi}}(i)}{4}\}$$

$$(\Delta_{\mathrm{Lo}}(i) \leq \frac{\Delta_{\mathrm{Hi}}(i)}{4}; \text{Optimal value dominance } (\mu_{\mathrm{Hi}}(i^*) - \mu_{\mathrm{Lo}}(i^*) \leq \frac{\Delta_{\min}}{2} \leq \frac{\Delta_{\mathrm{Hi}}(i)}{8}); \varepsilon < \frac{\Delta_{\min}}{4} \leq \frac{\Delta_{\mathrm{Hi}}(i)}{8})$$

$$\cap \{N_{\mathrm{Lo}}^k(i) \geq \frac{128\alpha \log f(k)}{\Delta_{\mathrm{Hi}}^2(i)}\} \cap \{i = \underline{\pi}_{\mathrm{Lo}}^k\} \cap \{\pi_{\mathrm{Hi}}^k = i\}]$$

$$\leq \mathbb{I}[\{\underline{\mu}_{\mathrm{Lo}}^k(i) - \mu_{\mathrm{Lo}}(i) + \frac{\Delta_{\mathrm{Hi}}(i)}{4} \leq \overline{\mu}_{\mathrm{Hi}}^k(i) - \mu_{\mathrm{Hi}}(i) - \frac{\Delta_{\mathrm{Hi}}(i)}{4}\} \cap \{N_{\mathrm{Lo}}^k(i) \geq \frac{128\alpha \log f(k)}{\Delta_{\mathrm{Hi}}^2(i)}\} \cap \{i = \underline{\pi}_{\mathrm{Lo}}^k\} \cap \{\pi_{\mathrm{Hi}}^k = i\}]$$

$$(\Delta_{\mathrm{Lo}}(i) + \frac{\Delta_{\min}}{2} + \varepsilon + \frac{\Delta_{\mathrm{Hi}}(i)}{4} \leq \frac{\Delta_{\mathrm{Hi}}(i)}{4} + \frac{\Delta_{\mathrm{Hi}}(i)}{8} + \frac{\Delta_{\mathrm{Hi}}(i)}{16} + \frac{\Delta_{\mathrm{Hi}}(i)}{4} < \frac{3\Delta_{\mathrm{Hi}}(i)}{4})$$

$$\leq \mathbb{I}[\{\underline{\mu}_{\mathrm{Lo}}^k(i) - \mu_{\mathrm{Lo}}(i) + \frac{\Delta_{\mathrm{Hi}}(i)}{4} \leq 0\} \cap \{N_{\mathrm{Lo}}^k(i) \geq \frac{128\alpha \log f(k)}{\Delta_{\mathrm{Hi}}^2(i)}\} \cap \{i = \underline{\pi}_{\mathrm{Lo}}^k\} \cap \{\pi_{\mathrm{Hi}}^k = i\}]$$

$$+ \mathbb{I}[\{0 \leq \overline{\mu}_{\mathrm{Hi}}^k(i) - \mu_{\mathrm{Hi}}(i) - \frac{\Delta_{\mathrm{Hi}}(i)}{4}\} \cap \{i = \underline{\pi}_{\mathrm{Lo}}^k\} \cap \{\pi_{\mathrm{Hi}}^k = i\}]$$

$$= \mathbb{I}[\{\widehat{\mu}_{\mathrm{Lo}}^k(i) - \mu_{\mathrm{Lo}}(i) \leq \sqrt{\frac{2\alpha \log f(k)}{N_{\mathrm{Lo}}^k(i)}} - \frac{\Delta_{\mathrm{Hi}}(i)}{4}\} \cap \{N_{\mathrm{Lo}}^k(i) \geq \frac{128\alpha \log f(k)}{\Delta_{\mathrm{Hi}}^2(i)}\}]$$

$$+ \mathbb{I}[\{\widehat{\mu}_{\mathrm{Hi}}^k(i) - \mu_{\mathrm{Hi}}(i) + \sqrt{\frac{2\alpha \log f(k)}{N_{\mathrm{Hi}}^k(i)}} \geq \frac{\Delta_{\mathrm{Hi}}(i)}{4}\} \cap \{i = \underline{\pi}_{\mathrm{Lo}}^k\} \cap \{\pi_{\mathrm{Hi}}^k = i\}]$$

$$\leq \mathbb{I}[\widehat{\mu}_{\mathrm{Lo}}^k(i) - \mu_{\mathrm{Lo}}(i) \leq -\sqrt{\frac{2\alpha \log f(k)}{N_{\mathrm{Lo}}^k(i)}}] + \mathbb{I}[\{\widehat{\mu}_{\mathrm{Hi}}^k(i) - \mu_{\mathrm{Hi}}(i) + \sqrt{\frac{2\alpha \log f(k)}{N_{\mathrm{Hi}}^k(i)}} \geq \frac{\Delta_{\mathrm{Hi}}(i)}{4}\} \cap \{i = \underline{\pi}_{\mathrm{Lo}}^k\} \cap \{\pi_{\mathrm{Hi}}^k = i\}].$$

$$(13)$$

By taking the expectation over both sides of Eq. (12), we have:

$$\mathbb{E}[N_{\mathrm{Hi}}^K(i)]$$

$$\leq \bar{k}_i + \sum_{k=\bar{k}_i+1}^{K} \Pr(\widehat{\mu}_{\mathrm{Lo}}^k(i) - \mu_{\mathrm{Lo}}(i) \leq -\sqrt{\frac{2\alpha \log f(k)}{N_{\mathrm{Lo}}^k(i)}}) + \sum_{k=1}^{K} \Pr(0 \geq \widehat{\mu}_{\mathrm{Hi}}^k(i_{\mathrm{Hi}}^*) + \sqrt{\frac{2\alpha \log f(k)}{N_{\mathrm{Hi}}^k(i_{\mathrm{Hi}}^*)}} - \mu_{\mathrm{Hi}}(i_{\mathrm{Hi}}^*))$$

$$+ \mathbb{E}[\sum_{k=\bar{k}_i+1}^{K} \mathbb{I}[\{\widehat{\mu}_{\mathrm{Hi}}^k(i) - \mu_{\mathrm{Hi}}(i) + \sqrt{\frac{2\alpha \log f(k)}{N_{\mathrm{Hi}}^k(i)}} \geq \frac{\Delta_{\mathrm{Hi}}(i)}{4}\} \cap \{i = \underline{\pi}_{\mathrm{Lo}}^k\} \cap \{\pi_{\mathrm{Hi}}^k = i\}]]$$

$$+ \mathbb{E}[\sum_{k=1}^{K} \mathbb{I}[\widetilde{\mu}_{\mathrm{Hi}}^k(i) + \sqrt{\frac{2\alpha \log f(k)}{k}} - \mu_{\mathrm{Hi}}(i) - \Delta_{\mathrm{Hi}}(i) \geq 0]]$$

$$\leq \bar{k}_i + \sum_{k=\bar{k}_i+1}^{K} \Pr(\widehat{\mu}_{\text{Lo}}^k(i) - \mu_{\text{Lo}}(i) \leq -\sqrt{\frac{2\alpha \log f(k)}{N_{\text{Lo}}^k(i)}}) + \sum_{k=1}^{K} \Pr(0 \geq \widehat{\mu}_{\text{Hi}}^k(i_{\text{Hi}}^*) + \sqrt{\frac{2\alpha \log f(k)}{N_{\text{Hi}}^k(i_{\text{Hi}}^*)}} - \mu_{\text{Hi}}(i_{\text{Hi}}^*))$$

$$+ 2\mathbb{E}[\sum_{k=1}^{K} \mathbb{I}[\widetilde{\mu}_{\text{Hi}}^k(i) + \sqrt{\frac{2\alpha \log f(k)}{k}} - \mu_{\text{Hi}}(i) \geq \frac{\Delta_{\text{Hi}}(i)}{4}]]$$

$$\leq \bar{k}_i + \sum_{k=\bar{k}_i+1}^{K} \frac{2}{k^{2\alpha-1}} + 2 + \frac{64}{\Delta_{\text{Hi}}^2(i)}(\alpha \log f(K) + \sqrt{\pi \alpha \log f(K)} + 1) \qquad \text{(Lem. 8.2 in [17])}$$

$$= O(\frac{1}{\Delta_{\text{Hi}}^2(i)} \log \frac{AK}{\Delta_{\min}}).$$

Since $k_{\max} = \text{Poly}(A, \frac{1}{\Delta_{\min}})$, combining both cases, we have:

$$\text{Regret}_K(M_{\text{Hi}}) = \sum_{i \neq i^*} \Delta_{\text{Hi}}(i) \cdot \mathbb{E}[N_{\text{Hi}}^K(i)] \leq \sum_{i \neq i^*} \Delta_{\text{Hi}}(i)\mathbb{E}[N_{\text{Hi}}^{k_{\max}}(i)] + \sum_{k=k_{\max}+1}^{K} \Pr(\pi_{\text{Hi}}^k \neq i_{\text{Hi}}^*)$$

$$= O(\sum_{i \neq i^*} \frac{1}{\Delta_{\text{Hi}}(i)} \log \frac{A}{\Delta_{\min}}).$$

**Case 2: $M_{\text{Hi}}$ and $M_{\text{Lo}}$ are not $\varepsilon$-close** In that case, we will use $i_{\text{Lo}}^*$ and $i_{\text{Hi}}^*$ to denote optimal arm in $M_{\text{Lo}}$ and $M_{\text{Hi}}$, respectively, and either $i_{\text{Lo}}^* = i_{\text{Hi}}^*$ but $\mu_{\text{Lo}}(i_{\text{Lo}}^*) \geq \mu_{\text{Lo}}(i_{\text{Hi}}^*) + \varepsilon$, or $i_{\text{Lo}}^* \neq i_{\text{Hi}}^*$ and as a result of Assump. B:

$$\mu_{\text{Hi}}(i_{\text{Lo}}^*) = \mu_{\text{Hi}}(i_{\text{Hi}}^*) - \Delta_{\text{Hi}}(i_{\text{Lo}}^*) \leq \mu_{\text{Lo}}(i_{\text{Lo}}^*) + \frac{\Delta_{\min}}{2} - \Delta_{\text{Hi}}(i_{\text{Lo}}^*) \leq \mu_{\text{Lo}}(i_{\text{Lo}}^*) - \frac{\Delta_{\text{Hi}}(i_{\text{Lo}}^*)}{2}. \quad (14)$$

Next, we first separate arms other than $i_{\text{Lo}}^*$ and $i_{\text{Hi}}^*$ into three cases:

**Case 2-(a) $i \neq i_{\text{Lo}}^*, i \neq i_{\text{Hi}}^*$ and $0 < \Delta_{\text{Hi}}(i) \leq 4\Delta_{\text{Lo}}(i)$** The analysis is the same as Case 1-(a), and we have $\mathbb{E}[N_{\text{Hi}}^K(i)] = O(\frac{1}{\Delta_{\text{Hi}}^2(i)} \log \frac{AK}{\Delta_{\min}})$.

**Case 2-(b) $i \neq i_{\text{Lo}}^*, i \neq i_{\text{Hi}}^*$ and $\Delta_{\text{Hi}}(i) > 4\Delta_{\text{Lo}}(i) > 0$** The analysis is the same as Case 1-(b), and we have $\mathbb{E}[N_{\text{Hi}}^K(i)] = O(\frac{1}{\Delta_{\text{Hi}}^2(i)} \log \frac{AK}{\Delta_{\min}})$.

**Case 2-(c) Others** If $i_{\text{Lo}}^* = i_{\text{Hi}}^*$, $M_{\text{Hi}}$ suffers no regret when choosing $i = i_{\text{Lo}}^*$, and therefore:

$$\text{Regret}_K(M_{\text{Hi}}) = \sum_{i \neq i_{\text{Hi}}^*} \Delta_{\text{Hi}}(i) \cdot \mathbb{E}[N_{\text{Hi}}^k(i)] \leq \sum_{i \neq i_{\text{Hi}}^*} \Delta_{\text{Hi}}(i) \cdot O(\frac{1}{\Delta_{\text{Hi}}^2(i)} \log \frac{AK}{\Delta_{\min}}) = O(\sum_{i \neq i_{\text{Hi}}^*} \frac{1}{\Delta_{\text{Hi}}(i)} \log \frac{AK}{\Delta_{\min}}).$$

In the following, we study the case when $i_{\text{Lo}}^* \neq i_{\text{Hi}}^*$. For arm $i = i_{\text{Lo}}^*$, we define $k_{\max}' = \frac{c_{\max}'\alpha}{\Delta_{\text{Hi}}(i_{\text{Lo}}^*)^2} \log \frac{\alpha A}{\Delta_{\min}}$, where $c_{\max}'$ is the minimal constant, such that for all $k \geq \frac{c_{\max}'\alpha}{\Delta_{\text{Hi}}(i_{\text{Lo}}^*)^2} \log \frac{\alpha A}{\Delta_{\min}}$, we always have $k \geq \frac{1024\alpha \log f(k)}{\Delta_{\text{Hi}}(i_{\text{Lo}}^*)^2}$. Similar to Eq. (13), we check the following event for $k \geq k_{\max}'$:

$$\mathbb{I}[\{\underline{\mu}_{\text{Lo}}^k(\pi_{\text{Lo}}^k) \leq \overline{\mu}_{\text{Hi}}^k(\pi_{\text{Lo}}^k) + \varepsilon\} \cap \{N_{\text{Lo}}^k(\pi_{\text{Lo}}^k) > k/2\} \cap \{i_{\text{Lo}}^* = \pi_{\text{Lo}}^k\} \cap \{\pi_{\text{Hi}}^k = i_{\text{Lo}}^*\}]$$

$$= \mathbb{I}[\{\widehat{\mu}_{\text{Lo}}^k(i_{\text{Lo}}^*) - \mu_{\text{Lo}}(i_{\text{Lo}}^*) - \sqrt{\frac{2\alpha \log f(k)}{N_{\text{Lo}}^k(i_{\text{Lo}}^*)}} \leq \widehat{\mu}_{\text{Hi}}^k(i_{\text{Lo}}^*) - \mu_{\text{Hi}}(i_{\text{Lo}}^*) + (\mu_{\text{Hi}}(i_{\text{Lo}}^*) - \mu_{\text{Lo}}(i_{\text{Lo}}^*)) + \sqrt{\frac{2\alpha \log f(k)}{N_{\text{Hi}}^k(i_{\text{Lo}}^*)}} + \varepsilon\}$$

$$\cap \{N_{\text{Lo}}^k(\pi_{\text{Lo}}^k) > k/2\} \cap \{i_{\text{Lo}}^* = \pi_{\text{Lo}}^k\} \cap \{\pi_{\text{Hi}}^k = i_{\text{Lo}}^*\}]$$

$$\leq \mathbb{I}[\{\widehat{\mu}_{\text{Lo}}^k(i_{\text{Lo}}^*) - \mu_{\text{Lo}}(i_{\text{Lo}}^*) - \sqrt{\frac{2\alpha \log f(k)}{N_{\text{Lo}}^k(i_{\text{Lo}}^*)}} \leq \widehat{\mu}_{\text{Hi}}^k(i_{\text{Lo}}^*) - \mu_{\text{Hi}}(i_{\text{Lo}}^*) - \frac{\Delta_{\text{Hi}}(i_{\text{Lo}}^*)}{4} + \sqrt{\frac{2\alpha \log f(k)}{N_{\text{Hi}}^k(i_{\text{Lo}}^*)}}\}$$

$$\text{(As a result of Eq. (14), } \mu_{\text{Hi}}(i_{\text{Lo}}^*) - \mu_{\text{Lo}}(i_{\text{Lo}}^*) + \varepsilon \leq -\frac{\Delta_{\text{Hi}}(i_{\text{Lo}}^*)}{2} + \frac{\Delta_{\min}}{4} \leq -\frac{\Delta_{\text{Hi}}(i_{\text{Lo}}^*)}{4}\text{)}$$

$$\cap \{N_{\text{Lo}}^k(\pi_{\text{Lo}}^k) > k/2\} \cap \{i_{\text{Lo}}^* = \pi_{\text{Lo}}^k\} \cap \{\pi_{\text{Hi}}^k = i_{\text{Lo}}^*\}]$$

$$\leq \mathbb{I}[\{\widehat{\mu}_{\mathrm{Lo}}^k(i_{\mathrm{Lo}}^*) - \mu_{\mathrm{Lo}}(i_{\mathrm{Lo}}^*) - \sqrt{\frac{2\alpha \log f(k)}{N_{\mathrm{Lo}}^k(i_{\mathrm{Lo}}^*)}} \leq -\frac{\Delta_{\mathrm{Hi}}(i_{\mathrm{Lo}}^*)}{8}\} \cap \{N_{\mathrm{Lo}}^k(\pi_{\mathrm{Lo}}^k) > k/2\} \cap \{i_{\mathrm{Lo}}^* = \underline{\pi}_{\mathrm{Lo}}^k\} \cap \{\pi_{\mathrm{Hi}}^k = i_{\mathrm{Lo}}^*\}]$$

$$+ \mathbb{I}[\{\widehat{\mu}_{\mathrm{Hi}}^k(i_{\mathrm{Lo}}^*) - \mu_{\mathrm{Hi}}(i_{\mathrm{Lo}}^*) + \sqrt{\frac{2\alpha \log f(k)}{N_{\mathrm{Hi}}^k(i_{\mathrm{Lo}}^*)}} \geq \frac{\Delta_{\mathrm{Hi}}(i_{\mathrm{Lo}}^*)}{8}\} \cap \{N_{\mathrm{Lo}}^k(\pi_{\mathrm{Lo}}^k) > k/2\} \cap \{i_{\mathrm{Lo}}^* = \underline{\pi}_{\mathrm{Lo}}^k\} \cap \{\pi_{\mathrm{Hi}}^k = i_{\mathrm{Lo}}^*\}]$$

$$\leq \mathbb{I}[\widehat{\mu}_{\mathrm{Lo}}^k(i_{\mathrm{Lo}}^*) - \mu_{\mathrm{Lo}}(i_{\mathrm{Lo}}^*) \leq -\sqrt{\frac{2\alpha \log f(k)}{N_{\mathrm{Lo}}^k(i_{\mathrm{Lo}}^*)}}] + \mathbb{I}[\{\widehat{\mu}_{\mathrm{Hi}}^k(i_{\mathrm{Lo}}^*) - \mu_{\mathrm{Hi}}(i_{\mathrm{Lo}}^*) + \sqrt{\frac{2\alpha \log f(k)}{N_{\mathrm{Hi}}^k(i_{\mathrm{Lo}}^*)}} \geq \frac{\Delta_{\mathrm{Hi}}(i_{\mathrm{Lo}}^*)}{8}\} \cap \{\pi_{\mathrm{Hi}}^k = i_{\mathrm{Lo}}^*\}].$$

$$(15)$$

Therefore, by taking the expectation on both side of Eq. (12) and leveraging the above bound, we have:

$$\mathbb{E}[N_{\mathrm{Hi}}^k(i_{\mathrm{Lo}}^*)] \leq k_{\max}' + \mathbb{E}[\sum_{k=k_{\max}'+1}^K \mathbb{I}[\widehat{\mu}_{\mathrm{Lo}}^k(i_{\mathrm{Lo}}^*) - \mu_{\mathrm{Lo}}(i_{\mathrm{Lo}}^*) \leq -\sqrt{\frac{2\alpha \log f(k)}{N_{\mathrm{Lo}}^k(i_{\mathrm{Lo}}^*)}}]]$$

$$+ \mathbb{E}[\sum_{k=k_{\max}'+1}^K \mathbb{I}[\{\widehat{\mu}_{\mathrm{Hi}}^k(i_{\mathrm{Lo}}^*) - \mu_{\mathrm{Hi}}(i_{\mathrm{Lo}}^*) + \sqrt{\frac{2\alpha \log f(k)}{N_{\mathrm{Hi}}^k(i_{\mathrm{Lo}}^*)}} \geq \frac{\Delta_{\mathrm{Hi}}(i_{\mathrm{Lo}}^*)}{8}\} \cap \{\pi_{\mathrm{Hi}}^k = i_{\mathrm{Lo}}^*\}]]$$

$$+ \sum_{k=1}^K \Pr(0 \geq \widehat{\mu}_{\mathrm{Hi}}^k(i_{\mathrm{Hi}}^*) + \sqrt{\frac{2\alpha \log f(k)}{N_{\mathrm{Hi}}^k(i_{\mathrm{Hi}}^*)}} - \mu_{\mathrm{Hi}}(i_{\mathrm{Hi}}^*))$$

$$+ \mathbb{E}[\sum_{k=1}^K \mathbb{I}[\widetilde{\mu}_{\mathrm{Hi}}^k(i_{\mathrm{Lo}}^*) + \sqrt{\frac{2\alpha \log f(K)}{k}} - \mu_{\mathrm{Hi}}(i_{\mathrm{Lo}}^*) - \Delta_{\mathrm{Hi}}(i_{\mathrm{Lo}}^*) \geq 0]]$$

$$\leq k_{\max}' + \sum_{k=k_{\max}'+1}^K \Pr(\widehat{\mu}_{\mathrm{Lo}}^k(i_{\mathrm{Lo}}^*) - \mu_{\mathrm{Lo}}(i_{\mathrm{Lo}}^*) \leq -\sqrt{\frac{2\alpha \log f(k)}{N_{\mathrm{Lo}}^k(i_{\mathrm{Lo}}^*)}})$$

$$+ \sum_{k=1}^K \Pr(0 \geq \widehat{\mu}_{\mathrm{Hi}}^k(i_{\mathrm{Hi}}^*) + \sqrt{\frac{2\alpha \log f(k)}{N_{\mathrm{Hi}}^k(i_{\mathrm{Hi}}^*)}} - \mu_{\mathrm{Hi}}(i_{\mathrm{Hi}}^*))$$

$$+ 2\mathbb{E}[\sum_{k=1}^K \mathbb{I}[\widetilde{\mu}_{\mathrm{Hi}}^k(i_{\mathrm{Lo}}^*) + \sqrt{\frac{2\alpha \log f(K)}{k}} - \mu_{\mathrm{Hi}}(i_{\mathrm{Lo}}^*) \geq \frac{\Delta_{\mathrm{Hi}}(i_{\mathrm{Lo}}^*)}{8}]]$$

$$\leq k_{\max}' + 2\sum_{k=k_{\max}'+1}^K \frac{2}{f(k)^\alpha} + 2 \cdot (1 + \frac{128}{\Delta_{\mathrm{Hi}}^2(i_{\mathrm{Lo}}^*)}(\alpha \log f(K) + \sqrt{\pi\alpha \log f(K)} + 1))$$

$$= O(\frac{1}{\Delta_{\mathrm{Hi}}^2(i_{\mathrm{Lo}}^*)} \log \frac{AK}{\Delta_{\min}})$$

As a result, for arbitrary $K$, we also have:

$$\mathrm{Regret}_K(M_{\mathrm{Hi}}) = \sum_{i \neq i_{\mathrm{Hi}}^*} \Delta_{\mathrm{Hi}}(i) \cdot \mathbb{E}[N_{\mathrm{Hi}}^k(i)] \leq \sum_{i \neq i_{\mathrm{Hi}}^*} \Delta_{\mathrm{Hi}}(i) \cdot O(\frac{1}{\Delta_{\mathrm{Hi}}^2(i)} \log \frac{AK}{\Delta_{\min}}) = O(\sum_{i \neq i_{\mathrm{Hi}}^*} \frac{1}{\Delta_{\mathrm{Hi}}(i)} \log \frac{AK}{\Delta_{\min}}).$$

$$\square$$

# D  Proofs for RL Setting with Single Source/Low-Tier Task

## D.1  Missing Algorithms, Conditions and Notations

**Condition D.1** (Condition on $\mathrm{Alg}^{\mathrm{Lo}}$). $\mathrm{Alg}^{\mathrm{Lo}}$ is an algorithm which returns deterministic policies at each iteration, and there exists $C_1, C_2$ only depending on $S, A, H$ and $\Delta_{\min}$ but independent of $k$, such that for arbitrary $k \geq 2$, we have $\Pr(\mathcal{E}_{\mathrm{Alg}^{\mathrm{Lo}},k}) \geq 1 - \frac{1}{k^\alpha}$ for $\mathcal{E}_{\mathrm{Alg}^{\mathrm{Lo}},k}$ defined below:

$$\mathcal{E}_{\mathrm{Alg}^{\mathrm{Lo}},k} := \{\sum_{\widetilde{k}}^k V_{\mathrm{Lo},1}^*(s_1) - V_{\mathrm{Lo},1}^{\pi_{\mathrm{Lo}}^{\widetilde{k}}}(s_1) \leq C_1 + \alpha C_2 \log k\}.$$

---

**Algorithm 6:** ModelLearning

---
1 **Input**: Dataset $D$.
2 **for** $h = 1, 2, ..., H$ **do**
3    **for** $s_h \in \mathcal{S}_h, a_h \in \mathcal{A}_h$ **do**
4       Use $N_h(s_h, a_h)$ and $N_h(s_h, a_h, s_{h+1})$ to denote the number of times state, action (next state) occurs in the dataset $D$.
5       $\widehat{\mathbb{P}}_h(\cdot|s_h, a_h) \leftarrow \begin{cases} 0, & \text{if } N_h(s_h, a_h) = 0; \\ \frac{N_h(s_h, a_h, \cdot)}{N_h(s_h, a_h)}, & \text{otherwise.} \end{cases}$
6    **end**
7 **end**
8 **return** $\{\widehat{\mathbb{P}}_1, \widehat{\mathbb{P}}_2, ..., \widehat{\mathbb{P}}_H\}$.

---

**Remark D.2.** *We consider such condition to avoid unnecessary discussion on analyzing $Alg^{Lo}$. Although most of the existing near-optimal algorithms fixed the confidence level before the running of algorithm, as analyzed in Appx. G in [13], one may combine those algorithms with doubling trick to realize Cond. D.1 for $Alg^{Hi}$, only at the cost of increase the the regret of $Alg^{Lo}$ to $O(\log^2 K)$.*

**Condition D.3** (Condition on function **Bonus** in Alg. 2). Given a confidence sequence $\{\delta_k\}_{k=1}^K$ with $\delta_1, \delta_2, ..., \delta_K \in (0, 1/2)$, we define the following event at iteration $k \in [K]$ during the running of Alg. 2:

$$\mathcal{E}_{\textbf{Bonus},k} := \bigcap_{\substack{(\cdot)\in\{\text{Hi},\text{Lo}\}, \\ h\in[H], \\ s_h\in\mathcal{S}_h, a_h\in\mathcal{A}_h}} \left\{ \{ H \cdot \|\widehat{\mathbb{P}}^k_{(\cdot),h}(s_h, a_h) - \mathbb{P}_{(\cdot),h}(s_h, a_h)\|_1 < b^k_{(\cdot),h}(s_h, a_h) \leq B_1 \sqrt{\frac{\log(B_2/\delta_k)}{N^k_{(\cdot),h}(s_h, a_h)}} \} \right\}$$

we consider the choice of **Bonus** such that there exists such a $B_1$ and $B_2$ only depending on $S, A, H$ but independent of $\delta_k$, $k$ or $\Delta$, and $\Pr(\mathcal{E}_{\textbf{Bonus},k}) \geq 1 - \delta_k$ holds for any $k \in [K]$.[4]

For simplicity, in Cond. D.3, we directly control the $l_1$-norm of the error of model estimation. Our analysis framework is compatible with other bonus term for sharper analysis. We provide a simple example for the choice of $B_1$ and $B_2$ for completeness:

**Example D.4.** By Hoeffding's inequality and union bound, w.p. $1 - \delta$, for all $s_{h+1}, s_h, a_h$, we should have:

$$|\widehat{\mathbb{P}}^k_{(\cdot),h}(s_{h+1}|s_h, a_h) - \mathbb{P}_{(\cdot),h}(s_{h+1}|s_h, a_h)| \leq \sqrt{\frac{1}{2N^k_{(\cdot),h}(s_h, a_h)} \log \frac{S^2 A}{\delta}}.$$

which implies:

$$H \cdot \|\widehat{\mathbb{P}}^k_{(\cdot),h}(\cdot|s_h, a_h) - \mathbb{P}_{(\cdot),h}(\cdot|s_h, a_h)\|_1 = O(SH \sqrt{\frac{\log(SA/\delta)}{N^k_{(\cdot),h}(s_h, a_h)}})$$

Therefore, one can choose $B_1 = \Theta(SH)$ and $B_2 = \Theta(SA)$.

Finally, we introduce the following concentration events about the deviation of the empirical visitation frequency and its expectation:

$$\mathcal{E}_{\textbf{Con},k} := \bigcap_{\substack{h\in[H], \\ s_h\in\mathcal{S}_h, \\ a_h\in\mathcal{A}_h}} \left\{ \{ \frac{1}{2}\sum_{\widetilde{k}=1}^k d^{\pi_{\text{Lo}}^{\widetilde{k}}}(s_h, a_h) - \alpha \log(2SAHk) \leq N^k_{\text{Lo},h}(s_h, a_h) \leq e\sum_{\widetilde{k}=1}^k d^{\pi_{\text{Lo}}^{\widetilde{k}}}(s_h, a_h) + \alpha \log(2SAHk) \} \right.$$

$$\left. \cap \{ \frac{1}{2}\sum_{\widetilde{k}=1}^k d^{\pi_{\text{Hi}}^{\widetilde{k}}}(s_h, a_h) - \alpha \log(2SAHk) \leq N^k_{\text{Hi},h}(s_h, a_h) \leq e\sum_{\widetilde{k}=1}^k d^{\pi_{\text{Hi}}^{\widetilde{k}}}(s_h, a_h) + \alpha \log(2SAHk) \} \right\}.$$

(16)

---
[4]Note that we do not require the knowledge of $\Delta_i$'s to compute $b_{k,h}$.

## D.2 Some Basic Lemma

**Lemma D.5** (Underestimation). *Given a **Bonus** satisfying Cond. D.3, at each iteration $k$ during the running of Alg. 2, on the events $\mathcal{E}_{\textbf{Bonus},k}$ defined in Cond. D.3, $\forall h \in [H], \forall s_h \in \mathcal{S}_h, a_h \in \mathcal{A}_h$, we have:*

$$\underline{Q}_{Hi,h}^{\pi_{Hi}^k}(s_h, a_h) \leq Q_{Hi,h}^{\pi_{Hi}^k}(s_h, a_h) \leq Q_{Hi,h}^*(s_h, a_h) \tag{17}$$

$$\underline{Q}_{Lo,h}^k(s_h, a_h) \leq Q_{Lo,h}^{\pi_{Lo}^k}(s_h, a_h) \leq Q_{Lo,h}^*(s_h, a_h) \tag{18}$$

$$Q_{Lo,h}^*(s_h, a_h) - \underline{Q}_{Lo,h}^k(s_h, a_h) \leq 2\mathbb{E}_{\pi_{Lo}^*}\Big[\sum_{h'=h}^{H} \min\{H, b_{Lo,h'}^k(s_{h'}, a_{h'})\}|s_{h'}, a_{h'}\Big]. \tag{19}$$

*Proof.* Note that the relationship between $Q^\pi$ and $Q^*$ will always hold, and therefore, we only compare the underestimation part.

According to the initialization, we have $\underline{Q}_{Hi,h}^k(s_h, a_h) = Q_{Hi,h}^{\pi_{Hi}^k}(s_h, a_h) = Q_{Lo,h}^*(s_h, a_h)$ at $h = H+1$. Under the event of $\mathcal{E}_{\textbf{Bonus},k}$, at iteration $k$, suppose we have the inequality Eq. (17) holds for step $h+1$ for some $h \in [H]$, then at step $h$, we have:

$$\underline{Q}_{Hi,h}^{\pi_{Hi}^k}(s_h, a_h) - Q_{Hi,h}^{\pi_{Hi}^k}(s_h, a_h) = \widehat{\mathbb{P}}_{Hi,h}^k \underline{V}_{Hi,h+1}^{\pi_{Hi}^k}(s_h, a_h) - b_{Hi,h}^k(s_h, a_h) - \mathbb{P}_{Hi,h} V_{Hi,h+1}^{\pi_{Hi}^k}(s_h, a_h)$$

$$= (\widehat{\mathbb{P}}_{Hi,h}^k - \mathbb{P}_{Hi,h}) \underline{V}_{Hi,h+1}^{\pi_{Hi}^k}(s_h, a_h) - b_{Hi,h}^k(s_h, a_h) + \mathbb{P}_{Hi,h}(\underline{V}_{Hi,h+1}^{\pi_{Hi}^k} - V_{Hi,h+1}^{\pi_{Hi}^k})(s_h, a_h)$$

$$\leq \mathbb{P}_{Hi,h}(\underline{Q}_{Hi,h+1}^k(\cdot, \pi_{Hi}^k) - Q_{Hi,h+1}^{\pi_{Hi}^k}(\cdot, \pi_{Hi}^k))(s_h, a_h) \leq 0.$$

where the first inequality is because $(\widehat{\mathbb{P}}_{Hi,h}^k - \mathbb{P}_{Hi,h}) \underline{V}_{Hi,h+1}^{\pi_{Hi}^k}(s_h, a_h) \leq H \cdot \|\widehat{\mathbb{P}}_{Hi,h}^k(s_h, a_h) - \mathbb{P}_{Hi,h}(s_h, a_h)\|_1 \leq b_{Hi,h}^k(s_h, a_h)$. The proof for Eq. (18) is similar (except $\underline{Q}_{Lo,h}^k$ is the greedy value $\underline{\pi}_{Lo}^k$ instead of $\pi_{Lo}^k$). Besides,

$$Q_{Lo,h}^*(s_h, a_h) - \underline{Q}_{Lo,h}^k(s_h, a_h)$$

$$= \min\{H, \mathbb{P}_{Lo,h} V_{Lo,h+1}^*(s_h, a_h) - \widehat{\mathbb{P}}_{Lo,h}^k \underline{V}_{Lo,h+1}^k(s_h, a_h) + b_{Lo,h}^k(s_h, a_h)\}$$

$$\leq \min\{H, (\mathbb{P}_{Lo,h} - \widehat{\mathbb{P}}_{Lo,h}^k)\underline{V}_{Lo,h+1}^k(s_h, a_h) + b_{Lo,h}^k(s_h, a_h)\} + \mathbb{P}_{Lo,h}(V_{Lo,h+1}^* - \underline{V}_{Lo,h+1}^k)(s_h, a_h)$$

$$\leq 2\min\{H, b_{Lo,h}^k(s_h, a_h)\} + \mathbb{P}_{Lo,h}(V_{Lo,h+1}^* - \underline{Q}_{Lo,h+1}^k(\cdot, \pi_{Lo}^*))(s_h, a_h)$$

$$\leq ... \leq 2\mathbb{E}_{\pi_{Lo}^*}\Big[\sum_{h'=h}^{H} \min\{H, b_{Lo,h'}^k(s_{h'}, a_{h'})\}|s_h, a_h\Big].$$

$\square$

**Theorem D.6** (Extended from Thm. 4.7 in [13]). *For an arbitrary sequence of deterministic policies $\pi^1, \pi^2, ..., \pi^K$, there must exist a sequence of deterministic optimal policies $\pi^{1,*}, \pi^{2,*}, ..., \pi^{K,*}$, such that $\forall h \in [H], s_h \in \mathcal{S}_h, a_h \in \mathcal{A}_h$:*

$$\Big|\sum_{k=1}^{K} d^{\pi^k}(s_h, a_h) - \sum_{k=1}^{K} d^{\pi^{k,*}}(s_h, a_h)\Big| \leq \frac{1}{\Delta_{\min}}\Big(\sum_{k=1}^{K} V_1^*(s_1) - V_1^{\pi^k}(s_1)\Big).$$

*Proof.* We first define the following events:

$$\mathcal{E}_{k,h,\pi} := \{\pi_{k,h}(s_h) \neq \pi_h(s_h)\}, \quad \widetilde{\mathcal{E}}_{k,h,\pi} := \mathcal{E}_{k,h,\pi} \cap \bigcap_{h'=1}^{h} \mathcal{E}_{k,h'-1,\pi}^{\complement}, \quad \bar{\mathcal{E}}_{k,\pi} := \bigcup_{h=1}^{H} \mathcal{E}_{k,h,\pi}.$$

From Thm. 4.7 in [13], we already know $\sum_{k=1}^{K} d^{\pi^k}(s_h, a_h) - \sum_{k=1}^{K} d^{\pi^{k,*}}(s_h, a_h) \geq -\frac{1}{\Delta_{\min}}\Big(\sum_{k=1}^{K} V_1^*(s_1) - V_1^{\pi^k}(s_1)\Big)$. Next, we start with the second step in the proof of Lem.

E.3 in [13]: by choosing $\delta_{s_h,a_h} := \mathbb{I}[S_h = s_h, A_h = a_h]$ (which equals one if the state action is $(s_h, a_h)$ at step $h$ and otherwise 0) as reward function, we have:

$$d^\pi(s_h, a_h) - d^{\pi^k}(s_h, a_h) = V_1^\pi(s_1; \delta_{s_h,a_h}) - V_1^{\pi^k}(s_1; \delta_{s_h,a_h})$$

$$= \mathbb{E}_{\pi^k}\left[\sum_{h'=1}^{h} \mathbb{I}[\widetilde{\mathcal{E}}_{k,h',\pi}](V_{h'}^\pi(s_{h'}; \delta_{s_h,a_h}) - V_{h'}^{\pi^k}(s_{h'}; \delta_{s_h,a_h}))\right]$$

$$(V_{h'}^\pi = V_{h'}^{\pi^k} = 0 \text{ for all } h' \geq h + 1)$$

Starting from here, we do something differently:

$$d^\pi(s_h, a_h) - d^{\pi^k}(s_h, a_h) \geq \mathbb{E}_{\pi^k}\left[\sum_{h'=1}^{h} -\mathbb{I}[\widetilde{\mathcal{E}}_{k,h',\pi}]V_{h'}^{\pi^k}(s_{h'}; \delta_{s_h,a_h})\right] \qquad (V_{h'}^\pi \geq 0)$$

$$\geq -\mathbb{E}_{\pi^k}\left[\sum_{h'=1}^{h} \mathbb{I}[\widetilde{\mathcal{E}}_{k,h',\pi}]\right] \qquad (V_{h'}^{\pi^k} \leq 1)$$

$$\geq -\mathbb{E}_{s_1,a_1,s_2,a_2...,s_H,a_H \sim \pi^k}[\mathbb{I}[\bar{\mathcal{E}}_{k,\pi}]] = -\Pr(\bar{\mathcal{E}}_{k,\pi}|\pi^k)$$

Therefore, combining with the results in Lem. E.3, we can conclude that:

$$d^\pi(s_h, a_h) - d^{\pi^k}(s_h, a_h) \geq -\Pr(\bar{\mathcal{E}}_{k,\pi}|\pi^k)$$

We define $\pi^{k,*}$ to be a policy that equals $\pi^k$ on those states where $\pi^k$ is optimal, and takes the optimal action when $\pi^k$ is non-optimal, then we have:

$$V_1^{\pi^{k,*}}(s_1) - V_1^{\pi^k}(s_1) = \mathbb{E}_{\pi^k}\left[\sum_{h=1}^{H} \mathbb{I}[\widetilde{\mathcal{E}}_{k,h,\pi^{k,*}}](V_h^{\pi^{k,*}}(s_h) - V_h^{\pi^k}(s_h))\right]$$

$$\geq \mathbb{E}_{\pi^k}\left[\sum_{h=1}^{H} \mathbb{I}[\widetilde{\mathcal{E}}_{k,h,\pi^{k,*}}](V_h^{\pi^{k,*}}(s_h) - Q_h^{\pi^{k,*}}(s_h, \pi^k(s_h)))\right]$$

$$\geq \mathbb{E}_{\pi^k}\left[\sum_{h=1}^{H} \mathbb{I}[\widetilde{\mathcal{E}}_{k,h,\pi^{k,*}}]\Delta_{\min}\right] = \Delta_{\min}\Pr(\bar{\mathcal{E}}_{k,\pi^{k,*}}|\pi^k)$$

$$\geq \Delta_{\min}(d^{\pi^k}(s_h, a_h) - d^{\pi^{k,*}}(s_h, a_h)).$$

Sum over all $k \in [K]$, we have

$$\sum_{k=1}^{K} d^{\pi^k}(s_h, a_h) - \sum_{k=1}^{K} d^{\pi^{k,*}}(s_h, a_h) \leq \frac{1}{\Delta_{\min}}\left(\sum_{k=1}^{K} V_1^*(s_1) - V_1^{\pi^k}(s_1)\right).$$

which completes the proof. □

**Corollary D.7** (Unique Optimal Policy). *Under Assump. A, Thm. D.6 implies that:*

$$\left|\sum_{k=1}^{K} d_{Lo}^{\pi_{Lo}^k}(s_h, a_h) - K d^{\pi_{Lo}^*}(s_h, a_h)\right| \leq \frac{1}{\Delta_{\min}}\left(\sum_{k=1}^{K} V_{Lo,1}^*(s_1) - V_{Lo,1}^{\pi_{Lo}^k}(s_1)\right)$$

**Lemma D.8.** *Let $\mathcal{F}_i$ for $i, 1...$ be a filtration and $X_1, ...X_n$ be a sequence of Bernoulli random variables with $\Pr(X_i = 1|\mathcal{F}_{i-1}) = P_i$ with $P_i$ being $\mathcal{F}_{i-1}$-measurable and $X_i$ being $\mathcal{F}_i$ measurable. It holds that*

$$\Pr(\exists n : \sum_{t=1}^{n} X_t < \frac{1}{2}\sum_{t=1}^{n} P_t - W) \leq e^{-W}; \quad \Pr(\exists n : \sum_{t=1}^{n} X_t > e\sum_{t=1}^{n} P_t - W) \leq e^{-W}.$$

*Proof.* The first inequality has been proven in Lemma F.4 of [6]. Here we adopt similar techniques to prove the second one.

We first define $m_t := e^{X_t - eP_t}$, since $X_t$ is a Bernoulli random variable with $\Pr(X_t = 1) = P_t$, we should have:

$$\mathbb{E}_{X_t}[e^{X_t - eP_t}|\mathcal{F}_{t-1}] = \frac{eP_t + (1 - P_t)}{e^{eP_t}} \leq \frac{eP_t + (1 - P_t)}{eP_t + 1} \leq 1$$

where in the last but two step, we use $e^x \geq x + 1$. Therefore, $M_n := \prod_{t=1}^n m_t = e^{\sum_{t=1}^n X_t - eP_t}$ is a supermartingale. By Markov inequality, we have:

$$\Pr(\sum_{t=1}^n X_t - eP_t \geq W) = \Pr(M_n \geq e^W) \leq \frac{\mathbb{E}[M_n]}{e^W} \leq e^{-W}.$$

As a result, for a fixed $n$, we have $\Pr(\sum_{t=1}^n X_t \geq eP_t + W) \leq e^{-W}$. After a similar discussion about stopping time as [6], we have $\Pr(\exists n : \sum_{t=1}^n X_t \geq eP_t + W) \leq e^{-W}$. $\qquad\square$

As a direct result of Lem. D.8, we have the following result:

**Lemma D.9.** *For arbitrary $k \geq 1$, and arbitrary $\alpha > 2$, $\Pr(\mathcal{E}_{\textbf{Con},k}) \geq 1 - \frac{1}{k^\alpha}$.*

## D.3 Analysis of $\text{Alg}^{\text{Lo}}$

**Lemma D.10** (The relationship between $d_{\text{Lo}}^*$ and $N_{\text{Lo}}^k$). *There exists a constant $c_{occup}$ which is independent of $\lambda, S, A, H$ and gap $\Delta$, s.t., for all $k \geq k_{occup} := c_{occup} \frac{C_1 + \alpha C_2}{\lambda \Delta_{\min}} \log(\frac{\alpha C_1 C_2 SAH}{\lambda \Delta_{\min}})$, on the events of $\mathcal{E}_{\text{Alg}^{\text{Lo}},k}$ and $\mathcal{E}_{\textbf{Con},k}$, $N_{Lo,h}^k(s_h, a_h) \geq \frac{\lambda}{3} k$ implies that $d_{\text{Lo}}^*(s_h, a_h) \geq \frac{\lambda}{9} > 0$, and conversely, if $d_{Lo}^*(s_h, a_h) \geq \lambda$, we must have $N_{Lo,h}^k(s_h) \geq N_{Lo,h}^k(s_h, \pi_{Lo}^*) \geq \frac{\lambda}{3} k$.*

*Proof.* On the event of $\mathcal{E}_{\text{Alg}^{\text{Lo}},k}$ and $\mathcal{E}_{\textbf{Con},k}$, as a result of Cor. D.7, $N_{\text{Lo},h}^k(s_h, a_h) \geq \frac{\lambda}{3} k$ implies:

$$\frac{\lambda}{3} k \leq N_{\text{Lo},h}^k(s_h, a_h) \leq ekd_{\text{Lo}}^*(s_h, a_h) + \alpha \log(2SAHk) + \frac{1}{\Delta_{\min}}(C_1 + \alpha C_2 \log k)$$

There should exists a constant $c_{occup}$, such that, $\frac{3-e}{9} \lambda k \geq \alpha \log(2SAHk) + \frac{1}{\Delta_{\min}}(C_1 + \alpha C_2 \log k)$ can be satisfied for all $k \geq c_{occup} \frac{C_1 + \alpha C_2}{\lambda \Delta_{\min}} \log(\frac{\alpha C_1 C_2 SAH}{\lambda \Delta_{\min}})$, which implies that:

$$d_{\text{Lo}}^*(s_h, a_h) \geq \frac{\frac{\lambda}{3} k - \frac{3-e}{9} \lambda k}{ek} \geq \frac{\lambda}{9}.$$

On the other hand, if $d_{\text{Lo}}^*(s_h, a_h) \geq \lambda$, on the event $\mathcal{E}_{\textbf{Con},k}$ and Cor. D.7, we have:

$$
\begin{aligned}
N_{\text{Lo},h}^k(s_h, a_h) &\geq \frac{1}{2} kd_{\text{Lo}}^*(s_h, a_h) - \alpha \log(2SAHk) - \frac{1}{\Delta_{\min}}(C_1 + \alpha C_2 \log k) \\
&\geq \frac{\lambda}{2} k - \alpha \log(2SAHk) - \frac{1}{\Delta_{\min}}(C_1 + \alpha C_2 \log k).
\end{aligned}
$$

with the same $c_{occup}$, we have:

$$N_{\text{Lo},h}^k(s_h) \geq N_{\text{Lo},h}^k(s_h, a_h) \geq \frac{\lambda}{2} k - \frac{3-e}{9} \lambda k = \frac{3+2e}{18} \lambda k \geq \frac{\lambda}{3} k.$$

which finishes the proof. $\qquad\square$

**Lemma D.11** (Convergence Speed of PVI). *There exists an absolute constant $c_\Xi$, such that for arbitrary fixed $\xi > 0$ and $\lambda > 0$, and for arbitrary*

$$k \geq c_\Xi \max\{\frac{\alpha B_1^2 H^2 S}{\lambda^2 \xi^2} \log(\frac{\alpha HSAB_1 B_2}{\lambda \xi}), \frac{(C_1 + \alpha C_2)SH}{\Delta_{\min} \lambda \xi} \log \frac{C_1 C_2 SAH}{\Delta_{\min} \lambda \xi}\}. \qquad (20)$$

*on the event $\mathcal{E}_{\textbf{Con},k}, \mathcal{E}_{\textbf{Bonus},k}$ and $\mathcal{E}_{\text{Alg}^{\text{Lo}},k}$, for arbitrary $h \in [H], s_h \in \mathcal{S}_h$ with $N_{Lo,h}^k(s_h) > \frac{\lambda}{3}$, we have*

$$V_{Lo,h}^*(s_h) - \underline{V}_{Lo,h}^k(s_h) \leq \xi.$$

*Proof.* As a result of Lem. D.10, considering $c_\Xi \geq c_{occup}$, on the events of $\mathcal{E}_{\text{Alg}^{\text{Lo}},k}$ and $\mathcal{E}_{\text{Con},k}$, $N_{\text{Lo},h}^k(s_h) > \frac{\lambda}{3}$ implies $d_{\text{Lo}}^*(s_h) \geq \frac{\lambda}{9}$. According to the Lem. D.5, for arbitrary $s_h$ with $d_{\text{Lo}}^{\pi^*}(s_h) > 0$

$$V_{\text{Lo},h}^*(s_h) - \underline{V}_{\text{Lo},h}^k(s_h) \leq 2\mathbb{E}_{\pi_{\text{Lo}}^*, M_{\text{Lo}}}\Big[\sum_{h'=h}^H \min\{b_{\text{Lo},h'}^k(s_{h'}, \pi_{\text{Lo}}^*), H\}|s_h\Big]$$

$$= 2\sum_{h'=h}^H \sum_{s_{h'}, \pi_{\text{Lo}}^*} d_{\text{Lo}}^{\pi_{\text{Lo}}^*}(s_{h'}, \pi_{\text{Lo}}^*|s_h) \min\{b_{\text{Lo},h'}^k(s_{h'}, \pi_{\text{Lo}}^*), H\}$$

$$\leq \frac{2}{d_{\text{Lo}}^{\pi_{\text{Lo}}^*}(s_h)} \sum_{h'=h}^H \sum_{s_{h'}, \pi_{\text{Lo}}^*} d_{\text{Lo}}^{\pi_{\text{Lo}}^*}(s_{h'}, \pi_{\text{Lo}}^*) \min\{b_{\text{Lo},h'}^k(s_{h'}, \pi_{\text{Lo}}^*), H\}$$

$$\leq \frac{18}{\lambda} \mathbb{E}_{\pi_{\text{Lo}}^*, M_{\text{Lo}}}\Big[\sum_{h'=h}^H \min\{b_{\text{Lo},h'}^k(s_{h'}, \pi_{\text{Lo}}^*), H\}|s_1\Big]$$

Given threshold $\xi$, we define the following set: $\mathcal{Y}_{\geq h}^\xi := \bigcup_{h' \geq h}\{s_h | d_{\text{Lo}}^*(s_h) > \frac{\lambda\xi}{36SH}\}$. Note that for those $s_h \in \mathcal{S}_h \setminus \mathcal{Y}_{\geq h}^\xi$, we have:

$$\sum_{h'=h}^H \sum_{s_{h'} \in \mathcal{S}_{h'} \setminus \mathcal{Y}_{\geq h}^\xi} \frac{2}{\lambda}\mathbb{E}_{\pi_{\text{Lo}}^*, M_{\text{Lo}}}[\min\{b_{\text{Lo},h'}^k(s_{h'}, \pi_{\text{Lo}}^*), H\}|s_1] \leq SH \cdot \frac{18}{\lambda} \cdot \frac{\lambda\xi}{36SH} = \frac{\xi}{2}.$$

In the following, we study the bonus term for $s_h \in \mathcal{Y}_{\geq h}^\xi$. According to Lem. D.9, on the event of $\mathcal{E}_{\text{Alg}^{\text{Lo}},k}$ and $\mathcal{E}_{\text{Con},k}$, we have:

$$N_{\text{Lo},h}^k(s_h, a_h) \geq \frac{k}{2}d_{\text{Lo}}^*(s_h, a_h) - \frac{1}{\Delta_{\min}}(C_1 + \alpha C_2 \log k) - \alpha \log(2SAHk).$$

We define:

$$k_{s_h} := \arg\min_k \quad s.t. \quad k d_{\text{Lo}}^*(s_h, a_h)/4 \geq \frac{1}{\Delta_{\min}}(C_1 + \alpha C_2 \log k) + \alpha \log(2SAHk), \quad \forall k' \geq k$$

which implies that $k_{s_h} = c_0 \cdot \frac{C_1 + \alpha C_2}{\Delta_{\min} d_{\text{Lo}}^*(s_h, a_h)} \log \frac{C_1 C_2 SAH}{\Delta_{\min} d_{\text{Lo}}^*(s_h, a_h)} \leq c_0' \cdot \frac{(C_1 + \alpha C_2)SH}{\Delta_{\min}\lambda\xi} \log \frac{C_1 C_2 SAH}{\Delta_{\min}\lambda\xi}$ for some absolute constants $c_0$ and $c_0'$, where the second step we use $d_{\text{Lo}}^*(s_h, a_h) > \lambda\xi/36SH$. Therefore, for $k \geq k_{s_h}$, we have:

$$b_{\text{Lo},h}^k(s_h, a_h) \leq B_1\sqrt{\frac{\log(B_2/\delta_k)}{N_{\text{Lo},h}^k(s_h, a_h)}} \leq 2B_1\sqrt{\frac{\log(B_2/\delta_k)}{k d_{\text{Lo}}^*(s_h, a_h)}}.$$

which implies that:

$$\mathbb{E}_{\pi_{\text{Lo}}^*, M_{\text{Lo}}}[\min\{b_{\text{Lo},h}^k(s_h, \pi_{\text{Lo}}^*), H\}|s_1] = 2d_{\text{Lo}}^*(s_h, \pi_{\text{Lo}}^*) \cdot B_1\sqrt{\frac{\log(B_2/\delta_k)}{k d_{\text{Lo}}^*(s_h, \pi_{\text{Lo}}^*)}} = 2B_1\sqrt{\frac{d_{\text{Lo}}^*(s_h, \pi_{\text{Lo}}^*)\log(B_2/\delta_k)}{k}}.$$

Therefore,

$$\sum_{h'=h}^H \sum_{s_{h'} \in \mathcal{Y}_{\geq h}^\xi} \frac{18}{\lambda}\mathbb{E}_{\pi_{\text{Lo}}^*, M_{\text{Lo}}}[\min\{b_{\text{Lo},h'}^k(s_{h'}, \pi_{\text{Lo}}^*), H\}|s_1]$$

$$\leq \frac{36}{\lambda}B_1 \sum_{h'=h}^H \sum_{s_{h'} \in \mathcal{S}_{h'} \setminus \mathcal{Y}_{\geq h}^\xi} \sqrt{\frac{d_{\text{Lo}}^*(s_{h'}, \pi_{\text{Lo}}^*)\log(B_2/\delta_k)}{k}} \leq \frac{36}{\lambda}B_1\sqrt{\frac{\log(B_2/\delta_k)}{k}} \sum_{h'=h}^H \sum_{s_{h'} \in \mathcal{S}_{h'} \setminus \mathcal{Y}_{\geq h}^\xi} \sqrt{d_{\text{Lo}}^*(s_{h'}, \pi_{\text{Lo}}^*)}$$

$$\leq \frac{36}{\lambda}B_1\sqrt{\frac{\log(B_2/\delta_k)}{k}}\sqrt{SH\sum_{h'=h}^H \sum_{s_{h'} \in \mathcal{S}_{h'} \setminus \mathcal{Y}_{\geq h}^\xi} d_{\text{Lo}}^*(s_{h'}, \pi_{\text{Lo}}^*)} \leq \frac{36}{\lambda}B_1 H\sqrt{\frac{S\log(B_2/\delta_k)}{k}}.$$

Recall that $\delta_k = O(1/SAHk^\alpha)$, the RHS is less than $\frac{\xi}{2}$ when:

$$k \geq c_0'' \frac{\alpha B_1^2 H^2 S}{\lambda^2 \xi^2} \log(\frac{\alpha HSAB_1 B_2}{\lambda \xi}).$$

for some constant $c_0''$. Therefore, by choosing $c_\Xi = \max\{c_0', c_0''\}$ we can conclude that, as long as:

$$k \geq c_\Xi \max\{\frac{\alpha B_1^2 H^2 S}{\lambda^2 \xi^2} \log(\frac{\alpha HSAB_1 B_2}{\lambda \xi}), \frac{(C_1 + \alpha C_2)SH}{\Delta_{\min} \lambda \xi} \log \frac{C_1 C_2 SAH}{\Delta_{\min} \lambda \xi}\}.$$

we have $V^*_{\text{Lo},h}(s_h) - \underline{V}^k_{\text{Lo},h}(s_h) \leq \xi$.  $\square$

## D.4 Analysis of Regret on $M_{\text{Hi}}$

For the simplification of the notation, in the following, we will denote:

$$\zeta^k(s_h) := \mathbb{I}[\{\underline{Q}^k_{\text{Lo},h}(s_h, \pi^k_{\text{Lo},h}) \leq \widetilde{Q}^k_{\text{Hi},h}(s_h, \pi^k_{\text{Lo},h}) + \varepsilon\} \cap \{N^k_{\text{Lo},h}(s_h) > \frac{\lambda}{3}k\}].$$

In another word, $\zeta^k(s_h) = 1$ if and only if we will trust $M_{\text{Lo}}$ at state $s_h$ in Alg. 2 and therefore set $\pi^k_{\text{Hi}}$ by exploiting information from $M_{\text{Lo}}$. Next, we define the surplus.

**Definition D.12** (Definition of Surplus in Pessimistic Algorithm setting)**.** We define the surplus for pessimistic estimation in $M_{\text{Lo}}$ and optimistic estimation in $M_{\text{Hi}}$:

$$\mathbf{E}^k_{\text{Hi},h}(s_h, a_h) = \widetilde{Q}^k_{\text{Hi},h}(s_h, a_h) - \mathbb{P}_{\text{Hi},h} \widetilde{V}^k_{\text{Hi},h+1}(s_h, a_h) + \mathbb{P}_{\text{Hi},h} \underline{V}^{\pi^k_{\text{Hi}}}_{\text{Hi},h+1}(s_h, a_h) - \underline{Q}^{\pi^k_{\text{Hi}}}_{\text{Hi},h}(s_h, a_h).$$

We also define:

$$\ddot{\mathbf{E}}^k_{\text{Hi},h}(s_h, a_h) := \text{Clip}\left[\mathbf{E}^k_{\text{Hi},h}(s_h, a_h)\bigg| \max\{\frac{\Delta_{\min}}{4eH}, \frac{\Delta_{\text{Hi}}(s_h, a_h)}{4e}\}\right].$$

and

$$\ddot{Q}^\pi_{\text{Hi},h}(s_h, a_h) := r_{\text{Hi},h}(s_h, a_h) + \mathbb{P}_{\text{Hi},h}\ddot{V}^\pi_{\text{Hi},h+1}(s_h, a_h) + e\ddot{\mathbf{E}}^k_{\text{Hi},h}(s_h, a_h), \quad \ddot{V}^\pi_{\text{Hi},h}(s_h) := \ddot{Q}^\pi_{\text{Hi},h}(s_h, \pi).$$

We first show that $\widetilde{V}^k_{\text{Hi},h}$ will be an overestimation eventually.

**Theorem 4.3.** *There exists $k_{ost} = Poly(S, A, H, \lambda^{-1}, \Delta^{-1}_{\min})$, such that, for all $k \geq k_{ost}$, on some event $\mathcal{E}_k$ with $\mathbb{P}(\mathcal{E}_k) \leq 3\delta_k$, we have $Q^*_{Hi,h}(s_h, a_h) \leq \widetilde{Q}^k_{Hi,h}(s_h, a_h)$, $V^*_{Hi,h}(s_h) \leq \widetilde{V}^k_{Hi,h}(s_h), \forall h \in [H], s_h \in \mathcal{S}_h, a_h \in \mathcal{A}_h$ and*

$$V^*_{Hi,1}(s_1) - V^{\pi^k_{Hi}}_{Hi,1}(s_1) \leq 2e\mathbb{E}_{\pi^k_{Hi}}\left[\sum_{h=1}^H Clip\left[\min\{H, 4b^k_{Hi,h}(s_h, a_h)\}\big| \frac{\Delta_{\min}}{4eH} \vee \frac{\Delta_{Hi}(s_h, a_h)}{4e}\right]\right]. \quad (1)$$

*Proof.* In this theorem, $\mathcal{E}_k$ denotes the event $\mathcal{E}_{\text{Alg}^{\text{Lo}},k} \cap \mathcal{E}_{\textbf{Bonus},k} \cap \mathcal{E}_{\textbf{Con},k}$.

**Part 1: Proof of Overestimation** We do the proof by induction. First of all, note that the overestimation is true for horizon $h = H + 1$, since all the value is zero. Next, we assume the overestimation is true for step $h + 1$, we will show it holds for step $h$. We first show $\widetilde{Q}^k_{\text{Hi},h}$ is an overestimation:

$$\widetilde{Q}^k_{\text{Hi},h}(s_h, a_h) - Q^*_{\text{Hi},h}(s_h, a_h)$$
$$= \min\{H - Q^*_{\text{Hi},h}(s_h, a_h), \widehat{\mathbb{P}}^k_{\text{Hi},h}\widetilde{V}^k_{\text{Hi},h+1}(s_h, a_h) + b^k_{\text{Hi},h}(s_h, a_h) - \mathbb{P}_{\text{Hi},h}V^*_{\text{Hi},h+1}(s_h, a_h)\}$$
$$= \min\{H - Q^*_{\text{Hi},h}(s_h, a_h), (\widehat{\mathbb{P}}^k_{\text{Hi},h} - \mathbb{P}_{\text{Hi},h})\widetilde{V}^k_{\text{Hi},h+1}(s_h, a_h) + b^k_{\text{Hi},h}(s_h, a_h) + \mathbb{P}_{\text{Hi},h}(\widetilde{V}^k_{\text{Hi},h+1} - V^*_{\text{Hi},h+1})(s_h, a_h)\}$$
$$\geq \min\{H - Q^*_{\text{Hi},h}(s_h, a_h), \mathbb{P}_{\text{Hi},h}(\widetilde{V}^k_{\text{Hi},h+1} - V^*_{\text{Hi},h+1})(s_h, a_h)\} \geq 0$$

where the first inequality is because of event $\mathcal{E}_{\textbf{Bonus},k}$. In the following, we separate to three cases:

**Case 1:** $\zeta^k(s_h, a_h) = 0$ In this case,

$$\widetilde{V}^k_{\text{Lo},h}(s_h) = \max_a \widetilde{Q}^k_{\text{Hi},h}(s_h, a) \geq \widetilde{Q}^k_{\text{Hi},h}(s_h, \pi^*_{\text{Hi}}) \geq Q^*_{\text{Hi},h}(s_h, \pi^*_{\text{Hi}}) = V^*_{\text{Hi},h}(s_h).$$

**Case 2:** $\zeta^k(s_h, a_h) = 1$ **and** $\underline{\pi}^k_{\textbf{Lo},h}(s_h) = \pi^*_{\textbf{Hi},h}(s_h)$   In this case, as a result of Lem. D.5, we have:

$$\widetilde{V}^k_{\text{Lo},h}(s_h) = \widetilde{Q}^k_{\text{Hi},h}(s_h, \pi^*_{\text{Hi}}) + \frac{1}{H}(\widetilde{Q}^k_{\text{Hi},h}(s_h, \pi^*_{\text{Hi}}) - \underline{Q}^{\pi^k_{\text{Hi}}}_{\text{Hi},h}(s_h, \pi^*_{\text{Hi}})) \geq \widetilde{Q}^k_{\text{Hi},h}(s_h, \pi^*_{\text{Hi}}) \geq Q^*_{\text{Hi},h}(s_h, \pi^*_{\text{Hi}}) = V^*_{\text{Hi},h}(s_h).$$

**Case 3:** $\zeta^k(s_h, a_h) = 1$ **and** $\underline{\pi}^k_{\textbf{Lo},h}(s_h) \neq \pi^*_{\textbf{Hi},h}(s_h)$   This case is more complicated. Intuitively, we want to show that after $k_{ost}$, in case 3, the "uncerntainty" must be high, and therefore, adding $1/H$ of the uncerntainty interval will ensure the overestimation.

As a result of Lem. D.11, we choose $k_{ost}$ by plugging $\xi = \frac{\Delta_{\min}}{4(H+1)}$ into Eq. (20), which yields:

$$k_{ost} := c_{ost} \cdot \max\{\alpha \frac{B_1^2 H^2 S}{\lambda^2 \Delta_{\min}^2} \log(\alpha H S A B_1 B_2), \frac{(C_1 + \alpha C_2) S H}{\lambda \Delta_{\min}^2} \log \frac{C_1 C_2 S A H}{\Delta_{\min}\lambda}\}, \qquad (21)$$

for some constant $c_{ost}$. Then, for arbitrary $k \geq k_{ost}$, on the event of $\mathcal{E}_{\textbf{Bonus},k}, \mathcal{E}_{\textbf{Con},k}, \mathcal{E}_{\text{Alg}^{\text{Lo}},k}$, case 3 implies that:

$$V^*_{\text{Lo},h}(s_h) - \underline{V}^k_{\text{Lo},h}(s_h) \leq \frac{\Delta_{\min}}{4(H+1)}.$$

Combining with Lem. D.5, it directly implies that $\underline{\pi}^k_{\text{Lo},h}(s_h) = \pi^*_{\text{Lo},h}(s_h)$. Therefore, in the following, we directly use $\pi^*_{\text{Lo},h}$ to refer $\underline{\pi}^k_{\text{Lo},h}(s_h)$. Then first observation is that, under Cond. B, in this case, we have:

$$
\begin{aligned}
V^*_{\text{Hi},h}(s_h) - \widetilde{Q}^k_{\text{Hi},h}(s_h, \pi^*_{\text{Lo}}) \leq& V^*_{\text{Hi},h}(s_h) - \underline{Q}^k_{\text{Lo},h}(s_h, \pi^*_{\text{Lo},h}) + \varepsilon \\
=& V^*_{\text{Hi},h}(s_h) - \underline{V}^k_{\text{Lo},h}(s_h) + \varepsilon \\
\leq& V^*_{\text{Lo},h}(s_h) + \frac{\Delta_{\min}}{2(H+1)} - \underline{V}^k_{\text{Lo},h}(s_h) + \varepsilon \\
=& V^*_{\text{Lo},h}(s_h) - \underline{V}^k_{\text{Lo},h}(s_h) + \frac{\Delta_{\min}}{2(H+1)} + \varepsilon \\
\leq& \frac{\Delta_{\min}}{H+1} \leq \frac{1}{H+1}(V^*_{\text{Hi},h}(s_h) - Q^*_{\text{Hi},h}(s_h, \pi^*_{\text{Lo}}))
\end{aligned}
$$

which implies that:

$$
\begin{aligned}
V^*_{\text{Hi},h}(s_h) \leq& \widetilde{Q}^k_{\text{Hi},h}(s_h, \pi^*_{\text{Lo}}) + \frac{1}{H}(\widetilde{Q}^k_{\text{Hi},h}(s_h, \pi^*_{\text{Lo}}) - Q^*_{\text{Hi},h}(s_h, \pi^*_{\text{Lo}})) \\
\leq& \widetilde{Q}^k_{\text{Hi},h}(s_h, \pi^*_{\text{Lo}}) + \frac{1}{H}(\widetilde{Q}^k_{\text{Hi},h}(s_h, \pi^*_{\text{Lo}}) - \underline{Q}^{\pi^k_{\text{Hi}}}_{\text{Hi},h}(s_h, \pi^*_{\text{Lo}})) \qquad \text{(Lem. D.5)} \\
=& \widetilde{V}^k_{\text{Lo},h}(s_h)
\end{aligned}
$$

which finishes the proof for overestimation.

**Part 2: Proof for Eq.** (1)   The following proof relies on Lem. D.13 and Lem. D.14, whose proofs we just provide after finishing the proof for this theorem. The first observation is, for arbitrary policy $\pi_{\text{Hi}}$,

$$
\begin{aligned}
V^*_{\text{Hi},h}(s_h) - V^{\pi_{\text{Hi}}}_{\text{Hi},h}(s_h) =& Q^*_{\text{Hi},h}(s_h, \pi^*_{\text{Hi}}) - Q^*_{\text{Hi},h}(s_h, \pi_{\text{Hi}}) \\
=& \Delta_{\text{Hi}}(s_h, \pi_{\text{Hi}}) + Q^*_{\text{Hi},h}(s_h, \pi_{\text{Hi}}) - Q^*_{\text{Hi},h}(s_h, \pi_{\text{Hi}}) \\
=& \Delta_{\text{Hi}}(s_h, \pi_{\text{Hi}}) + \mathbb{P}_{\text{Hi},h}(V^*_{\text{Hi},h+1} - V^{\pi_{\text{Hi}}}_{\text{Hi},h+1})(s_h, \pi_{\text{Hi}}) \\
=& ... \\
=& \mathbb{E}_{\pi_{\text{Hi}}, M_{\text{Hi}}}[\sum_{h'=h}^{H} \Delta_{\text{Hi}}(s_{h'}, a_{h'})|s_h]. \qquad (22)
\end{aligned}
$$

Besides, according to the definition of $\ddot{\mathbf{E}}_{\text{Hi},h}$, we have:

$$\ddot{V}^{\pi^k_{\text{Hi}}}_{\text{Hi},h}(s_h) - V^{\pi^k_{\text{Hi}}}_{\text{Hi},h}(s_h) = e\mathbb{E}_{\pi^k_{\text{Hi}}}[\sum_{h'=h}^{H} \ddot{\mathbf{E}}^k_{\text{Hi}h'}(s_{h'}, a_{h'})|s_h]$$

$$\geq e\mathbb{E}_{\pi_{\text{Hi}}^k}\Big[\sum_{h'=h}^{H}\mathbf{E}_{\text{Hi}h'}^k(s_{h'},a_{h'})-\varepsilon_{\text{Clip}}-\frac{\Delta_{\text{Hi}}(s_{h'},a_{h'})}{4e}\Big|s_h\Big]$$

$$\geq e\mathbb{E}_{\pi_{\text{Hi}}^k}\Big[\sum_{h'=h}^{H}\mathbf{E}_{\text{Hi}h'}^k(s_{h'},a_{h'})-\frac{\Delta_{\text{Hi}}(s_{h'},a_{h'})}{4e}\Big|s_h\Big]-eH\cdot\frac{\Delta_{\min}}{4eH}.$$

$$\geq\widetilde{V}_{\text{Hi},h}^k(s_h)-V_{\text{Hi},h}^{\pi_{\text{Hi}}^k}(s_h)-\frac{1}{4}(V_{\text{Hi},h}^*(s_h)-V_{\text{Hi},h}^{\pi_{\text{Hi}}^k}(s_h))-\frac{\Delta_{\min}}{4}.$$
$$\text{(Eq. (22) and Lem. D.14)}$$

$$\geq\frac{3}{4}(V_{\text{Hi},h}^*(s_h)-V_{\text{Hi},h}^{\pi_{\text{Hi}}^k}(s_h))-\frac{\Delta_{\min}}{4}.\qquad\text{(Overestimation)}$$

If $\pi_{\text{Hi}}^k(s_h)\neq\pi_{\text{Hi}}^*(s_h)$, since $V_{\text{Hi},h}^*(s_h)-V_{\text{Hi},h}^{\pi_{\text{Hi}}^k}(s_h)\geq\Delta_{\text{Hi}}(s_h,\pi_{\text{Hi}}^k)$, we further have:

$$\ddot{V}_{\text{Hi},h}^{\pi_{\text{Hi}}^k}(s_h)-V_{\text{Hi},h}^{\pi_{\text{Hi}}^k}(s_h)\geq\frac{1}{2}(V_{\text{Hi},h}^*(s_h)-V_{\text{Hi},h}^{\pi_{\text{Hi}}^k}(s_h))+\frac{\Delta_{\text{Hi}}(s_h,\pi_{\text{Hi}}^k)}{4}-\frac{\Delta_{\min}}{4}\geq\frac{1}{2}(V_{\text{Hi},h}^*(s_h)-V_{\text{Hi},h}^{\pi_{\text{Hi}}^k}(s_h)).$$

otherwise,

$$\ddot{V}_{\text{Hi},h}^{\pi_{\text{Hi}}^k}(s_h)-V_{\text{Hi},h}^{\pi_{\text{Hi}}^k}(s_h)=\ddot{\mathbf{E}}_{\text{Hi},h}(s_h,\pi_{\text{Hi}}^k)+\mathbb{E}_{\pi_{\text{Hi}}^k,M_{\text{Hi}}}[\ddot{V}_{\text{Hi},h+1}^{\pi_{\text{Hi}}^k}(s_{h+1})-V_{\text{Hi},h+1}^{\pi_{\text{Hi}}^k}(s_{h+1})|s_h]$$

$$\geq\mathbb{E}_{\pi_{\text{Hi}}^k,M_{\text{Hi}}}[\ddot{V}_{\text{Hi},h+1}^{\pi_{\text{Hi}}^k}(s_{h+1})-V_{\text{Hi},h+1}^{\pi_{\text{Hi}}^k}(s_{h+1})|s_h].$$

Therefore, we have:

$$\ddot{V}_{\text{Hi},1}^{\pi_{\text{Hi}}^k}(s_1)-V_{\text{Hi},1}^{\pi_{\text{Hi}}^k}(s_1)\geq\mathbb{E}_{\pi_{\text{Hi}}^k,M_{\text{Hi}}}\Big[\sum_{h=1}^{H}\mathbb{I}[\widetilde{\mathcal{E}}_{\text{Hi},h}^k](\ddot{V}_{\text{Hi},h}^{\pi_{\text{Hi}}^k}(s_h)-V_{\text{Hi},h}^{\pi_{\text{Hi}}^k}(s_h))\Big]$$

$$\geq\frac{1}{2}\mathbb{E}_{\pi_{\text{Hi}}^k,M_{\text{Hi}}}\Big[\sum_{h=1}^{H}\mathbb{I}[\widetilde{\mathcal{E}}_{\text{Hi},h}^k](V_{\text{Hi},h}^*(s_h)-V_{\text{Hi},h}^{\pi_{\text{Hi}}^k}(s_h))\Big]$$

$$=\frac{1}{2}(V_{\text{Hi},h}^*(s_1)-V_{\text{Hi},h}^{\pi_{\text{Hi}}^k}(s_1)).$$

Combining Lem. D.13 and the definition of $\ddot{V}_{\text{Hi},1}^{\pi_{\text{Hi}}^k}(s_1)$, we can finish the proof for Eq. (1). $\qquad\square$

**Lemma D.13** (Upper and lower bounds of the surplus). *For arbitrary $k$, on the event of $\mathcal{E}_{\textbf{Bonus},k}$, we have:*

$$\forall h\in[H],\ \forall s_h\in\mathcal{S}_h,a_h\in\mathcal{A}_h,\quad 0\leq\mathbf{E}_{Hi,h}^k(s_h,a_h)\leq\min\{H,4b_{Hi,h}^k(s_h,a_h)\}$$

*Proof.* According to Alg. 2, we should have $\widetilde{Q}_{\text{Hi},h}^k,\ \widetilde{V}_{\text{Hi},h+1}^k,\ \underline{V}_{\text{Hi},h+1}^{\pi_{\text{Hi}}^k},\ \underline{Q}_{\text{Hi},h}^{\pi_{\text{Hi}}^k}\in[0,H]$. By Lem. D.5 and Thm. 4.3, we also have $\widetilde{V}_{\text{Hi},h+1}^k(\cdot)\geq\underline{V}_{\text{Hi},h+1}^{\pi_{\text{Hi}}^k}(\cdot)$, which implies that $\mathbf{E}_{\text{Hi},h}^k(\cdot,\cdot)\leq H$. Besides,

$$\mathbf{E}_{\text{Hi},h}^k(s_h,a_h)=(\widehat{\mathbb{P}}_{\text{Hi},h}^k-\mathbb{P}_{\text{Hi},h})\widetilde{V}_{\text{Hi},h+1}^k(s_h,a_h)+(\mathbb{P}_{\text{Hi},h}-\widehat{\mathbb{P}}_{\text{Hi},h}^k)\underline{V}_{\text{Hi},h+1}^{\pi_{\text{Hi}}^k}(s_h,a_h)+2b_{\text{Hi},h}^k(s_h,a_h).$$

On the event of $\mathcal{E}_{\textbf{Bonus},k}$, we have $0\leq\mathbf{E}_{\text{Hi},h}^k(s_h,a_h)\leq4b_{\text{Hi},h}^k(s_h,a_h)$, which finishes the proof. $\quad\square$

**Lemma D.14** (Relationship between surplus and overestimation gap). *Under the same condition of Thm. 4.3,*

$$\widetilde{V}_{Hi,h}^k(s_h)-V_{Hi,h}^{\pi_{Hi}^k}(s_h)\leq e\mathbb{E}_{\pi_{Hi}^k,M_{Hi}}\Big[\sum_{h'=h}^{H}\mathbf{E}_{Hi,h}^k(s_h,\pi_{Hi}^k)|s_h\Big].$$

*Proof.*

$$\widetilde{V}_{\text{Hi},h}^k(s_h)-V_{\text{Hi},h}^{\pi_{\text{Hi}}^k}(s_h)\leq\widetilde{V}_{\text{Hi},h}^k(s_h)-\underline{V}_{\text{Hi},h}^{\pi_{\text{Hi},k}}(s_h)\qquad\text{(Lem. D.5)}$$

$$\leq(1+\frac{1}{H})(\widetilde{Q}_{\text{Hi},h}^k(s_h,\pi_{\text{Hi}}^k)-\underline{Q}_{\text{Hi},h}^{\pi_{\text{Hi},k}}(s_h,\pi_{\text{Hi}}^k))\qquad\text{(Update rule in Alg. 2 and }\widetilde{Q}_{\text{Hi},h}^k\geq\underline{Q}_{\text{Hi},h}^{\pi_{\text{Hi},k}})$$

$$=(1+\frac{1}{H})(\widetilde{Q}^k_{\text{Hi},h}(s_h,\pi^k_{\text{Hi}}) - \mathbb{P}_{\text{Hi},h}\widetilde{V}^k_{\text{Hi},h+1}(s_h,\pi^k_{\text{Hi}}) + \mathbb{P}_{\text{Hi},h}\underline{V}^k_{\text{Hi},h+1}(s_h,\pi^k_{\text{Hi}}) - \underline{Q}^{\pi_{\text{Hi},k}}_{\text{Hi},h}(s_h,\pi^k_{\text{Hi}}))$$

$$+(1+\frac{1}{H})\mathbb{P}_{\text{Hi},h}(\widetilde{V}^k_{\text{Hi},h+1} - \underline{V}^k_{\text{Hi},h+1})(s_h,\pi^k_{\text{Hi}})$$

$$=(1+\frac{1}{H})\mathbf{E}^k_{\text{Hi},h}(s_h,\pi^k_{\text{Hi}}) + (1+\frac{1}{H})\mathbb{P}_{\text{Hi},h}(\widetilde{V}^k_{\text{Hi},h+1} - \underline{V}^k_{\text{Hi},h+1})(s_h,\pi^k_{\text{Hi}})$$

$$\leq ...$$

$$\leq \mathbb{E}_{\pi^k_{\text{Hi}},M_{\text{Hi}}}[\sum_{h'=h}^{H}(1+\frac{1}{H})^{h'-h+1}\mathbf{E}^k_{\text{Hi},h}(s_h,\pi^k_{\text{Hi}})]$$

$$\leq e\mathbb{E}_{\pi^k_{\text{Hi}},M_{\text{Hi}}}[\sum_{h'=h}^{H}\mathbf{E}^k_{\text{Hi},h}(s_h,\pi^k_{\text{Hi}})].$$

$\square$

In the following lemma, we show the benefits of transfer action between similar states.

**Lemma D.15** (Benefits on Similar States). *If $s_h$ in $M_{Hi}$ is $\varepsilon$-close to $s_h$ in $M_{Lo}$, i.e. satisfying the property in Def. 2.1, and $d^*_{Lo}(s_h) \geq \lambda$, where $\lambda$ is the hyper-parameter in Alg. 2, then, for arbitrary $k \geq k_{occup} := c_{occup}\frac{C_1+\alpha C_2}{\lambda\Delta_{\min}}\log(\frac{\alpha C_1 C_2 SAH}{\lambda\Delta_{\min}})$, on the events of $\mathcal{E}_{Alg^{Lo},k}, \mathcal{E}_{Con,k}, \mathcal{E}_{Bonus,k}$, we have $\pi^k_{Hi}(s_h) = \pi^*_{Hi}(s_h)$.*

*Proof.* As a result of Lem. D.10, we have, for arbitrary $s_h$ with $d^*_{\text{Lo}}(s_h) = d^*_{\text{Lo}}(s_h,\pi^*_{\text{Lo}}) \geq \lambda$, after $k \geq c_{occup}\frac{C_1+\alpha C_2}{\lambda\Delta_{\min}}\log(\frac{\alpha C_1 C_2 SAH}{\lambda\Delta_{\min}})$, we should have:

$$N^k_{\text{Lo},h}(s_h) \geq N^k_{\text{Lo},h}(s_h,\pi^*_{\text{Lo}}) \geq \frac{\lambda}{3}k.$$

On the events of $\mathcal{E}_{\textbf{Bonus},k}$, and the value dominance condition Def. B, we also have:

$$\underline{Q}^k_{\text{Lo},h}(s_h,\underline{\pi}^k_{\text{Lo},h}) \leq \widetilde{Q}^k_{\text{Hi},h}(s_h,\underline{\pi}^k_{\text{Lo},h}) + \varepsilon.$$

which implies that the algorithm will choose the "trust" branch and choose $\pi^k_{\text{Hi},h}(s_h) = \underline{\pi}^k_{\text{Lo},h}(s_h)$.

On the other hand, if there is another $a_h \neq \pi^*_{\text{Lo}}(s_h)$ satisfying $N^k_{\text{Lo},h}(s_h,a_h) \geq \frac{\lambda}{3}k$, by applying Lem. D.10 again, we can make a contradication and therefore, we must have

$$N^k_{\text{Lo},h}(s_h,\pi^*_{\text{Lo}}) \geq \frac{\lambda}{3}k > N^k_{\text{Lo},h}(s_h,a_h), \quad \forall a_h \neq \pi^*_{\text{Lo}}(s_h).$$

which implies that $\pi^k_{\text{Hi}}(s_h) = \pi^*_{\text{Lo}}(s_h)$ in Alg. 2. Given that $s_h$ is $\varepsilon$-close between $M_{\text{Hi}}$ and $M_{\text{Lo}}$, we directly have $\pi^k_{\text{Hi}}(s_h) = \pi^*_{\text{Hi}}(s_h)$. $\square$

**Theorem D.16** (Detailed Version of Thm.4.2). *Under Assump. A, B and C, Cond. D.1 for $Alg^{Lo}$ and Cond. D.3 for **Bonus** function, by running Alg. 2 with $\varepsilon = \frac{\Delta_{\min}}{4(H+1)}$, $\alpha > 2$, an any $\lambda > 0$, we have*

$$Regret_K(M_{Hi}) = O\Big(H \cdot \max\{\alpha\frac{S^3 H^4}{\lambda^2\Delta^2_{\min}}\log(\alpha SAH), \frac{(C_1+\alpha C_2)SH}{\lambda\Delta^2_{\min}}\log\frac{C_1 C_2 SAH}{\Delta_{\min}\lambda}\}$$

$$+\sum_{h=1}^{H}\sum_{(s_h,a_h)\in\mathcal{C}^*_h}\frac{SH^2}{\Delta_{\min}}\log(SAH(K\wedge\frac{1}{\lambda\Delta_{\min}d^*_{Hi}(s_h)}))$$

$$+SH\sum_{h=1}^{H}\sum_{(s_h,a_h)\in\mathcal{S}_h\times\mathcal{A}_h\backslash\mathcal{C}^\lambda_h}(\frac{H}{\Delta_{\min}}\wedge\frac{1}{\Delta_{Hi}(s_h,a_h)})\log(SAHK)\Big)$$

$$=O\Big(SH\sum_{h=1}^{H}\sum_{(s_h,a_h)\in\mathcal{S}_h\times\mathcal{A}_h\backslash\mathcal{C}^\lambda_h}(\frac{H}{\Delta_{\min}}\wedge\frac{1}{\Delta_{Hi}(s_h,a_h)})\log(SAHK)\Big). \quad (23)$$

*Proof.* We consider $k_{start} := \max\{k_{ost}, k_{occup}\}$. We first study the regret part after $k \geq k_{start}$:

$$\mathbb{E}[\sum_{k=k_{start}+1}^{K} V_{\text{Hi},1}^{*}(s_1) - V_{\text{Hi},1}^{\pi_{\text{Hi}}^k}(s_1)]$$

$$\leq \sum_{k=k_{start}+1}^{K} 2e\mathbb{E}_{\pi_{\text{Hi}}^k}[\sum_{h=1}^{H} \text{Clip}\left[\min\{H, 4b_{\text{Hi},h}^k(s_h, a_h)\}\Big|\frac{\Delta_{\min}}{4eH} \vee \frac{\Delta_{\text{Hi}}(s_h, a_h)}{4e}\right] \mathbb{I}[\mathcal{E}_{\text{Bonus},k} \cap \mathcal{E}_{\text{Alg}^{\text{Lo}},k} \cap \mathcal{E}_{\text{Con},k}]]$$

$$+ \sum_{k=k_{start}+1}^{K} H \cdot \Pr(\mathcal{E}_{\text{Bonus},k}^{\complement} \cup \mathcal{E}_{\text{Alg}^{\text{Lo}},k}^{\complement} \cup \mathcal{E}_{\text{Con},k}^{\complement}).$$

Since the failure rate for events $\mathcal{E}_{\text{Bonus},k}, \mathcal{E}_{\text{Alg}^{\text{Lo}},k}, \mathcal{E}_{\text{Con},k}$ is only at the level of $k^{-\Theta(\alpha)}$, the second part is constant, and we mainly focus on the first term. For all state action $(s_h, a_h)$, and for all $k \geq k_{start}$, we have:

$$\mathbb{E}_{\pi_{\text{Hi}}^k}[\text{Clip}\left[\min\{H, 4b_{\text{Hi},h}^k(s_h, a_h)\}\Big|\frac{\Delta_{\min}}{4eH} \vee \frac{\Delta_{\text{Hi}}(s_h, a_h)}{4e}\right] \mathbb{I}[\mathcal{E}_{\text{Bonus},k} \cap \mathcal{E}_{\text{Alg}^{\text{Lo}},k} \cap \mathcal{E}_{\text{Con},k}]]$$

$$=d_{\text{Hi}}^{\pi_{\text{Hi}}^k}(s_h, a_h)\text{Clip}\left[\min\{H, 4B_1\sqrt{\frac{\log(B_2 k^{\alpha})}{N_{\text{Hi},h}^k(s_h, a_h)}}\}\Big|\frac{\Delta_{\min}}{4eH} \vee \frac{\Delta_{\text{Hi}}(s_h, a_h)}{4e}\right]$$

$$\leq d_{\text{Hi}}^{\pi_{\text{Hi}}^k}(s_h, a_h)\text{Clip}\left[\min\{H, 4B_1\sqrt{\frac{\alpha\log(B_2 K)}{N_{\text{Hi},h}^k(s_h, a_h)}}\}\Big|\frac{\Delta_{\min}}{4eH} \vee \frac{\Delta_{\text{Hi}}(s_h, a_h)}{4e}\right] \quad (24)$$

Under the event of $\mathcal{E}_{\text{Con},k}$, as a result of Lem. D.6, we have:

$$N_{\text{Hi},h}^k(s_h, a_h) \geq \frac{1}{2}\sum_{k'=1}^{k-1} d_{\text{Hi}}^{\pi_{\text{Hi}}^{k'}}(s_h, a_h) - \alpha\log(2SAHk) \geq \frac{1}{2}\sum_{k'=1}^{k-1} d_{\text{Hi}}^{\pi_{\text{Hi}}^{k'}}(s_h, a_h) - \alpha\log(2SAHK)$$

We denote $\tau_{s_h, a_h}^K := \min_k \ s.t. \ \forall k' \geq k, \ \frac{1}{4}\sum_{k'=1}^{k-1} d_{\text{Hi}}^{\pi_{\text{Hi}}^k}(s_h, a_h) \geq \alpha\log(2SAHK)$. Then we have:

$$(24) \leq d_{\text{Hi}}^{\pi_{\text{Hi}}^k}(s_h, a_h)\text{Clip}\left[\min\{H, 8B_1\sqrt{\frac{\alpha\log(B_2 K)}{\sum_{k'=1}^{k-1} d_{\text{Hi}}^{\pi_{\text{Hi}}^k}(s_h, a_h)}}\}\Big|\frac{\Delta_{\min}}{4eH} \vee \frac{\Delta_{\text{Hi}}(s_h, a_h)}{4e}\right], \quad \forall k \geq \tau_{s_h, a_h}^K.$$

Therefore, for arbitrary $s_h, a_H$, there exists an absolute constant $c_{s_h, a_H}$, such that:

$$\sum_{k=k_{start}+1}^{K} \mathbb{E}_{\pi_{\text{Hi}}^k}[\text{Clip}\left[\min\{H, 4b_{\text{Hi},h}^k(s_h, a_h)\}\Big|\frac{\Delta_{\min}}{4eH} \vee \frac{\Delta_{\text{Hi}}(s_h, a_h)}{4e}\right]]$$

$$\leq H \cdot \sum_{k=1}^{\tau_{s_h, a_h}^K} d_{\text{Hi}}^{\pi_{\text{Hi}}^k}(s_h, a_h) + \sum_{k=\tau_{s_h, a_h}^K+1}^{K} \mathbb{E}_{\pi_{\text{Hi}}^k}[\text{Clip}\left[\min\{H, 4B_1\sqrt{\frac{\alpha\log(B_2 K)}{N_{\text{Hi},h}^k(s_h, a_h)}}\Big|\frac{\Delta_{\min}}{4eH} \vee \frac{\Delta_{\text{Hi}}(s_h, a_h)}{4e}\right]]$$

$$\leq c_{s_h, a_h} H\log(2SAHK)) + \sum_{k=\tau_{s_h, a_h}^K+1}^{K} d_{\text{Hi}}^{\pi_{\text{Hi}}^k}(s_h, a_h)\text{Clip}\left[\min\{H, 8B_1\sqrt{\frac{\alpha\log(B_2 K)}{\sum_{k'=1}^{k-1} d_{\text{Hi}}^{\pi_{\text{Hi}}^k}(s_h, a_h)}}\}\Big|\frac{\Delta_{\min}}{4eH} \vee \frac{\Delta_{\text{Hi}}(s_h, a_h)}{4e}\right]$$

$$\leq c_{s_h, a_h} H\log(2SAHK) + c_{s_h, a_h} \cdot \int_{\alpha\log(2SAHK)}^{K/4} \text{Clip}\left[B_1\sqrt{\frac{\alpha\log(B_2 K)}{x}}\Big|\frac{\Delta_{\min}}{4eH} \vee \frac{\Delta_{\text{Hi}}(s_h, a_h)}{4e}\right] dx$$

$$=O\left(H\log(2SAHK) + B_1(\frac{H}{\Delta_{\min}} \wedge \frac{1}{\Delta_{\text{Hi}}(s_h, a_h)})\log(B_2 K)\right).$$

As a result, we can establish the following regret upper bound (note that $k_{start} = O(k_{ost})$):

$$\mathbb{E}[\sum_{k=1}^{K} V_{\text{Hi},1}^{*}(s_1) - V_{\text{Hi},1}^{\pi_{\text{Hi}}^k}(s_1)]$$

$$\leq \sum_{k=k_{start}+1}^{K} 2e\mathbb{E}_{\pi_{\text{Hi}}^k}[\sum_{h=1}^{H} \text{Clip}\left[\min\{H, 4b_{\text{Hi},h}^k(s_h, a_h)\}\Big| \frac{\Delta_{\min}}{4eH} \vee \frac{\Delta_{\text{Hi}}(s_h, a_h)}{4e}\right] \mathbb{I}[\mathcal{E}_{\textbf{Bonus},k} \cap \mathcal{E}_{\text{Alg}^{\text{Lo}},k} \cap \mathcal{E}_{\textbf{Con},k}]]$$

$$+ Hk_{start} + \sum_{k=k_{start}+1}^{K} H \cdot \Pr(\mathcal{E}_{\textbf{Bonus},k}^{\complement} \cup \mathcal{E}_{\text{Alg}^{\text{Lo}},k}^{\complement} \cup \mathcal{E}_{\textbf{Con},k}^{\complement})$$

$$= O\left( k_{start} \cdot H + SAH^2 \log(2SAHK) + B_1 \log(B_2 K) \sum_{h=1}^{H} \sum_{s_h, a_h} \left(\frac{H}{\Delta_{\min}} \wedge \frac{1}{\Delta_{\text{Hi}}(s_h, a_h)}\right)\right).$$

$$(25)$$

As introduced in maintext, because of the knowledge transfer, we may expect to achieve constant regret on some special state action pairs, and we analyze them in the following.

**Type 1:** $(s_h, a_h) \in \mathcal{C}_h^{\lambda,1} \cup \mathcal{C}_h^{\lambda,2}$**: Constant Regret because of Low Visitation Probability** As discussed in Lem. D.15, for $s_h \in \mathcal{Z}_h^{\varepsilon,\lambda}$, since $k_{start} \geq k_{occup} := c_{occup}\frac{C_1 + \alpha C_2}{\lambda \Delta_{\min}} \log(\frac{\alpha C_1 C_2 SAH}{\lambda \Delta_{\min}})$, on the event of $\mathcal{E}_{\text{Alg}^{\text{Lo}},k}, \mathcal{E}_{\textbf{Bonus},k}$ and $\mathcal{E}_{\textbf{Con},k}$, we have $\forall a_h \neq \pi_{\text{Hi}}^*(s_h)$, $d_{\text{Hi}}^{\pi_{\text{Hi}}^k}(s_h, a_h) = 0$. Moreover, according to the definition of $\mathcal{C}_h^{\lambda,2}$, we also have $\forall (s_h, a_h) \in \mathcal{C}_h^{\lambda,2}$, $d_{\text{Hi}}^{\pi_{\text{Hi}}^k}(s_h, a_h) = 0$. Therefore, for all $h \in [H]$ and $(s_h, a_h) \in \mathcal{C}_h^{\lambda,1} \cup \mathcal{C}_h^{\lambda,2}$:

$$\sum_{k=k_{start}+1}^{K} \mathbb{E}_{\pi_{\text{Hi}}^k}[\text{Clip}\left[\min\{H, 4b_{\text{Hi},h}^k(s_h, a_h)\}\Big| \frac{\Delta_{\min}}{4eH} \vee \frac{\Delta_{\text{Hi}}(s_h, a_h)}{4e}\right] \mathbb{I}[\mathcal{E}_{\textbf{Bonus},k} \cap \mathcal{E}_{\text{Alg}^{\text{Lo}},k} \cap \mathcal{E}_{\textbf{Con},k}]] = 0.$$

**Type 2:** $(s_h, a_h) \in \mathcal{C}_h^*$**, i.e.** $d_{\textbf{Hi}}^*(s_h, a_h) = d_{\textbf{Hi}}^*(s_h) > 0$ Because of the sub-linear regret in Eq. (25), we may expect that $N_{\text{Hi},h}^k(s_h, a_h) \approx \sum_{k=1}^{K} d_{\text{Hi}}^{\pi_{\text{Hi}}^k}(s_h, a_h) \sim O(Kd_{\text{Hi}}^*(s_h, a_h))$ when $K$ is large enough. To see this, note that $\sum_{\widetilde{k}=1}^{k}(V_{\text{Hi},1}^*(s_1) - V_{\text{Hi},1}^{\pi_{\text{Hi}}^{\widetilde{k}}}(s_1)) - \mathbb{E}[\sum_{\widetilde{k}=1}^{k} V_{\text{Hi},1}^*(s_1) - V_{\text{Hi},1}^{\pi_{\text{Hi}}^{\widetilde{k}}}(s_1)]$ is a martingale difference sequence with bounded difference. We define $\mathcal{E}_{\text{Alg}^{\text{Hi}},k} := \{\sum_{\widetilde{k}=1}^{k} V_{\text{Hi},1}^*(s_1) - V_{\text{Hi},1}^{\pi_{\text{Hi}}^{\widetilde{k}}}(s_1) \geq H\sqrt{2\alpha k \log k} + \mathbb{E}[\sum_{\widetilde{k}=1}^{k} V_{\text{Hi},1}^*(s_1) - V_{\text{Hi},1}^{\pi_{\text{Hi}}^{\widetilde{k}}}(s_1)]\}$, according to the Azuma-Hoeffding inequality, we have:

$$\Pr(\mathcal{E}_{\text{Alg}^{\text{Hi}},k}) \leq \exp(\frac{-2\alpha H^2 k \log k}{2kH^2}) \leq \frac{1}{k^\alpha}.$$

On the event of $\mathcal{E}_{\textbf{Bonus},k}, \mathcal{E}_{\text{Alg}^{\text{Lo}},k}, \mathcal{E}_{\textbf{Con},k}$ and $\mathcal{E}_{\text{Alg}^{\text{Hi}},k}$, as a result of Thm. D.6, we have:

$$|\sum_{\widetilde{k}=1}^{k} d^{\pi_{\text{Hi}}^{\widetilde{k}}}(s_h, a_h) - kd_{\text{Hi}}^*(s_h, a_h)| \leq H\sqrt{2\alpha k \log k} + (25)$$

To make sure $\sum_{\widetilde{k}=1}^{k} d^{\pi_{\text{Hi}}^{\widetilde{k}}}(s_h, a_h) \geq \frac{k}{2}d_{\text{Hi}}^*(s_h, a_h)$, we expect:

$$\frac{k}{2}d_{\text{Hi}}^*(s_h, a_h) \geq H\sqrt{2\alpha k \log k} + (25)$$

which can be satisfied by:

$$k \geq \bar{\tau}_{s_h} := \bar{c}_{s_h}^* \frac{1}{(d_{\text{Hi}}^*(s_h))^2}\text{Poly}(S, A, H, \lambda^{-1}, \Delta_{\min}^{-1})$$

for some constant $\bar{c}_{s_h}^*$. As a result, for $k \geq \max\{\tau_{s_h,a_h}^K, \bar{\tau}_{s_h}\} + 1$, we should have:

$$\text{Clip}\left[\min\{H, 4b_{\text{Hi},h}^k(s_h, a_h)\}\Big| \frac{\Delta_{\min}}{4eH} \vee \frac{\Delta_{\text{Hi}}(s_h, a_h)}{4e}\right]$$

$$\leq \text{Clip}\left[\min\{H, 8B_1\sqrt{\frac{\alpha \log(kB_2)}{\sum_{k'=1}^{k-1} d_{\text{Hi}}^{\pi_{\text{Hi}}^k}(s_h, a_h)}}\}\Big| \frac{\Delta_{\min}}{4eH}\right] \leq \text{Clip}\left[8B_1\sqrt{\frac{2\alpha \log(kB_2)}{kd_{\text{Hi}}^*(s_h, a_h)}}\Big| \frac{\Delta_{\min}}{4eH}\right].$$

$$(\Delta_{\text{Hi}}(s_h, a_h) = 0)$$

Note that $8B_1\sqrt{\frac{2\alpha\log(kB_2)}{kd_{\mathrm{Hi}}^*(s_h,a_h)}} = 8B_1\sqrt{\frac{2\alpha\log(kB_2)}{kd_{\mathrm{Hi}}^*(s_h)}} \le \frac{\Delta_{\min}}{4eH}$ can be satisfied when $k \ge \tau'_{s_h} := c'_{s_h}\frac{\alpha H^2 B_1^2}{\Delta_{\min}^2 d_{\mathrm{Hi}}^*(s_h)}\log(\frac{\alpha B_1 B_2 H}{\Delta_{\min} d_{\mathrm{Hi}}^*(s_h)})$ for some absolute constant $c'_{s_h}$. Therefore, $\min\{H, 4b_{\mathrm{Hi},h}^k(s_h,a_h)\} \le \frac{\Delta_{\min}}{4eH}$ and the regret will not increase after $k \ge \tau_{s_h,a_h}^* := c_{s_h,a_h}^* \max\{k_{start}, \tau_{s_h,a_h}, \bar\tau_{s_h}, \tau'_{s_h}\} = \mathrm{Poly}(S,A,H,\lambda^{-1},\Delta_{\min}^{-1},(d_{\mathrm{Hi}}^*(s_h))^{-1})$, for some absolute constant $c_{s_h,a_h}^*$, which implies that,

$$\sum_{k=k_{start}+1}^{K} \mathbb{E}_{\pi_{\mathrm{Hi}}^k}\left[\mathrm{Clip}\left[\min\{H, 4b_{\mathrm{Hi},h}^k(s_h,a_h)\}\Big|\frac{\Delta_{\min}}{4eH}\vee\frac{\Delta_{\mathrm{Hi}}(s_h,a_h)}{4e}\right]\mathbb{I}[\mathcal{E}_{\mathbf{Bonus},k}\cap\mathcal{E}_{\mathrm{Alg^{Lo}},k}\cap\mathcal{E}_{\mathbf{Con},k}]\right]$$

$$=\sum_{k=k_{start}}^{\tau_{s_h,a_h}^*} \mathbb{E}_{\pi_{\mathrm{Hi}}^k}\left[\mathrm{Clip}\left[\min\{H, 4b_{\mathrm{Hi},h}^k(s_h,a_h)\}\Big|\frac{\Delta_{\min}}{4eH}\vee\frac{\Delta_{\mathrm{Hi}}(s_h,a_h)}{4e}\right]\mathbb{I}[\mathcal{E}_{\mathbf{Bonus},k}\cap\mathcal{E}_{\mathrm{Alg^{Lo}},k}\cap\mathcal{E}_{\mathbf{Con},k}]\right]$$

$$=O\left(\frac{HB_1}{\Delta_{\min}}\log(SAHB_2\min\{K,\frac{1}{\lambda\Delta_{\min}d_{\mathrm{Hi}}^*(s_h)}\})\right).$$

where in last step, we use the fact that for $\Delta_{\mathrm{Hi}}(s_h,a_h)=0$:

$$O\left(H\log(2SAHK) + B_1\log(B_2K)\cdot\left(\frac{H}{\Delta_{\min}}\wedge\frac{1}{\Delta_{\mathrm{Hi}}(s_h,a_h)}\right)\right) = O\left(\frac{HB_1}{\Delta_{\min}}\log(SAHB_2K)\right).$$

As a summary, recall $k_{start} = \max\{k_{ost}, k_{occup}\}$, we have:

$$\mathrm{Regret}_K(M_{\mathrm{Hi}}) = O\Big(H\cdot\max\{\alpha\frac{B_1^2 H^2 S}{\lambda^2\Delta_{\min}^2}\log(\alpha HSAB_1B_2),\frac{(C_1+\alpha C_2)SH}{\lambda\Delta_{\min}^2}\log\frac{C_1C_2SAH}{\Delta_{\min}\lambda}\}$$

$$+\sum_{h=1}^{H}\sum_{(s_h,a_h)\in\mathcal{C}_h^*}\frac{HB_1}{\Delta_{\min}}\log(SAHB_2(K\wedge\frac{1}{\lambda\Delta_{\min}d_{\mathrm{Hi}}^*(s_h)}))$$

$$+\sum_{h=1}^{H}\sum_{(s_h,a_h)\in\mathcal{S}_h\times\mathcal{A}_h\backslash\mathcal{C}_h^\lambda}H\log(2SAHK) + B_1(\frac{H}{\Delta_{\min}}\wedge\frac{1}{\Delta_{\mathrm{Hi}}(s_h,a_h)})\log(B_2K)\Big).$$

By considering $B_1 = O(SH)$ and $B_2 = O(SA)$ in Example D.4, and omitting all the constant terms independent w.r.t. $K$, we can rewrite the above upper bound to:

$$\mathrm{Regret}_K(M_{\mathrm{Hi}}) = O\Big(H\cdot\max\{\alpha\frac{S^3 H^4}{\lambda^2\Delta_{\min}^2}\log(\alpha SAH),\frac{(C_1+\alpha C_2)SH}{\lambda\Delta_{\min}^2}\log\frac{C_1C_2SAH}{\Delta_{\min}\lambda}\}$$

$$+\sum_{h=1}^{H}\sum_{(s_h,a_h)\in\mathcal{C}_h^*}\frac{SH^2}{\Delta_{\min}}\log(SAH(K\wedge\frac{1}{\lambda\Delta_{\min}d_{\mathrm{Hi}}^*(s_h)}))$$

$$+SH\sum_{h=1}^{H}\sum_{(s_h,a_h)\in\mathcal{S}_h\times\mathcal{A}_h\backslash\mathcal{C}_h^\lambda}(\frac{H}{\Delta_{\min}}\wedge\frac{1}{\Delta_{\mathrm{Hi}}(s_h,a_h)})\log(SAHK)\Big)$$

$$=O\Big(SH\sum_{h=1}^{H}\sum_{(s_h,a_h)\in\mathcal{S}_h\times\mathcal{A}_h\backslash\mathcal{C}_h^\lambda}(\frac{H}{\Delta_{\min}}\wedge\frac{1}{\Delta_{\mathrm{Hi}}(s_h,a_h)})\log(SAHK)\Big).$$

$\square$

# E    Proofs for Tiered MAB with Multiple Source/Low-Tier Tasks

**Lemma 5.3.** *[Absorbing to Similar Task] Under Assump. A, B and C, there exists a constant $c^*$, s.t., if there exists at least one $w^* \in [W]$ such that $M_{Lo,w^*}$ is $\frac{\widetilde\Delta_{\min}}{4}$-close to $M_{Hi}$, by running Alg. 3 with $\varepsilon = \frac{\widetilde\Delta_{\min}}{4}$ and $\alpha > 2$, for any $k \ge k^* := c^*\frac{\alpha A}{\Delta_{\min}^2}\log\frac{\alpha AW}{\Delta_{\min}}$, we have $\Pr(\pi_{Hi}^k \ne i_{Hi}^*) = O(\frac{A}{k^{2\alpha-2}})$.*

*Proof.* In the following, we denote $\mathcal{W}^* := \{w \in [W] | i^*_{\text{Lo},w} = i^*_{\text{Hi}}, \ \mu_{\text{Lo}}(i^*_{\text{Hi},w^*}) \leq \mu_{\text{Hi}}(i^*_{\text{Hi}}) + \frac{\widetilde{\Delta}_{\min}}{4}\}$. $\widetilde{\mathcal{W}}^* := \{w \in [W] | i^*_{\text{Lo},w} = i^*_{\text{Hi}}\}$. In another word, $\mathcal{W}^*$ includes all transferable tasks, while $\widetilde{\mathcal{W}}^*$ includes all the tasks which share the optimal action $i^*_{\text{Hi}}$ regardless of whether the value function are close enough or not.

Consider the event $\mathcal{E} := \{\exists k' \in [\frac{k}{2}, k], \ s.t. \ \mathcal{W}^* \cap \mathcal{I}^{k'} = \emptyset\} \cup \{\exists k' \in [\frac{k}{2}, k], \ \exists w \in [W], \ s.t. \ \underline{\pi}^{k'}_{\text{Lo},w} \neq i^*_{\text{Hi}}\}$, note that on its complement: $\mathcal{E}^{\complement} := \{\forall k' \in [\frac{k}{2}, k], \ \mathcal{W}^* \cap \mathcal{I}^{k'} \neq \emptyset\} \cap \{\forall k' \in [\frac{k}{2}, k], \ \forall w \in [W], \ \underline{\pi}^{k'}_{\text{Lo},w} = i^*_{\text{Lo},w}\}$, if $\pi^k_{\text{Hi}} \neq i^*_{\text{Hi}}$ still happens, we must have: $\{\forall k' \in [\frac{k}{2}, k], \ \exists w \notin \widetilde{\mathcal{W}}^*, \ w \in \mathcal{I}^{k'}, \ w^{k'} = w\}$. That's because, if $\mathcal{E}^{\complement}$ holds, and $\mathcal{I}^{k'} \subset \widetilde{\mathcal{W}}^*$ for some $k'$, no matter which task in $\widetilde{\mathcal{W}}^*$ is chosen as $w^{k'}$, for all $\widetilde{k} \in [k', k]$, no matter whether $w^{\widetilde{k}}$ changes or not, the action we transfer is always $i^*_{\text{Hi}}$ (i.e. $\pi^{\widetilde{k}}_{\text{Hi}} = i^*_{\text{Hi}}$), because of the action inheritance startegy. Therefore,

$$\Pr(\pi^k_{\text{Hi}} \neq i^*_{\text{Hi}}) \leq \Pr(\mathcal{E}) + \Pr(\mathcal{E}^{\complement} \cap \{\pi^k_{\text{Hi}} \neq i^*_{\text{Hi}}\})$$

$$\leq \Pr(\mathcal{E}) + \Pr(\mathcal{E}^{\complement} \cap \{\forall k' \in [\frac{k}{2}, k], \ \exists w \notin \widetilde{\mathcal{W}}^*, \ w \in \mathcal{I}^{k'}, \ w^{k'} = w\})$$

$$\leq \sum_{k'=\frac{k}{2}}^{k} \Pr(w^* \notin \mathcal{I}^{k'}) + \sum_{k'=\frac{k}{2}}^{k} \sum_{w=1}^{W} \Pr(\underline{\pi}^k_{\text{Lo},w} \neq i^*_{\text{Lo},w})$$

$$(\Pr(\mathcal{W}^* \cap \mathcal{I}^{k'} = \emptyset) \leq \Pr(w^* \notin \mathcal{I}^{k'}))$$

$$+ \sum_{k'=\frac{k}{2}}^{k} \sum_{w=1}^{W} \Pr(\mathcal{E}^{\complement} \cap \{\forall k' \in [\frac{k}{2}, k], \ \exists w \notin \widetilde{\mathcal{W}}^*, \ w \in \mathcal{I}^{k'}, \ w^{k'} = w\}). \quad (26)$$

For the first and second term, by considering $f(k) = 1 + 16A^2 W(k+1)^2$, with a similar discussion as Lem. C.2 and Lem. C.5, we have, for arbitrary $k \geq k^{[W]}_{\max} := \max_{w \in [W]}\{k^w_{\max}\}$, where $k^w_{\max} := c(A + \frac{\alpha A}{\Delta^2_{\text{Lo}}(i)} \log(1 + \frac{\alpha A W}{\Delta_{\min}}))$ for some constant $c$ is an analogue of $k_{\max}$ defined in Lem. C.5 specified on task $t$:

$$\sum_{k'=\frac{k}{2}}^{k} \Pr(w^* \notin \mathcal{I}^{k'}) + \sum_{w=1}^{W} \sum_{k'=\frac{k}{2}}^{k} \Pr(\underline{\pi}^k_{\text{Lo},w} \neq i^*_{\text{Lo},w}) \leq \frac{2^{2\alpha}}{T \cdot k^{2\alpha-1}} + \frac{2 \cdot 2^{2\alpha} AW}{T \cdot k^{2\alpha-1}} + \frac{2 \cdot 2^{2\alpha-1} A^2 T}{T \cdot k^{2\alpha-2}} \leq \frac{3 \cdot 2^{2\alpha-1} A^2}{k^{2\alpha-2}}.$$

$$(27)$$

Therefore, we mainly focus on the second term. We denote $k'' := c'' \frac{\alpha A}{\Delta^2_{\min}} \log \frac{\alpha A W}{\Delta_{\min}}$ for some constant $c''$, such that for all $\widetilde{k} \geq k''$, we always have $\widetilde{k} \geq 2A \cdot \frac{512\alpha \log f(\widetilde{k})}{\Delta^2_{\min}}$. Therefore, for arbitrary $w \notin \widetilde{\mathcal{W}}^*$, and arbitrary $\widetilde{k} \geq \max\{k^{[W]}_{\max}, k''\}$ we have:

$$\Pr(\{w \in \mathcal{I}^{\widetilde{k}}\} \cap \{N^{\widetilde{k}}_{\text{Hi}}(i^*_{\text{Lo},w}) \geq \frac{512\alpha \log f(\widetilde{k})}{\Delta^2_{\min}}\})$$

$$\leq \Pr(\{w \in \mathcal{I}^{\widetilde{k}}\} \cap \{N^{\widetilde{k}}_{\text{Hi}}(i^*_{\text{Lo},w}) \geq \frac{512\alpha \log f(\widetilde{k})}{\Delta^2_{\min}}\} \cap \{N^{\widetilde{k}}_{\text{Lo},w}(i^*_{\text{Lo},w}) > \frac{\widetilde{k}}{2}\}) + \Pr(\{N^{\widetilde{k}}_{\text{Lo},w}(i^*_{\text{Lo},w}) \leq \frac{\widetilde{k}}{2}\})$$

$$\leq \frac{2A}{T \cdot \widetilde{k}^{2\alpha-1}} + \Pr(\{\underline{\mu}^{t,\widetilde{k}}_{\text{Lo}}(i^*_{\text{Lo},w}) \leq \overline{\mu}^{t,\widetilde{k}}_{\text{Hi}}(i^*_{\text{Lo},w}) + \varepsilon\} \cap \{N^{\widetilde{k}}_{\text{Lo},w}(i^*_{\text{Lo},w}) > \frac{\widetilde{k}}{2}\} \cap \{N^{\widetilde{k}}_{\text{Hi}}(i^*_{\text{Lo},w}) \geq \frac{512\alpha \log f(\widetilde{k})}{\Delta^2_{\min}}\})$$

For the second part, it equals:

$$\Pr(\{\widehat{\mu}^k_{\text{Lo}}(i^*_{\text{Lo}}) - \mu_{\text{Lo}}(i^*_{\text{Lo}}) - \sqrt{\frac{2\alpha \log f(k)}{N^k_{\text{Lo},w}(i^*_{\text{Lo}})}} \leq \widehat{\mu}^k_{\text{Hi}}(i^*_{\text{Lo}}) - \mu_{\text{Hi}}(i^*_{\text{Lo}}) + (\mu_{\text{Hi}}(i^*_{\text{Lo}}) - \mu_{\text{Lo}}(i^*_{\text{Lo}})) + \sqrt{\frac{2\alpha \log f(k)}{N^k_{\text{Hi}}(i^*_{\text{Lo}})}} + \varepsilon\}$$

$$\cap \{N^k_{\text{Lo},w}(i^*_{\text{Lo},w}) > \frac{k}{2}\} \cap \{N^k_{\text{Hi}}(i^*_{\text{Lo},w}) \geq \frac{512\alpha \log f(k)}{\Delta^2_{\min}}\})$$

$$\leq \Pr(\{\widehat{\mu}^k_{\text{Hi}}(i^*_{\text{Lo}}) - \mu_{\text{Lo}}(i^*_{\text{Lo}}) - \sqrt{\frac{2\alpha \log f(k)}{N^k_{\text{Lo},w}(i^*_{\text{Lo}})}} \leq \widehat{\mu}^k_{\text{Hi}}(i^*_{\text{Lo}}) - \mu_{\text{Hi}}(i^*_{\text{Lo}}) - \frac{\Delta_{\min}}{4} + \sqrt{\frac{2\alpha \log f(k)}{N^k_{\text{Hi}}(i^*_{\text{Lo}})}}\}$$

$$\cap \{N_{\text{Lo},w}^k(i_{\text{Lo},w}^*) > \frac{k}{2}\} \cap \{N_{\text{Hi}}^k(i_{\text{Lo},w}^*) \geq \frac{512\alpha \log f(k)}{\Delta_{\min}^2}\})$$

$$\leq \Pr(\widehat{\mu}_{\text{Lo}}^k(i_{\text{Lo}}^*) - \mu_{\text{Lo}}(i_{\text{Lo}}^*) \leq -\sqrt{\frac{2\alpha \log f(k)}{N_{\text{Lo},w}^k(i_{\text{Lo}}^*)}}) + \Pr(\sqrt{\frac{2\alpha \log f(k)}{N_{\text{Hi}}^k(i_{\text{Lo}}^*)}} \leq \widehat{\mu}_{\text{Hi}}^k(i_{\text{Lo}}^*) - \mu_{\text{Hi}}(i_{\text{Lo}}^*))$$

$$(\sqrt{\frac{2\alpha \log f(k)}{N_{\text{Lo},w}^k(i_{\text{Lo}}^*)}}, \sqrt{\frac{2\alpha \log f(k)}{N_{\text{Hi}}^k(i_{\text{Lo}}^*)}} \leq \frac{\Delta_{\min}}{16})$$

$$\leq \frac{1}{T \cdot k^{2\alpha}}.$$

Therefore, we can conclude that

$$\forall k \geq \max\{k_{\max}^{[W]}, k''\}, \quad \Pr(\{w \in \mathcal{I}^k\} \cap \{N_{\text{Hi}}^k(i_{\text{Lo},w}^*) \geq \frac{512\alpha \log f(k)}{\Delta_{\min}^2}\}) \leq \frac{4}{T \cdot k^{2\alpha}} + \frac{2A}{T \cdot k^{2\alpha-1}} \leq \frac{3A}{T \cdot k^{2\alpha-1}}.$$

Next, we are ready to upper bound the second term in Eq. (26). The key observation is that, as long as the event $\{\forall k' \in [\frac{k}{2}, k], \exists w \notin \widetilde{\mathcal{W}}^*, w \in \mathcal{I}^{k'}, w^{k'} = w\}$ happens, no matter what the sequence $\{w^{k'}\}_{k'=k/2}^k$ is, since we only have $A - 1$ sub-optimal arms, and $i_{\text{Lo},w^{k'}}^* \neq i_{\text{Hi}}^*$ for all $k' \in [k/2, k]$, there must be an arm which has been taken for at least $\frac{k}{2(A-1)}$ times from step $k/2$ to $k$, therefore,

$$\Pr(\forall k' \in [\frac{k}{2}, k], \exists w \notin \widetilde{\mathcal{W}}^*, w \in \mathcal{I}^{k'}, w^{k'} = w)$$

$$\leq \Pr(\exists k' \in [\frac{k}{2}, k], \exists w \notin \widetilde{\mathcal{W}}^*, s.t. N_{\text{Hi}}^{k'}(i_{\text{Lo},w}^*) = N_{\text{Hi}}^{k/2}(i_{\text{Lo},w}^*) + \frac{k}{2(A-1)} - 1, w \in \mathcal{I}^{k'})$$

$$\leq \sum_{k'=\frac{k}{2}}^k \sum_{w \notin \widetilde{\mathcal{W}}^*} \Pr(\{w \in \mathcal{I}^{k'}\} \cap \{N_{\text{Hi}}^{k'}(i_{\text{Lo},w}^*) \geq \frac{k}{2(A-1)} - 1, w \in \mathcal{I}^{k'}\})$$

$$\leq \sum_{k'=\frac{k}{2}}^k \sum_{w \notin \widetilde{\mathcal{W}}^*} \Pr(\{w \in \mathcal{I}^{k'}\} \cap \{N_{\text{Hi}}^{k'}(i_{\text{Lo},w}^*) \geq \frac{512\alpha \log f(k)}{\Delta_{\min}^2}\})$$

$$\leq \frac{3 \cdot 2^{2\alpha-1} A}{k^{2\alpha-2}}. \tag{28}$$

According to the definition of $k_{\max}^{[W]}$ and $k''$, there must exists a constant $c^*$ such that $\max\{k'', k_{\max}^{[W]}\} \leq c^* \frac{\alpha A}{\Delta_{\min}^2} \log \frac{\alpha AW}{\Delta_{\min}}$. By choosing such $c^*$, and combining Eq. (27) and Eq. (28), we have:

$$\Pr(\pi_{\text{Hi}}^k \neq i_{\text{Hi}}^*) \leq \frac{3 \cdot 2^{2\alpha-1} A^2}{k^{2\alpha-2}} + \frac{3 \cdot 2^{2\alpha-1} A}{k^{2\alpha-2}} \leq \frac{16A}{(k/2)^{2\alpha-2}} = O(\frac{A}{k^{2\alpha-2}}).$$

$\square$

**Lemma E.1** (Extension of Lem. C.6). *For arbitrary $K \geq A + 1$ and arbitrary $1 \leq k_0 \leq K$, we have:*

$$N_{Hi}^K(i) \leq k_0 + \sum_{k=k_0+1}^K \mathbb{I}[\{w^k \neq \text{Null}\} \cap \{\underline{\mu}_{Lo,w^k}^k(\underline{\pi}_{Lo,w^k}^k) \leq \overline{\mu}_{Hi}^k(\underline{\pi}_{Lo,w^k}^k) + \varepsilon\}$$

$$\cap \{N_{Lo,w^k}^k(\underline{\pi}_{Lo,w^k}^k) > k/2\} \cap \{i = \underline{\pi}_{Lo,w^k}^k\} \cap \{\pi_{Hi}^k = i\}]$$

$$+ \sum_{k=k_0+1}^K \mathbb{I}[0 \geq \widehat{\mu}_{Hi}^k(i^*) + \sqrt{\frac{2\alpha \log f(k)}{N_{Hi}^k(i^*)}} - \mu_{Hi}(i^*)]$$

$$+ \sum_{k=k_0+1}^K \mathbb{I}[\widetilde{\mu}_{Hi}^k(i) + \sqrt{\frac{2\alpha \log f(K)}{k}} - \mu_{Hi}(i) - \Delta_{Hi}(i) \geq 0] \tag{29}$$

We omit the proof here since it is almost the same as Lem. C.6, except that we need to specify $w^k$.

**Theorem 5.1.** *[Tiered MAB with Multiple Source Tasks] Under Assump. A, B, and C, by running Alg. 3 with $\mathcal{M}_{Lo} = \{M_{Lo,w}\}_{w=1}^{W}$ and $M_{Lo}$, with $\varepsilon = \frac{\widetilde{\Delta}_{\min}}{4}$ and $\alpha > 2$, we always have: $Regret_K(M_{Hi}) = O(\sum_{\Delta_{Hi}(i)>0} \frac{1}{\Delta_{Hi}(i)} \log(WK))$. Moreover, if at least one task in $\mathcal{M}_{Lo}$ is $\frac{\widetilde{\Delta}_{\min}}{4}$-close to $M_{Hi}$, we further have: $Regret_K(M_{Hi}) = O(\sum_{\Delta_{Hi}(i)>0} \frac{1}{\Delta_{Hi}(i)} \log \frac{AW}{\Delta_{\min}})$.*

*Proof.* One key observation is that when analyzing $N_{Hi}^K$, we only need to analyze the second term in Eq. (29), since the others can be directly bounded.

We first study the case when there exists $w^*$ such that $M_{Hi}$ and $M_{Lo,w^*}$ are $\varepsilon$-close. As a result of Lem. 5.3, we only need to upper bound the regret before step $k^*$. Similar to the proof of Thm. 4.1, we separate two cases for each $k$ and $i$.

**Case 1-(a)** $w^k \neq \texttt{Null}$, $0 < \Delta_{\mathbf{Hi}}(i) \leq 4\Delta_{\mathbf{Lo},w^k}(i)$   In the following, we will define $\widetilde{k}_{t,i} := 3 + c \cdot \frac{3\alpha}{\Delta_{Lo,w}^2(i)} \log(T + \frac{\alpha AT}{\Delta_{\min}})$ (i.e. similar to the role of $\widetilde{k}_i$ in Lem. C.5 with specified task index $t$). Since $\Delta_{Hi}(i) \leq 4\Delta_{Lo,w^k}(i)$, we define $\widetilde{k}_{[W],i} := 3 + c \cdot \frac{48\alpha}{\Delta_{Hi}^2(i)} \log(T + \frac{\alpha AT}{\Delta_{\min}})$. In this case, obviously $\widetilde{k}_{[W],i} \geq \widetilde{k}_{t,i}$. As a result of Lem. C.5,

$$\mathbb{I}[\{w^k \neq \texttt{Null}\} \cap \{\underline{\mu}_{Lo,w^k}^k(\underline{\pi}_{Lo,w^k}^k) \leq \overline{\mu}_{Hi}^k(\underline{\pi}_{Lo,w^k}^k) + \varepsilon\} \cap \{N_{Lo,w^k}^k(\underline{\pi}_{Lo,w^k}^k) > k/2\} \cap \{i = \underline{\pi}_{Lo,w^k}^k\} \cap \{\pi_{Hi}^k = i\}]$$

$$\leq \mathbb{I}[k \leq \widetilde{k}_{[W],i}] + \mathbb{I}[\{w^k \neq \texttt{Null}\} \cap \{k > \widetilde{k}_{[W],i}\} \cap \{N_{Lo,w^k}^k(\underline{\pi}_{Lo,w^k}^k) > k/2\}]. \tag{30}$$

by taking expectation, we have:

$$\Pr(\mathbb{I}[\{w^k \neq \texttt{Null}\} \cap \{\underline{\mu}_{Lo,w^k}^k(\underline{\pi}_{Lo,w^k}^k) \leq \overline{\mu}_{Hi}^k(\underline{\pi}_{Lo,w^k}^k) + \varepsilon\}$$

$$\cap \{N_{Lo,w^k}^k(\underline{\pi}_{Lo,w^k}^k) > k/2\} \cap \{i = \underline{\pi}_{Lo,w^k}^k\} \cap \{\pi_{Hi}^k = i\}])$$

$$\leq \mathbb{I}[k \leq \widetilde{k}_{[W],i}] + \frac{2}{k^{2\alpha-1}}.$$

**Case 1-(b)** $w^k \neq \texttt{Null}$, $\Delta_{\mathbf{Hi}}(i) > 4\Delta_{\mathbf{Lo},w^k}(i) \geq 0$   We consider $\bar{k}_i := \frac{c_{Hi,i}\alpha}{\Delta_{Hi}^2(i)} \log \frac{AW}{\Delta_{\min}}$, where $c_{Hi,i}$ is the minimal constant, such that when $k \geq \frac{c_{Hi,i}\alpha}{\Delta_{Hi}^2(i)} \log \frac{\alpha AW}{\Delta_{\min}}$, we always have $k \geq \frac{256\alpha \log f(k)}{\Delta_{Hi}^2(i)}$. With a similar discussion as Eq. (13), we have:

$$\mathbb{I}[\{w^k \neq \texttt{Null}\} \cap \{\underline{\mu}_{Lo,w^k}^k(\underline{\pi}_{Lo,w^k}^k) \leq \overline{\mu}_{Hi}^k(\underline{\pi}_{Lo,w^k}^k) + \varepsilon\} \cap \{N_{Lo,w^k}^k(\underline{\pi}_{Lo,w^k}^k) > k/2\} \cap \{i = \underline{\pi}_{Lo,w^k}^k\} \cap \{\pi_{Hi}^k = i\}]$$

$$\leq \mathbb{I}[k < \bar{k}_i] + \mathbb{I}[\{w^k \neq \texttt{Null}\} \cap \{\{\underline{\mu}_{Lo,w^k}^k(\underline{\pi}_{Lo,w^k}^k) \leq \overline{\mu}_{Hi}^k(\underline{\pi}_{Lo,w^k}^k) + \varepsilon\}\}$$

$$\cap \{N_{Lo,w^k}^k(i) \geq \frac{128\alpha \log f(k)}{\Delta_{Hi}^2(i)}\} \cap \{i = \underline{\pi}_{Lo}^k\} \cap \{\pi_{Hi}^k = i\}]$$

$$\leq \mathbb{I}[k < \bar{k}_i] + \mathbb{I}[\{w^k \neq \texttt{Null}\} \cap \{\widehat{\mu}_{Lo,w^k}^k(i) - \mu_{Lo,w^k}(i) \leq -\sqrt{\frac{2\alpha \log f(k)}{N_{Lo,w^k}^k(i)}}\}]$$

$$+ \mathbb{I}[\{\widehat{\mu}_{Hi}^k(i) - \mu_{Hi}(i) + \sqrt{\frac{2\alpha \log f(k)}{N_{Hi}^k(i)}} \geq \frac{\Delta_{Hi}(i)}{4}\} \cap \{\pi_{Hi}^k = i\}]. \tag{31}$$

We denote $k_i^* := \max\{\widetilde{k}_{[W],i}, \bar{k}_i\}$. Combining Eq. (30) and (31) with Eq. (29), for arbitrary $K$, we have (recall that $\widetilde{\mu}_{Hi}^k(i)$ is defined to be the average of $k$ random samples from reward distribution of arm $i$ in $M_{Hi}$):

$$\mathbb{E}[N_{Hi}^K(i)]$$

$$\leq \sum_{k=1}^{K} \mathbb{I}[k \leq \widetilde{k}_{[W],i}] + \sum_{k=1}^{K} \frac{2}{k^{2\alpha-1}} + \sum_{k=1}^{K} \mathbb{I}[k < \bar{k}_i]$$

$$+ \sum_{k=1}^{K} \Pr(\{w^k \neq \texttt{Null}\} \cap \{\widehat{\mu}_{Lo,w^k}^k(i) - \mu_{Lo,w^k}(i) \leq -\sqrt{\frac{2\alpha \log f(k)}{N_{Lo,w^k}^k(i)}}\})$$

$$+\mathbb{E}[\sum_{k=1}^{K}\mathbb{I}[\{\widehat{\mu}_{\mathrm{Hi}}^{k}(i)-\mu_{\mathrm{Hi}}(i)+\sqrt{\frac{2\alpha\log f(k)}{N_{\mathrm{Hi}}^{k}(i)}}\geq\frac{\Delta_{\mathrm{Hi}}(i)}{4}\}\cap\{\pi_{\mathrm{Hi}}^{k}=i\}]]$$

$$+\sum_{k=1}^{K}\Pr(0\geq\widehat{\mu}_{\mathrm{Hi}}^{k}(i^{*})+\sqrt{\frac{2\alpha\log f(k)}{N_{\mathrm{Hi}}^{k}(i^{*})}}-\mu_{\mathrm{Hi}}(i^{*}))$$

$$+\sum_{k=1}^{K}\mathbb{E}[\mathbb{I}[\widetilde{\mu}_{\mathrm{Hi}}^{k}(i)+\sqrt{\frac{2\alpha\log f(K)}{k}}-\mu_{\mathrm{Hi}}(i)-\Delta_{\mathrm{Hi}}(i)\geq 0]]$$

$$\leq\widetilde{k}_{[W],i}+\sum_{k=1}^{K}\frac{2}{k^{2\alpha-1}}+\bar{k}_{i}+2\cdot\sum_{k=1}^{K}\frac{1}{8Ak^{2\alpha}}+2\sum_{k=1}^{K}\mathbb{E}[\mathbb{I}[\widetilde{\mu}_{\mathrm{Hi}}^{k}(i)+\sqrt{\frac{2\alpha\log f(K)}{k}}-\mu_{\mathrm{Hi}}(i)-\frac{\Delta_{\mathrm{Hi}}(i)}{4}\geq 0]]$$

$$=O(\frac{1}{\Delta_{\mathrm{Hi}}^{2}(i)}\log WK). \tag{32}$$

As a result, combining with Lem. 5.3, we can conclude that:

$$\mathrm{Regret}_{K}(M_{\mathrm{Hi}})=\sum_{i\neq i^{*}}\Delta_{\mathrm{Hi}}(i)\mathbb{E}[N_{\mathrm{Hi}}^{k^{*}}(i)]+\sum_{k=k^{*}+1}^{K}\Pr(\pi_{\mathrm{Hi}}^{k}\neq i_{\mathrm{Hi}}^{*})=O(\sum_{i\neq i^{*}}\frac{1}{\Delta_{\mathrm{Hi}}(i)}\log\frac{AW}{\Delta_{\min}})$$

Next, we study the case when there is no task among $\{M_{\mathrm{Lo},w}\}_{w=1}^{W}$ close enough to $M_{\mathrm{Hi}}$. Similarly, we also decompose into three cases.

**Case 2-(a)** $w^{k}\neq\texttt{Null}, i\neq i_{\mathbf{Lo},w^{k}}^{*}, i\neq i_{\mathbf{Hi}}^{*}$ **and** $0<\Delta_{\mathbf{Hi}}(i)\leq 4\Delta_{\mathbf{Lo},w^{k}}(i)$   The result is the same as Eq. (30).

**Case 2-(b)** $w^{k}\neq\texttt{Null}, i\neq i_{\mathbf{Lo},w^{k}}^{*}, i\neq i_{\mathbf{Hi}}^{*}$ **and** $\Delta_{\mathbf{Hi}}(i)>4\Delta_{\mathbf{Lo},w^{k}}(i)>0$   The result is the same as Eq. (31).

**Case 2-(c)** $w^{k}\neq\texttt{Null}, i=i_{\mathbf{Lo},w^{k}}^{*}$   If $i_{\mathrm{Lo},w^{k}}^{*}=i_{\mathrm{Hi}}^{*}$, $M_{\mathrm{Hi}}$ suffers no regret when choosing $i_{\mathrm{Lo},w^{k}}^{*}$. Therefore, in the following, we only study the case when $i_{\mathrm{Lo},w^{k}}^{*}\neq i_{\mathrm{Hi}}^{*}$. For arm $i$ (note that $i=i_{\mathrm{Lo},w^{k}}^{*}$ in this case), we define $k_{\max}'=\frac{c_{\max}'\alpha}{\Delta_{\mathrm{Hi}}(i)^{2}}\log\frac{\alpha AT}{\Delta_{\min}}$, where $c_{\max}'$ is the minimal constant, such that for all $k\geq\frac{c_{\max}'\alpha}{\Delta_{\mathrm{Hi}}(i)^{2}}\log\frac{\alpha AT}{\Delta_{\min}}$, we always have $k\geq\frac{1024\alpha\log f(k)}{\Delta_{\mathrm{Hi}}(i)^{2}}$. With a similar discussion as Eq. (15), for the following event, we have:

$$\mathbb{I}[\{w^{k}\neq\texttt{Null}\}\cap\{\underline{\mu}_{\mathrm{Lo},w^{k}}^{k}(\underline{\pi}_{\mathrm{Lo},w^{k}}^{k})\leq\overline{\mu}_{\mathrm{Hi}}^{k}(\underline{\pi}_{\mathrm{Lo},w^{k}}^{k})+\varepsilon\}\cap\{N_{\mathrm{Lo},w^{k}}^{k}(\underline{\pi}_{\mathrm{Lo},w^{k}}^{k})>k/2\}\cap\{i=\underline{\pi}_{\mathrm{Lo},w^{k}}^{k}\}\cap\{\pi_{\mathrm{Hi}}^{k}=i\}]$$

$$\leq\mathbb{I}[k\leq k_{\max}']+\mathbb{I}[\{w^{k}\neq\texttt{Null}\}\cap\widehat{\mu}_{\mathrm{Lo},w^{k}}^{k}(i)-\mu_{\mathrm{Lo},w^{k}}(i)\leq-\sqrt{\frac{2\alpha\log f(k)}{N_{\mathrm{Lo},w^{k}}^{k}(i_{\mathrm{Lo}}^{*})}}]$$

$$+\mathbb{I}[\{\widehat{\mu}_{\mathrm{Hi}}^{k}(i)-\mu_{\mathrm{Hi}}(i)+\sqrt{\frac{2\alpha\log f(k)}{N_{\mathrm{Hi}}^{k}(i)}}\geq\frac{\Delta_{\mathrm{Hi}}(i)}{8}\}\cap\{\pi_{\mathrm{Hi}}^{k}=i\}].$$

Combining all the cases above, similar to Case 1, for arbitrary $i\neq i_{\mathrm{Hi}}^{*}$, we can conclude:

$$\mathbb{E}[N_{\mathrm{Hi}}^{K}(i)]$$

$$\leq\sum_{k=1}^{K}\mathbb{I}[k\leq\widetilde{k}_{[W],i}]+\sum_{k=1}^{K}\frac{2}{k^{2\alpha-1}}+\sum_{k=1}^{K}\mathbb{I}[k<\bar{k}_{i}]+\sum_{k=1}^{K}\Pr(\{w^{k}\neq\texttt{Null}\}\cap\{\widehat{\mu}_{\mathrm{Lo}}^{k}(i)-\mu_{\mathrm{Lo}}(i)\leq-\sqrt{\frac{2\alpha\log f(k)}{N_{\mathrm{Lo},w^{k}}^{k}(i)}}\})$$

$$+\mathbb{E}[\sum_{k=1}^{K}\mathbb{I}[\{\widehat{\mu}_{\mathrm{Hi}}^{k}(i)-\mu_{\mathrm{Hi}}(i)+\sqrt{\frac{2\alpha\log f(k)}{N_{\mathrm{Hi}}^{k}(i)}}\geq\frac{\Delta_{\mathrm{Hi}}(i)}{4}\}\cap\{\pi_{\mathrm{Hi}}^{k}=i\}]]$$

$$+\sum_{k=1}^{K}\Pr(0\geq\widehat{\mu}_{\mathrm{Hi}}^{k}(i^{*})+\sqrt{\frac{2\alpha\log f(k)}{N_{\mathrm{Hi}}^{k}(i^{*})}}-\mu_{\mathrm{Hi}}(i^{*}))$$

$$+\sum_{k=1}^{K}\mathbb{E}[\mathbb{I}[\widetilde{\mu}_{\mathrm{Hi}}^{k}(i)+\sqrt{\frac{2\alpha\log f(K)}{k}}-\mu_{\mathrm{Hi}}(i)-\Delta_{\mathrm{Hi}}(i)\geq 0]]$$

$$+ \sum_{k=1}^{K} \mathbb{I}[k \leq k'_{\max}] + \sum_{k=1}^{K} \Pr(\{w^k \neq \texttt{Null}\} \cap \{\widehat{\mu}_{\mathrm{Lo}}^k(i) - \mu_{\mathrm{Lo}}(i) \leq -\sqrt{\frac{2\alpha \log f(k)}{N_{\mathrm{Lo},w^k}^k(i)}}\})$$

$$+ \mathbb{E}[\sum_{k=1}^{K} \mathbb{I}[\{\widehat{\mu}_{\mathrm{Hi}}^k(i) - \mu_{\mathrm{Hi}}(i) + \sqrt{\frac{2\alpha \log f(k)}{N_{\mathrm{Hi}}^k(i)}} \geq \frac{\Delta_{\mathrm{Hi}}(i)}{4}\} \cap \{\pi_{\mathrm{Hi}}^k = i\}]]$$

$$\leq \widetilde{k}_{[W],i} + \sum_{k=1}^{K} \frac{2}{k^{2\alpha-1}} + \bar{k}_i + 2 \cdot \sum_{k=1}^{K} \frac{1}{8Ak^{2\alpha}} + 3\sum_{k=1}^{K} \mathbb{E}[\mathbb{I}[\widetilde{\mu}_{\mathrm{Hi}}^k(i) + \sqrt{\frac{2\alpha \log f(K)}{k}} - \mu_{\mathrm{Hi}}(i) - \frac{\Delta_{\mathrm{Hi}}(i)}{8} \geq 0]]$$

$$= O(\frac{1}{\Delta_{\mathrm{Hi}}^2(i)} \log WK).$$

which implies:

$$\mathrm{Regret}_K(M_{\mathrm{Hi}}) = \sum_{i \neq i^*} \Delta_{\mathrm{Hi}}(i)\mathbb{E}[N_{\mathrm{Hi}}^K(i)] = O(\sum_{i \neq i^*} \frac{1}{\Delta_{\mathrm{Hi}}(i)} \log WK).$$

$\square$

## F  Proofs for Tiered RL with Multiple Source/Low-Tier Tasks

We first introduce the notion of transferable states in this multi-source tasks setting. Comparing with Def. 2.2, we have an additional constraint on $d_{\mathrm{Hi}}^*(s_h) > 0$. This is because, to distinguish which source task to transfer, we require $s_h$ in $M_{\mathrm{Hi}}$ to be visited frequently enough for accurate estimation, and it is only possible for those $s_h$ on optimal trajectories given that we expect $\mathrm{Regret}_K(M_{\mathrm{Hi}})$ is at least near-optimal.

**Definition F.1** ($\lambda$-Transferable States in TRL-MST). Given any $\lambda > 0$, we say $s_h$ is $\lambda$-transferable if $d_{\mathrm{Hi}}^*(s_h) > 0$, and $\exists w \in [W]$, such that $d_{\mathrm{Lo},w}^*(s_h) \geq \lambda$ and $M_{\mathrm{Hi}}$ is $\frac{\widetilde{\Delta}_{\min}}{4(H+1)}$-close to $M_{\mathrm{Lo},w}$ on state $s_h$. We use $\mathcal{Z}_h^{\lambda,[W]}$ to denote the set of $\lambda$-transferable state at step $h \in [H]$.

**Definition F.2** (Benefitable States in TRL-MST). Similar to the single task case, we define $\mathcal{C}_h^{\lambda,[W],1} := \{(s_h, a_h)|s_h \in \mathcal{Z}^{\lambda,[W]}, a_h \neq \pi_{\mathrm{Hi},h}^*(s_h)\}$, $\mathcal{C}_h^{\lambda,[W],2} := \{(s_h,a_h)|\mathrm{Block}(\{\mathcal{C}_{h'}^{\lambda,[W],1}\}_{h'=1}^{h-1}, s_h) = \mathrm{True}, s_h \notin \mathcal{C}_h^{\lambda,[W],1}, a_h \in \mathcal{A}_h\}$ and $\mathcal{C}_h^* := \{(s_h,a_h)|d_{\mathrm{Hi}}^*(s_h,a_h) > 0\}$, which represents the three categories of state-action pairs with constant regret. We define $\mathcal{C}_h^{\lambda,[W]} := \mathcal{C}_h^{\lambda,[W],1} \cup \mathcal{C}_h^{\lambda,[W],2} \cup \mathcal{C}_h^*$, which captures the benefitable state-action pairs.

**Remark F.3** (Constant Regret in Entire $M_{\mathrm{Hi}}$). *For each individual task $M_{Lo,w}$, the additional constraint $d_{Hi}^*(s_h) > 0$ reduces the size of transferable states comparing with single task learning setting. However, if the tasks are diverse enough, we expect $\mathcal{C}_h^{\lambda,[W]}$ to be much larger than $\mathcal{C}_h^\lambda$ in Sec. 4.2 and we can achieve more benefits with only an additional cost of order $\log W$. Besides, if $\forall h, s_h$ with $d_{Hi}^*(s_h) > 0$ we have $s_h \in \mathcal{Z}_h^{\lambda,[W]}$, then, $\mathcal{C}_h^{\lambda,[W]} = \mathcal{S}_h \times \mathcal{A}_h$ and we can achieve constant regret for the entire $M_{Hi}$.*

### F.1  Additional Algorithms, Conditions and Notations

Our algorithm is provided in Alg. 7, which is extended from Alg. 2 and integrated with the "Trust till Failure" strategy introduced in bandit setting in Alg. 3. We consider the same **ModelLearning** and **Bonus** algorithm in Sec. D.1, but a different condition for $\mathrm{Alg}^{\mathrm{Lo}}$ listed below:

**Condition F.4** (Condition on $\mathrm{Alg}^{\mathrm{Lo}}$ in MT-TRL). $\mathrm{Alg}^{\mathrm{Lo}}$ is an algorithm which returns deterministic policies at each iteration for each task $M_{tO,w} \in \mathcal{M}_{\mathrm{Lo}}$, and there exists $C_1, C_2$ only depending on $S, A, H$ and $\Delta_{\min}$ but independent of $k$, such that for arbitrary $k \geq 2$, we have $\Pr(\mathcal{E}_{\mathrm{Alg}^{\mathrm{Lo}},[W],k}) \geq 1 - \frac{1}{k^\alpha}$ for $\mathcal{E}_{\mathrm{Alg}^{\mathrm{Lo}},[W],k}$ defined below:

$$\mathcal{E}_{\mathrm{Alg}^{\mathrm{Lo}},[W],k} := \bigcap_{w \in W} \{\sum_{\widetilde{k}=1}^{k} V_{\mathrm{Lo},w,1}^*(s_1) - V_{\mathrm{Lo},w,1}^{\pi_{\mathrm{Lo}}^{\widetilde{k}}}(s_1) \leq C_1 + \alpha C_2 \log Wk\}.$$

**Algorithm 7:** Robust Tiered RL with Multiple Low-Tier Tasks

---

**1 Input**: Ratio $\lambda \in (0,1)$; Bonus term computation function **Bonus**; Sequence of confidence level $(\delta_k)_{k \geq 1}$ with $\delta_k = 1/SAHWk^{\alpha}$; Model learning function **ModelLearning**.
   $\varepsilon < \Delta_{\min}/4(H+1)$

**2 Initialize**: $D^0_{\text{Lo},w} \leftarrow \{\}$ for $w \in [W]$, $D^0_{\text{Hi}} \leftarrow \{\}$, set
   $\underline{V}^k_{\text{Hi},h+1}, \underline{Q}^k_{\text{Hi},h+1}, \underline{V}^k_{\text{Lo},w,h+1}, \underline{Q}^k_{\text{Lo},w,h+1}, \widetilde{V}^k_{\text{Lo},w,h+1}, \widetilde{Q}^k_{\text{Lo},w,h+1}$ to be 0 for all $k = 1, 2, ....$

**3 for** $k = 1, 2, ...$ **do**

**4**    **for** $t = 1, 2, ...T$ **do**

**5**      $\pi^k_{\text{Lo},w} \leftarrow \text{Alg}^{\text{Lo}}(D^{k-1}_{\text{Lo},w})$; collect data from $M_{\text{Lo},w}$ with $\pi^k_{\text{Lo},w}$; update
        $D^k_{\text{Lo},w} \leftarrow D^{k-1}_{\text{Lo},w} \cup \{\tau^k_t\}$.

**6**      $\{\widehat{\mathbb{P}}^k_{\text{Lo},w,h}\}^H_{h=1} \leftarrow \textbf{ModelLearning}(D^{k-1}_{\text{Lo},w}), \quad \{b^k_{\text{Lo},h}\}^H_{h=1} \leftarrow \textbf{Bonus}(D^{k-1}_{\text{Lo},w}, \delta_k)$.

**7**      **for** $h = H, H-1..., 1$ **do**

**8**        $\underline{Q}^k_{\text{Lo},w,h}(\cdot,\cdot) \leftarrow \max\{0, r_{\text{Lo},h}(\cdot,\cdot) + \widehat{\mathbb{P}}^k_{\text{Lo},w,h}\underline{V}^k_{\text{Lo},w,h+1}(\cdot,\cdot) - b^k_{\text{Lo},h}(\cdot,\cdot)\}$.

**9**        $\underline{V}^k_{\text{Lo},w,h}(\cdot) = \max_a \underline{Q}^k_{\text{Lo},w,h}(\cdot,a), \quad \underline{\pi}^k_{\text{Lo},w,h}(\cdot) \leftarrow \arg\max_a \underline{Q}^k_{\text{Lo},w,h}(\cdot,a)$.

**10**      **end**

**11**    **end**

**12**    $\{\widehat{\mathbb{P}}^k_{\text{Hi},h}\}^H_{h=1} \leftarrow \textbf{ModelLearning}(D^{k-1}_{\text{Hi}}), \quad \{b^k_{\text{Hi},h}\}^H_{h=1} \leftarrow \textbf{Bonus}(D^{k-1}_{\text{Hi}}, \delta_k)$.

**13**    **for** $h = H, H-1..., 1$ **do**

**14**      $\underline{Q}^{\pi^k_{\text{Hi}}}_{\text{Hi},h}(\cdot,\cdot) \leftarrow \max\{0, r^k_{\text{Hi},h}(\cdot,\cdot) + \widehat{\mathbb{P}}^{\pi^k_{\text{Hi}}}_{\text{Hi},h}\underline{V}^k_{\text{Hi},h+1}(\cdot,\cdot) - b^k_{\text{Hi},h}(\cdot,\cdot)\}, \quad \underline{V}^{\pi^k_{\text{Hi}}}_{\text{Hi},h}(\cdot) =$
        $\underline{Q}^{\pi^k_{\text{Hi}}}_{\text{Hi},h}(\cdot, \pi^k_{\text{Hi}})$

**15**      $\widetilde{Q}^k_{\text{Hi},h}(\cdot,\cdot) \leftarrow \min\{H, r_{\text{Hi}}(\cdot,\cdot) + \widehat{\mathbb{P}}^k_{\text{Hi},h}\widetilde{V}^k_{\text{Hi},h+1}(\cdot,\cdot) + b^k_{\text{Hi},h}(\cdot,\cdot)\}$.

**16**      **for** $s_h \in \mathcal{S}_h$ **do**

**17**        $\mathcal{I}^k_{s_h} \leftarrow \{w \in [W] | \{\underline{V}^k_{\text{Lo},w,h}(s_h) \leq$
          $\widetilde{Q}^k_{\text{Hi},h}(s_h, \underline{\pi}^k_{\text{Lo},w,h}) + \varepsilon\} \cap \{\max_a N^k_{\text{Lo},w,h}(s_h,a) \geq \frac{\lambda}{3}k\}\}$.

**18**        **if** $\mathcal{I}^k(s_h) \neq \emptyset$ **then**

**19**          **if** $w^{k-1}_{s_h} \neq \texttt{Null}$ *and* $w^{k-1} \in \mathcal{I}^k_{s_h}$ **then** $w^k_{s_h} \leftarrow w^{k-1}_{s_h}$ ;

**20**          **else if** $w^{k-1}_{s_h} \neq \texttt{Null}$ *and* $\exists w \in \mathcal{I}^k(s_h)$ *s.t.*
            $\pi^{k-1}_{Hi}(s_h) = \arg\max_a N_{Lo,w,h}(s_h,a)$ **then** $w^k_{s_h} \leftarrow w$ ;

**21**          **else** $w^k_{s_h} \leftarrow \text{Unif}(\mathcal{I}^k_{s_h})$. ;

**22**          $\pi^k_{\text{Hi}}(s_h) \leftarrow \arg\max_a N^k_{\text{Lo},w^k_{s_h},h}(s_h,a)$.

**23**        **end**

**24**        **else** $w^k_{s_h} \leftarrow \texttt{Null}$, $\pi^k_{\text{Hi}}(s_h) \leftarrow \arg\max_a \widetilde{Q}^k_{\text{Hi},h}(s_h,a)$ ;

**25**        $\widetilde{V}^k_{\text{Hi},h}(s_h) \leftarrow \min\{H, \widetilde{Q}^k_{\text{Hi},h}(s_h,\pi^k_h) + \frac{1}{H}(\widetilde{Q}^k_{\text{Hi},h}(s_h,\pi^k_h) - \underline{Q}^{\pi^k_{\text{Hi}}}_{\text{Hi},h}(s_h,\pi^k_h))\}$

**26**      **end**

**27**    **end**

**28**    Deploy $\pi_{\text{Hi}}$ to interact with $M_{\text{Hi}}$ and receive $\tau^k_{\text{Hi}}$; update $D^k_{\text{Hi}} \leftarrow D^{k-1}_{\text{Hi}} \cup \{\tau^k_{\text{Hi}}\}$

**29 end**

Next, we introduce some notations. As analogues of $\mathcal{E}_{\textbf{Bonus},k}$ and $\mathcal{E}_{\textbf{Con},k}$ in single low-tier task setting, we consider the following events:

$$\mathcal{E}_{\textbf{Bonus},[W],k} := \bigcap_{\substack{(\cdot)\in\{\text{Hi},\text{Lo}_1,\dots,\text{Lo}_T\}, \\ h\in[H], \\ s_h\in\mathcal{S}_h, a_h\in\mathcal{A}_h}} \Big\{\{H\cdot\|\widehat{\mathbb{P}}^k_{(\cdot),h}(s_h,a_h)-\mathbb{P}_{(\cdot),h}(s_h,a_h)\|_1 < b^k_{(\cdot),h}(s_h,a_h) \le B_1\sqrt{\frac{\log(B_2/\delta_k)}{N^k_{(\cdot),h}(s_h,a_h)}}\}\Big\};$$

$$\mathcal{E}_{\textbf{Con},[W],k} := \bigcap_{\substack{h\in[H], \\ s_h\in\mathcal{S}_h, \\ a_h\in\mathcal{A}_h}} \Big\{\{\frac{1}{2}\sum_{\widetilde{k}=1}^{k} d^{\pi^{\widetilde{k}}_{\text{Hi}}}(s_h,a_h)-\alpha\log(2SAHWk) \le N^k_{\text{Hi},h}(s_h,a_h)$$

$$\le e\sum_{\widetilde{k}=1}^{k} d^{\pi^{\widetilde{k}}_{\text{Hi}}}(s_h,a_h)+\alpha\log(2SAHWk)\}$$

$$\cap\Big(\bigcap_{w\in[W]}\{\frac{1}{2}\sum_{\widetilde{k}=1}^{k} d^{\pi^{\widetilde{k}}_{\text{Lo},w}}(s_h,a_h)-\alpha\log(2SAHWk) \le N^k_{\text{Lo},w,h}(s_h,a_h)$$

$$\le e\sum_{\widetilde{k}=1}^{k} d^{\pi^{\widetilde{k}}_{\text{Lo},w}}(s_h,a_h)+\alpha\log(2SAHWk)\}\Big)\Big\}.$$

Under the choice of $\delta_k = 1/SAHWk^\alpha$, we have $\Pr(\mathcal{E}_{\textbf{Bonus},[W],k}) \ge 1-\frac{1}{k^\alpha}$. Besides, as a result of Lem. D.8, we have $\Pr(\mathcal{E}_{\textbf{Con},[W],k}) \ge 1-\frac{1}{k^\alpha}$.

### F.2 Analysis

In this sub-section, we introduce the analysis for Alg. 7. We first provide several lemma and theorem for preparation, which are extended from single task setting. We will omit the detailed proofs if they are almost the same expect there are multiple source tasks and the additional $W$ in the log factors.

**Lemma F.5** (Lem. D.11 in TRL-MST Setting). *There exists an absolute constant $c^{[W]}_\Xi$, such that for arbitrary fixed $\xi > 0$, and for arbitrary*

$$k \ge c^{[W]}_\Xi \max\{\frac{\alpha B_1^2 H^2 S}{\lambda^2\xi^2}\log(\frac{\alpha HSAB_1 B_2 W}{\lambda\xi}), \frac{(C_1+\alpha C_2)SH}{\Delta_{\min}\lambda\xi}\log\frac{C_1 C_2 SAHW}{\Delta_{\min}\lambda\xi}\}$$

*on the event $\mathcal{E}_{\textbf{Con},[W],k}, \mathcal{E}_{\textbf{Bonus},[W],k}$ and $\mathcal{E}_{Alg^{Lo},[W],k}$, $\forall h\in[H], s_h\in\mathcal{S}_h$ with $N^k_{Lo,h}(s_h) > \frac{\lambda}{3}$, we have*

$$V^*_{Lo,h}(s_h)-\underline{V}^k_{Lo,h}(s_h) \le \xi.$$

**Lemma F.6** (Lem. D.10 in TRL-MST Setting). *There exists a constant $c^{[W]}_{occup}$ which is independent of $\lambda, S, A, H$ and gap $\Delta$, s.t., for all $k\ge k^{[W]}_{occup} := c^{[W]}_{occup}\frac{C_1+\alpha C_2}{\lambda\Delta_{\min}}\log(\frac{\alpha C_1 C_2 SAHW}{\lambda\Delta_{\min}})$, on the events of $\mathcal{E}_{Alg^{Lo},[W],k}$ and $\mathcal{E}_{\textbf{Con},[W],k}$, forall $w\in[W]$, $N^k_{Lo,w,h}(s_h,a_h) \ge \frac{\lambda}{3}k$ implies that $d^*_{Lo,w}(s_h,a_h) \ge \frac{\lambda}{9}$, and conversely, if $d^*_{Lo,w}(s_h,a_h) \ge \lambda$, we must have $N^k_{Lo,w,h}(s_h) \ge N^k_{Lo,w,h}(s_h,\pi^*_{Lo,w}) \ge \frac{\lambda}{3}k$.*

**Theorem F.7.** *[Thm. 4.3 in TRL-MST Setting] There exists a constant $c^{[W]}_{overest}$, such that, for arbitrary $k\ge k^{[W]}_{ost}$ with*

$$k^{[W]}_{ost} := c^{[W]}_{overest}\cdot\max\{\alpha\frac{B_1^2 H^2 S}{\lambda^2\Delta_{\min}^2}\log(HSAWB_1 B_2), \frac{(C_1+\alpha C_2)SH}{\lambda\Delta_{\min}^2}\log\frac{C_1 C_2 SAHW}{\Delta_{\min}\lambda}\},$$

*on the event of $\mathcal{E}_{\textbf{Con},[W],k}, \mathcal{E}_{\textbf{Bonus},[W],k}, \mathcal{E}_{Alg^{Lo},[W],k}$, we have:*

$$Q^*_{Hi,h}(s_h,a_h) \le \widetilde{Q}^k_{Hi,h}(s_h,a_h), \quad V^*_{Hi,h}(s_h) \le \widetilde{V}^k_{Hi,h}(s_h), \quad \forall h\in[H], s_h\in\mathcal{S}_h, a_h\in\mathcal{A}_h.$$

*and*

$$V^*_{Hi,1}(s_1)-V^{\pi^k_{Hi}}_{Hi,1}(s_1) \le 2e\mathbb{E}_{\pi^k_{Hi}}[\sum_{h=1}^{H} Clip\Big[\min\{H, 4b^k_{Hi,h}(s_h,a_h)\}\Big|\frac{\Delta_{\min}}{4eH}\vee\frac{\Delta_{Hi}(s_h,a_h)}{4e}\Big]]. \quad (33)$$

Next, we first establish a $O(\log K)$-regret bound regardless of similarity between $\mathcal{M}_{\text{Lo}}$ and $M_{\text{Hi}}$, and use it to prove Lem. F.9, which will be further used to establish the tighter bound in Thm. 5.2.

**Theorem F.8** (Regret bound for general cases).

$$Regret_K(M_{Hi}) = O\left(k_{start}^{[W]} \cdot H + SAH^2\log(2SAHK) + B_1\log(B_2K)\sum_{h=1}^{H}\sum_{s_h,a_h}(\frac{H}{\Delta_{\min}}\wedge\frac{1}{\Delta_{Hi}(s_h,a_h)})\right).$$
(34)

*Proof.* Consider $k_{start}^{[W]} := \max\{k_{ost}^{[W]}, k_{occup}^{[W]}\}$. Similar to the proof of Thm. 4.2, we can conduct identical techniques and provide the following bounds for arbitrary $s_h, a_h$:

$$\sum_{k=k_{start}^{[W]}+1}^{K}\mathbb{E}_{\pi_{Hi}^k}[\text{Clip}\left[\min\{H, 4b_{Hi,h}^k(s_h,a_h)\}\Big|\frac{\Delta_{\min}}{4eH}\vee\frac{\Delta_{Hi}(s_h,a_h)}{4e}\right]]$$

$$\leq H\cdot\sum_{k=1}^{\tau_{s_h,a_h}^K}d_{Hi}^{\pi_{Hi}^k}(s_h,a_h) + \sum_{k=\tau_{s_h,a_h}^K+1}^{K}\mathbb{E}_{\pi_{Hi}^k}[\text{Clip}\left[\min\{H, 4B_1\sqrt{\frac{\alpha\log(KB_2)}{N_{Hi,h}^k(s_h,a_h)}}\}\Big|\frac{\Delta_{\min}}{4eH}\vee\frac{\Delta_{Hi}(s_h,a_h)}{4e}\right]]$$

$$= O\left(H\log(2SAHWK) + B_1(\frac{H}{\Delta_{\min}}\wedge\frac{1}{\Delta_{Hi}(s_h,a_h)})\log(B_2WK)\right).$$

where $\tau_{s_h,a_h}^K := \min_k\ s.t.\ \forall k'\geq k,\ \frac{1}{4}\sum_{k'=1}^{k-1}d_{Hi}^{\pi_{Hi}^{k'}}(s_h,a_h)\geq\alpha\log(2SAHWK)$. Then, we can conclude:

$$\mathbb{E}[\sum_{k=1}^{K}V_{Hi,1}^*(s_1) - V_{Hi,1}^{\pi_{Hi}^k}(s_1)]$$

$$\leq \sum_{k=k_{start}^{[W]}+1}^{K}2e\mathbb{E}_{\pi_{Hi}^k}[\sum_{h=1}^{H}\text{Clip}\left[\min\{H, 4b_{Hi,h}^k(s_h,a_h)\}\Big|\frac{\Delta_{\min}}{4eH}\vee\frac{\Delta_{Hi}(s_h,a_h)}{4e}\right]\mathbb{I}[\mathcal{E}_{\textbf{Bonus},[W],k}\cap\mathcal{E}_{\text{Alg}^{\text{Lo}},[W],k}\cap\mathcal{E}_{\textbf{Con},[W],k}]]$$

$$+ \sum_{k=k_{start}^{[W]}+1}^{K}H\cdot\text{Pr}(\mathcal{E}_{\textbf{Bonus},[W],k}^{\complement}\cup\mathcal{E}_{\text{Alg}^{\text{Lo}},[W],k}^{\complement}\cup\mathcal{E}_{\textbf{Con},[W],k}^{\complement}) + H\cdot k_{start}^{[W]}$$

$$= O\left(k_{ost}^{[W]}\cdot H + SAH^2\log(2SAHWK) + B_1\sum_{h=1}^{H}\sum_{s_h,a_h}(\frac{H}{\Delta_{\min}}\wedge\frac{1}{\Delta_{Hi}(s_h,a_h)})\log(B_2WK)\right).$$

$\square$

In the following, we try to show an extension of Lem. 5.3 in RL setting. To establish result, we require some techniques to overcome the difficulty raised by state transition. We first establish sub-linear regret bound for Alg. 7 based on the multi-task version of Eq. (1), and use it we can show that $\sum_{k'=k/2}^{k}d_{Hi}^*(s_h,a_h) > 0$ when $k$ is large enough, which imply that $\pi_{\text{Lo},w^{\tilde{k}}}^*(s_h) = \pi_{Hi}^*(s_h)$ for some $\tilde{k}\in[\frac{k}{2},k]$ with high probability. Under good events, by our task selection strategy, we can expect if $s_h\in\mathcal{Z}_h^{\lambda,[W]}$ and $\pi_{\text{Lo},w^{\tilde{k}}}^*(s_h) = \pi_{Hi}^*(s_h)$, from $\tilde{k}$ to $k$, either the trusted task does not change, or it will hand over to another one recommending the same action, which is exactly $\pi_{Hi}^*(s_h)$.

**Lemma F.9.** *[Absorb to Similar Task] Under Assump. A, B, C for all $s_h\in\mathcal{Z}_h^{\lambda,[W]}$, by running Alg. 7 in Appx. F.1 with $\varepsilon = \frac{\widetilde{\Delta}_{\min}}{4(H+1)}$, $\alpha > 2$ and arbitrary $\lambda > 0$, there exists $\iota_{s_h}^* = Poly(SAH, \lambda^{-1}, \Delta_{\min}^{-1}, 1/d_{Hi}^*(s_h), \log W)$, such that, $\forall k\geq\iota_{s_h}^*$, $\text{Pr}(\pi_{Hi}^k(s_h)\neq\pi_{Hi}^*(s_h)) = O(\frac{1}{k^{\alpha-1}})$.*

*Proof.* Similar to MAB setting, we define $\mathcal{W}_{s_h}^* := \{t\in[W]|\pi_{\text{Lo},w}^*(s_h) = \pi_{Hi}^*(s_h), V_{\text{Lo},w,h}(s_h)\leq V_{Hi,w,h}(s_h) + \frac{\widetilde{\Delta}_{\min}}{4(H+1)}, d_{\text{Lo},w}^*(s_h) > \lambda\}$ and $\widetilde{\mathcal{W}}_{s_h}^* := \{t\in[W]|\pi_{\text{Lo},w}^*(s_h) = \pi_{Hi}^*(s_h), d_{\text{Lo},w}^*(s_h) >$

$\lambda\}$. The key observation is that, when $k \geq k_{start}$:

$$\Pr(\pi_{\text{Hi}}^k(s_h) = \pi_{\text{Hi}}^*(s_h)) \geq \Pr(\underbrace{\{\forall k' \in [\frac{k}{2}, k], \, \mathcal{W}_{s_h}^* \subset \mathcal{I}_{s_h}^{k'}\}}_{\mathcal{E}_{k,1}} \cap \underbrace{\{\exists k' \in [\frac{k}{2}, k], \, s.t. \, d_{\text{Hi}}^{\pi_{\text{Hi}}^{k'}}(s_h, \pi_{\text{Hi}}^k) > 0\}}_{\mathcal{E}_{k,2}}$$

$$\cap \underbrace{\{\forall k' \in [\frac{k}{2}, k], \, \forall t \in \mathcal{I}_{s_h}^{k'}, \, \pi_{\text{Lo},w}^{k'}(s_h) = \pi_{\text{Lo},w^{k'}}^*(s_h)\}}_{\mathcal{E}_{k,3}}).$$

That's because if $\mathcal{E}_{k,2}$ holds at some $k'$, then, we can only have the following two cases: (1) $w_{s_h}^{k'} = w_{s_h}^*$: because of $\mathcal{E}_{k,1}$, we must have $w_{s_h}^k = w_{s_h}^*$, which implies $\pi_{\text{Hi}}^k(s_h) = \pi_{\text{Hi}}^*(s_h)$ as a result of $\mathcal{E}_{k,3}$; (2) $w_{s_h}^{k'} \neq w_{s_h}^*$: because of $\mathcal{E}_{k,1}$ and $\mathcal{E}_{k,3}$, $w^{\widetilde{k}}$ can only transfer inside $\widetilde{\mathcal{W}}_{s_h}^*$ for $\widetilde{k} \in [k', k]$, and therefore we still have $\pi_{\text{Hi}}^k(s_h) = \pi_{\text{Hi}}^*(s_h)$. In the following, we will provide a upper bound for $\Pr(\mathcal{E}_{k,1}^{\complement}), \Pr(\mathcal{E}_{k,2}^{\complement}), \Pr(\mathcal{E}_{k,3}^{\complement})$.

On the event of $\mathcal{E}_{\textbf{Bonus},[W],k}$, with a similar analysis as Lem. D.5 on $\underline{Q}_{\text{Lo},w,h}^k$, we have:

$$\Pr(\mathcal{E}_{k,1}^{\complement}) \leq \sum_{k'=\frac{k}{2}}^{k} \Pr(w_{s_h}^* \notin \mathcal{I}_{s_h}^{k'}) \leq \sum_{k'=\frac{k}{2}}^{k} \Pr(\mathcal{E}_{\textbf{Bonus},[W],k}^{\complement}) \leq \frac{2}{(k/2)^{\alpha-1}}.$$

Besides, for arbitrary $s_h$ with $d_{\text{Hi}}^*(s_h) > 0$, and arbitrary $k$, by Azuma-Hoeffding inequality and Thm. D.6, with probability at least $1 - \frac{1}{k^\alpha}$, we have:

$$\sum_{k'=k/2}^{k} d_{\text{Hi}}^{\pi_{\text{Hi}}^k}(s_h, \pi_{\text{Hi}}^*) \geq \sum_{k'=k/2}^{k} d_{\text{Hi}}^*(s_h, \pi_{\text{Hi}}^*) - H\sqrt{2\alpha k \log k} - \mathbb{E}[\sum_{k=1}^{K} V_{\text{Hi},1}^*(s_1) - V_{\text{Hi},1}^{\pi_{\text{Hi}}^k}(s_1)]$$

$$= \frac{k}{2} d_{\text{Hi}}^*(s_h) - H\sqrt{2\alpha k \log k} - \mathbb{E}[\sum_{k=1}^{K} V_{\text{Hi},1}^*(s_1) - V_{\text{Hi},1}^{\pi_{\text{Hi}}^k}(s_1)].$$

As a result of Thm. F.8, $H\sqrt{2\alpha k \log k} + \mathbb{E}[\sum_{k=1}^{K} V_{\text{Hi},1}^*(s_1) - V_{\text{Hi},1}^{\pi_{\text{Hi}}^k}(s_1)]$ will be sub-linear w.r.t. $k$, so there exists $\iota_{s_h} := c_{s_h}\text{Poly}(S, A, H, 1/\lambda, 1/\Delta_{\min}, 1/d_{\text{Hi}}^*(s_h))$ for some absolute constant $c_{s_h}$, such that for arbitrary $k \geq \iota_{s_h}$, $\sum_{k'=k/2}^{k} d_{\text{Hi}}^{\pi_{\text{Hi}}^k}(s_h, \pi_{\text{Hi}}^*) > 0$, which implies:

$$\forall k \geq \iota_{s_h}, \; \Pr(\mathcal{E}_{k,2}^{\complement}) = \Pr(\sum_{k'=k/2}^{k} d_{\text{Hi}}^{\pi_{\text{Hi}}^k}(s_h, \pi_{\text{Hi}}^*) = 0) \leq \frac{1}{k^\alpha}.$$

Moreover, as a result of Lem. F.6, when $k \geq k_{occup}^{[W]} := c_{occup}^{[W]} \frac{C_1 + \alpha C_2}{\lambda \Delta_{\min}} \log(\frac{\alpha C_1 C_2 SAHW}{\lambda \Delta_{\min}})$, on the event of $\mathcal{E}_{\textbf{Alg}^{\text{Lo}},[W],k}$ and $\mathcal{E}_{\textbf{Con},[W],k}$, if $t \in \mathcal{I}_{s_h}^k$, we must have $\max_a N_{\text{Lo},w,h}^k(s_h, a) > \lambda k$, which implies $\pi_{\text{Lo},w}^*(s_h) = \arg\max N_{\text{Lo},w,h}^k(s_h, a)$. Therefore, $\forall k \geq 2 \cdot k_{occup}^{[W]}$:

$$\Pr(\mathcal{E}_{k,3}^{\complement}) \leq \sum_{k'=k/2}^{k} \sum_{t \in [W]} \Pr(\{t \in \mathcal{I}_{s_h}^{k'}\} \cap \{\pi_{\text{Lo},w}^*(s_h) \neq \arg\max_a N_{\text{Lo},w,h}^{k'}(s_h, a)\})$$

$$\leq \sum_{k'=k/2}^{k} \Pr(\mathcal{E}_{\textbf{Alg}^{\text{Lo}},[W],k}^{\complement}) + \Pr(\mathcal{E}_{\textbf{Con},[W],k}^{\complement}) \leq \frac{4}{(k/2)^{\alpha-1}}.$$

Therefore, for arbitrary $k \geq \iota_{s_h}^* := \max\{k_{start}^{[W]}, \iota_{s_h}, k_{occup}^{[W]}\} = \text{Poly}(S, A, H, 1/\lambda, 1/\Delta_{\min}, 1/d_{\text{Hi}}^*(s_h), \log W)$, we have:

$$\Pr(\pi_{\text{Hi}}^k(s_h) = \pi_{\text{Hi}}^*(s_h)) \geq 1 - \frac{2}{(k/2)^{\alpha-1}} - \frac{2}{(k/2)^{\alpha-1}} - \frac{4}{(k/2)^{\alpha-1}} \geq 1 - \frac{8}{(k/2)^{\alpha-1}}.$$

which finishes the proof. $\qquad \square$

**Theorem F.10** (Detailed Version of Thm. 5.2). *Under Assump. A, B and C, Cond. D.3 and F.4, by running Alg. 7 in Appx. F.1, with $\varepsilon = \frac{\widehat{\Delta}_{\min}}{4(H+1)}$, $\alpha > 2$ and arbitrary $\lambda > 0$, we have[5]:*

$$
Regret_K(M_{Hi})
$$

$$
= O\Big( H \cdot \max\{\alpha \frac{B_1^2 H^2 S}{\lambda^2 \Delta_{\min}^2} \log(HSAWB_1B_2), \frac{(C_1+\alpha C_2)SH}{\lambda \Delta_{\min}^2} \log \frac{C_1 C_2 SAHW}{\Delta_{\min}\lambda} \}
$$

$$
+ \sum_{h=1}^{H} \sum_{(s_h,a_h)\in \mathcal{C}_h^{\lambda,[W],1}} H\log(2SAHW(K \wedge \frac{1}{\Delta_{\min}\lambda d_{Hi}^*(s_h)})) + \frac{B_1}{\Delta_{Hi}(s_h,a_h)} \log(SAHB_2W(K \wedge \frac{1}{\Delta_{\min}\lambda d_{Hi}^*(s_h)}))
$$

$$
+ \sum_{h=1}^{H} \sum_{(s_h,a_h)\in \mathcal{C}_h^{\lambda,[W],2}} H\log(2SAHW(K \wedge \frac{1}{\Delta_{\min}\lambda d_{Hi,\min}^*}))
$$

$$
+ B_1(\frac{H}{\Delta_{\min}} \wedge \frac{1}{\Delta_{Hi}(s_h,a_h)})\log(SAHB_2W(K \wedge \frac{1}{\Delta_{\min}\lambda d_{Hi,\min}^*}))
$$

$$
+ \sum_{h=1}^{H} \sum_{(s_h,a_h)\in \mathcal{C}_h^*} \frac{HB_1}{\Delta_{\min}} \log(SAHB_2W \min\{K, \frac{1}{\lambda \Delta_{\min}d_{Hi}^*(s_h)}\})
$$

$$
+ \sum_{h=1}^{H} \sum_{(s_h,a_h)\in \mathcal{S}_h\times\mathcal{A}_h\backslash(\mathcal{C}_h^{\lambda,[W],1}\cup \mathcal{C}_h^{\lambda,[W],2})} H\log(2SAHWK) + B_1(\frac{H}{\Delta_{\min}} \wedge \frac{1}{\Delta_{Hi}(s_h,a_h)})\log(B_2WK) \Big)
$$

$$
\tag{35}
$$

$$
= O(SH \sum_{h=1}^{H} \sum_{(s_h,a_h)\in \mathcal{S}_h\times\mathcal{A}_h\backslash\mathcal{C}_h^{\lambda,[W]}} \frac{H}{\Delta_{\min}} \wedge \frac{1}{\Delta_{Hi}(s_h,a_h)} \log(SAHWK)).
$$

*Proof.* Consider the same $k_{start}^{[W]} = \max\{k_{ost}^{[W]}, k_{occup}^{[W]}\}$. Similar to single task setting, we study the following two types of states.

**Type 1:** $(s_h, a_h) \in \mathcal{C}_h^{\lambda,[W],1} \cup \mathcal{C}_h^{\lambda,[W],2}$: **Constant Regret because of Low Visitation Probability**
For each $(s_h, a_h) \in \mathcal{C}_h^{\lambda,[W],1}$, by leveraging Lem. F.9, we have:

$$
\sum_{k=k_{start}^{[W]}+1}^{K} \mathbb{E}_{\pi_{Hi}^k}[\text{Clip}\left[\min\{H, 4b_{Hi,h}^k(s_h,a_h)\}\Big|\frac{\Delta_{\min}}{4eH} \vee \frac{\Delta_{Hi}(s_h,a_h)}{4e}\right] \mathbb{I}[\mathcal{E}_{\textbf{Bonus},[W],k} \cap \mathcal{E}_{\textbf{Alg}^{Lo},[W],k} \cap \mathcal{E}_{\textbf{Con},[W],k}]]
$$

$$
= \sum_{k=k_{start}^{[W]}+1}^{\iota_{s_h}^*} \mathbb{E}_{\pi_{Hi}^k}[\text{Clip}\left[\min\{H, 4b_{Hi,h}^k(s_h,a_h)\}\Big|\frac{\Delta_{\min}}{4eH} \vee \frac{\Delta_{Hi}(s_h,a_h)}{4e}\right] \mathbb{I}[\mathcal{E}_{\textbf{Bonus},[W],k} \cap \mathcal{E}_{\textbf{Alg}^{Lo},[W],k} \cap \mathcal{E}_{\textbf{Con},[W],k}]]
$$

$$
= O\Big(H\log(2SAHW(K \wedge \frac{1}{\Delta_{\min}\lambda d_{Hi}^*(s_h)})) + \frac{B_1}{\Delta_{Hi}(s_h,a_h)} \log(SAHB_2W(K \wedge \frac{1}{\Delta_{\min}\lambda d_{Hi}^*(s_h)})) \Big).
$$

Note that different from single task setting, the convergence speed of $\pi_{Hi}^k(s_h)$ to $\pi_{Hi}^*(s_h)$ for $s_h \in \mathcal{Z}^{\lambda,[W]}$ depends on $d_{Hi}^*(s_h)$. Therefore, for $(s_h, a_h) \in \mathcal{C}_h^{\lambda,[W],2}$, we can guarantee $d^{\pi_{Hi}^k(s_h,a_h)}$ decays to zero only after the convergence of all its ancester states in $\mathcal{Z}^{\lambda,[W]}$, and:

$$
\sum_{k=k_{start}^{[W]}+1}^{K} \mathbb{E}_{\pi_{Hi}^k}[\text{Clip}\left[\min\{H, 4b_{Hi,h}^k(s_h,a_h)\}\Big|\frac{\Delta_{\min}}{4eH} \vee \frac{\Delta_{Hi}(s_h,a_h)}{4e}\right] \mathbb{I}[\mathcal{E}_{\textbf{Bonus},[W],k} \cap \mathcal{E}_{\textbf{Alg}^{Lo},[W],k} \cap \mathcal{E}_{\textbf{Con},[W],k}]]
$$

$$
\leq \sum_{k=k_{start}^{[W]}+1}^{k_{s_h,a_h}} \mathbb{E}_{\pi_{Hi}^k}[\text{Clip}\left[\min\{H, 4b_{Hi,h}^k(s_h,a_h)\}\Big|\frac{\Delta_{\min}}{4eH} \vee \frac{\Delta_{Hi}(s_h,a_h)}{4e}\right] \mathbb{I}[\mathcal{E}_{\textbf{Bonus},[W],k} \cap \mathcal{E}_{\textbf{Alg}^{Lo},[W],k} \cap \mathcal{E}_{\textbf{Con},[W],k}]]
$$

---

[5]We only keep non-constant terms and defer the detailed result to Eq. (35).

$$=O\Big(H\log(2SAHW(K\wedge\frac{1}{\Delta_{\min}\lambda d^*_{\mathrm{Hi,min}}}))+B_1(\frac{H}{\Delta_{\min}}\wedge\frac{1}{\Delta_{\mathrm{Hi}}(s_h,a_h)})\log(SAHB_2W(K\wedge\frac{1}{\Delta_{\min}\lambda d^*_{\mathrm{Hi,min}}}))\Big).$$

where $k_{s_h,a_h}:=\max_{h'\in[h-1],s_{h'}\in\mathcal{C}_{h'}^{\lambda,[W],1}}\iota^*_{s_{h'}}$ and $d^*_{\mathrm{Hi,min}}:=\min_{s_h:d^*_{\mathrm{Hi}}(s_h)>0}d^*_{\mathrm{Hi}}(s_h)$ is the minimal probability to reach state by optimal policy in Hi.

**Type 2:** $(s_h,a_h)\in\mathcal{C}_h^*$, **i.e.** $d^*_{\mathbf{Hi}}(s_h,a_h)=d^*_{\mathbf{Hi}}(s_h)>0$   Similar to the discussion of Type 2 in the proof of Thm. 4.2, when $k\geq\tau^*_{s_h}$ for some $\tau^*_{s_h}=\mathrm{Poly}(S,A,H,\lambda^{-1},\Delta_{\min}^{-1},(d^*_{\mathrm{Hi}}(s_h))^{-1},\log W)$, we have $\sum_{k'=1}^{k-1}d^{\pi^k_{\mathrm{Hi}}}_{\mathrm{Hi}}(s_h,a_h)=O(kd^*_{\mathrm{Hi}}(s_h))$, and therefore,

$$\sum_{k=k_{start}^{[W]}+1}^{K}\mathbb{E}_{\pi_{\mathrm{Hi}}^k}[\mathrm{Clip}\Big[\min\{H,4b^k_{\mathrm{Hi},h}(s_h,a_h)\}\Big|\frac{\Delta_{\min}}{4eH}\vee\frac{\Delta_{\mathrm{Hi}}(s_h,a_h)}{4e}\Big]\,\Big|\,\mathcal{E}_{\mathbf{Bonus},[W],k},\mathcal{E}_{\mathrm{Alg^{Lo}},[W],k},\mathcal{E}_{\mathbf{Con},[W],k}]$$

$$=O\left(\frac{HB_1}{\Delta_{\min}}\log(SAHB_2W\min\{K,\frac{1}{\lambda\Delta_{\min}d^*_{\mathrm{Hi}}(s_h)}\})\right).$$

Therefore, we can conclude that:

$\mathrm{Regret}_K(M_{\mathrm{Hi}})$

$$=O\Big(H\cdot\max\{\alpha\frac{B_1^2H^2S}{\lambda^2\Delta_{\min}^2}\log(HSAWB_1B_2),\frac{(C_1+\alpha C_2)SH}{\lambda\Delta_{\min}^2}\log\frac{C_1C_2SAHW}{\Delta_{\min}\lambda}\}$$

$$+\sum_{h=1}^{H}\sum_{(s_h,a_h)\in\mathcal{C}_h^{\lambda,[W],1}}H\log(2SAHW(K\wedge\frac{1}{\Delta_{\min}\lambda d^*_{\mathrm{Hi}}(s_h)}))+\frac{B_1}{\Delta_{\mathrm{Hi}}(s_h,a_h)}\log(SAHB_2W(K\wedge\frac{1}{\Delta_{\min}\lambda d^*_{\mathrm{Hi}}(s_h)}))$$

$$+\sum_{h=1}^{H}\sum_{(s_h,a_h)\in\mathcal{C}_h^{\lambda,[W],2}}H\log(2SAHW(K\wedge\frac{1}{\Delta_{\min}\lambda d^*_{\mathrm{Hi,min}}}))$$

$$+B_1(\frac{H}{\Delta_{\min}}\wedge\frac{1}{\Delta_{\mathrm{Hi}}(s_h,a_h)})\log(SAHB_2W(K\wedge\frac{1}{\Delta_{\min}\lambda d^*_{\mathrm{Hi,min}}}))$$

$$+\sum_{h=1}^{H}\sum_{(s_h,a_h)\in\mathcal{C}_h^*}\frac{HB_1}{\Delta_{\min}}\log(SAHB_2W\min\{K,\frac{1}{\lambda\Delta_{\min}d^*_{\mathrm{Hi}}(s_h)}\})$$

$$+\sum_{h=1}^{H}\sum_{(s_h,a_h)\in\mathcal{S}_h\times\mathcal{A}_h\backslash(\mathcal{C}_h^{\lambda,[W],1}\cup\mathcal{C}_h^{\lambda,[W],2})}H\log(2SAHWK)+B_1(\frac{H}{\Delta_{\min}}\wedge\frac{1}{\Delta_{\mathrm{Hi}}(s_h,a_h)})\log(B_2WK)\Big).$$

We consider the similar Hoeffding bound for the choice of $B_1$ and $B_2$ as Example D.4, and by plugging $B_1=\Theta(SH)$ and $B_2=\Theta(SA)$ into the above equation and omitting all the terms independent with $K$, we have:

$$\mathrm{Regret}_K(M_{\mathrm{Hi}})=O(SH\sum_{h=1}^{H}\sum_{(s_h,a_h)\in\mathcal{S}_h\times\mathcal{A}_h\backslash\mathcal{C}_h^{\lambda,[W]}}\frac{H}{\Delta_{\min}}\wedge\frac{1}{\Delta_{\mathrm{Hi}}(s_h,a_h)}\log(SAHWK)).$$

$\square$

# G   Missing Details for Experiments

**Construction of Source and Target Tasks**   We first randomly construct the transition function of the high-tier task $M_{\mathrm{Hi}}$ (i.e. $\mathbb{P}_{\mathrm{Hi}}$ are randomly sampled and normalized to make sure their validity). Then, similarly, we randomly construct the reward function of $M_{\mathrm{Hi}}$ and shift the reward function to ensure $M_{\mathrm{Hi}}$ has unique optimal policy and $\Delta_{\min,\mathrm{Hi}}=0.1$.

Next, we construct the source tasks by randomly permute the transition matrix. In another word, for any $s_h$, we randomly permute $a_1,a_2,a_3$ to $a'_1,a'_2,a'_3$ and assign $\mathbb{P}_{h,\mathrm{Lo}}(\cdot|s_h,a'_i)\leftarrow\mathbb{P}_{h,\mathrm{Hi}}(\cdot|s_h,a_i)$ for $i\in[3]$. In this way, the Optimal Value Dominance (OVD) condition is ensured, and we can expect some of $s_h$ are transferable when $\pi^*_{\mathrm{Lo}}(s_h)=\pi^*_{\mathrm{Hi}}(s_h)$. When the number of source tasks $W>1$, we repeat the above process and construct $W$ different source tasks.