# OpenReview forum: "Robust Knowledge Transfer in Tiered Reinforcement Learning"
_NeurIPS.cc/2023/Conference — NeurIPS 2023 poster_

### Official Review · Reviewer_MBdo · 2023-06-10

**Soundness:** 3 good
**Presentation:** 2 fair
**Contribution:** 2 fair
**Rating:** 5
**Confidence:** 1

**Summary:**

The paper presents an extension of "Tiered-RL", a multi-fidelity RL framework where a "low-fidelity" environment is executed in parallel with the "high-fidelity" environment, with the purpose of training faster while keeping near-optimal regret. The paper is a theoretical exploration without empirical evaluation

**Strengths:**

- The paper presents a deep theoretical evaluation of a setting that is very relevant yet not very methodologically-explored: When we have related tasks of varied importances to be solved (such as sim2real), and one task can be leveraged to learn faster another one.
- An algorithm able to guarantee near-optimal regret while training in multiple tasks might be of use in security-critical applications such as robotics or medical domains.

**Weaknesses:**

- While the theoretical results sound exciting, I would expect at least a simple empirical evaluation of the proposed algorithm to be provided to show how hard it is to actually implement the algorithm in a practical domain.

- The "Tiered RL" setting sounds very similar (if not exactly the same) as the multi-fidelity RL modeling, that wasn't even cited by the authors. While there is still a novelty in the exact way the problem is solved by running all "tiers" (or fidelities) in parallel, the problem formulation seems exactly like multi-fidelity MDPs to me, and I would suggest to use the same multi-fidelity MDP formulation to keep consistency in the literature. Multi-fidelity MDPs are explicitly modeled in this paper:

Silva, Felipe Leno, et al. "Toward Multi-Fidelity Reinforcement Learning for Symbolic Optimization.", Adaptive and Learning Agents (ALA) workshop, 2023.

And the multi-fidelity RL problem has been explored in a similar way in the following papers

Sami Khairy and Prasanna Balaprakash. 2022. Multifidelity reinforcement learning with control variates. arXiv preprint arXiv:2206.05165 (2022).

Mark Cutler, Thomas J Walsh, and Jonathan P How. 2014. Reinforcement learning with multi-fidelity simulators. In 2014 IEEE International Conference on Robotics and Automation (ICRA). IEEE, 3888–3895.

At the very least all of those papers should have been included to your related works section.

**Questions:**

- A doubt regarding the setting. Would the agent care about the performance in all tiers? Or is it only important to optimize the performance in the highest tier? If only the highest tier matter, the setting is exactly the same as in multi-fidelity RL as said in the "weaknesses" section, otherwise, I would need more clarification of in which practical application this modeling would be useful.

**Limitations:**

No foreseeable negative societal impact

---

> ### Author Rebuttal · Authors · 2023-08-09
>
> We thank the reviewer for the valuable feedback. Below we address the concerns raised by the reviewer.
>
> ### Experiment
> Thanks for the suggestion. We conduct some empirical evaluation of our algorithm on  toy examples, which well validates our theory. Please refer to our global response for more details.
>
> ### Difference with Multi-fidelity RL and Questions about the setting
> Thanks for pointing out the references on multi-fidelity RL. We will properly cite them and add discussions about it.  We believe our Tiered RL setting has fundamental differences from multi-fidelity RL setting.
> Multi-fidelity RL considers the case when both low-fidelity, cheap data and high-fidelity, expensive data are available, and aims at solving the task with least queries to high-fidelity simulators by leveraging low-fidelity data.
> In contrast, as we motivated in our introduction, the Tiered RL setting sits in between transfer RL and Multi-Task RL, where source and multiple tasks are solved in parallel. With close inspection, we can see
>
> (1) In multi-fidelity RL, although there are simulators at different levels, there is only one task to be solved, while in Tiered RL, we have to solve source and target tasks.
> More importantly, as we clearly mentioned in the ontroduction section (line 45-48), although we want to benefit target tasks by knowledge transfer, we still expect source tasks do not sacrifice for that and still retain near-optimal regret. This objective inherits from [1], and it is reasonable for user-interacting applications with “tiered customer” structure. Please refer to [1] for more concrete examples.
>
> (2) In multi-fidelity RL, it is assumed that the low-fidelity data is cheap and abundant. However, in Tiered RL, like multi-task RL, the tasks are solved in parallel, so the samples from low-tier tasks are still limited (in each iteration, either source or target tasks can only collect one trajectory).  The data scarce in source tasks also makes our setting more challenging.
>
>
> [1] Huang et. al., Tiered reinforc ment learning: Pessimism in the face of uncertainty and constant regret.

---

> > ### Comment · Reviewer_MBdo · 2023-08-18
> >
> > Thanks for the response, I will keep my already-positive score.

---

### Official Review · Reviewer_BLAr · 2023-07-06

**Soundness:** 2 fair
**Presentation:** 2 fair
**Contribution:** 3 good
**Rating:** 5
**Confidence:** 3

**Summary:**

The authors look at the tiered reinforcement learning setting, which is a parallel transfer learning framework where the goal is to transfer knowledge from a low-tier source task to a high-tier target task in order to reduce the exploration risk of the high-tier task. Additionally, these tasks are solved in parallel. Contrary to previous related work, the authors do not assume the low-tier and high-tier tasks share dynamics or reward functions and focus on robust knowledge transfer without prior knowledge on task similarity. The authors use a condition called the “Optimal Value Dominance” to propose novel online learning algorithms that can achieve constant regret on partial states depending on the task similarity and near-optimal regret when the two tasks are dissimilar. Furthermore, for low-tier tasks, these algorithms keep near-optimal regret at very little cost. The authors also study the scenario when multiple low-tier tasks are present and propose a novel transfer source selection algorithm that can gather knowledge from all low-tier tasks and produce benefits on a much larger state-action space.

**Strengths:**

* The regret analysis of the robust tiered multi-armed bandit (MAB) models was very thorough.

**Weaknesses:**

* No experiments that compared performance with other tiered RL algorithms
* It would have nice to have had the related work in the manuscript instead of as a supplement.

**Questions:**

* What type of data have others used on this problem or similar RL problems? Can it be leveraged in this setting as well?

**Limitations:**

* The authors did not address the fact that they did not perform any experiments with either synthetic or real data on their model.

---

> ### Author Rebuttal · Authors · 2023-08-09
>
> We thank the reviewer for the valuable feedback. Below we address the concerns raised by the reviewer.
>
> ### Experiments that compared performance with other tiered RL algorithms
> We highlight that the general setting of tiered RL without prior knowledge of task similarity and/or with multiple source tasks has not been studied in the literature (which is precisely the gap our work aims to fill in); thus there exists no benchmarks or state-of-the-art baselines in this case for us to run comparison with.
>
> Following the reviewer’s comment, we have now conducted some empirical evaluation of our algorithm on toy examples, which well verifies our theory. Please refer to our global response and PDF attached there for more details.
>
> ### Related work
> Thanks for the suggestion. We are happy to move part of the related work (currently in Appx A.2 and A.3) from the supplementary material to the main paper in the revision for ease of reading and completeness as long as the space allows.  We apologize for the inconvenience and invite the reviewer to check these discussions in the Appendix for now.
> ### “What type of data …”
> To our knowledge, there is no prior work studying the theory of parallel transfer RL with single or multiple source tasks.
> For the normal transfer/multi-task RL setting, it is studied to transfer a fixed well-learned value functions [2], or data collected from other tasks [3].
> In our setting, the source tasks are not solved in advance, and the target task should leverage the information from them once available, so we transfer the trajectory data in source tasks directly, which actually contains all the raw information and can be used to further construct value or transition estimations.
>
> [1] Huang et. al.,  Tiered reinforcement learning: Pessimism in the face of uncertainty and constant regret.
>
> [2] Noah Golowich and Ankur Moitra. Can q-learning be improved with advice?
>
> [3] Chicheng Zhang and Zhi Wang. Provably efficient multi-task reinforcement learning with model transfer.

---

> > ### Comment · Reviewer_BLAr · 2023-08-13
> >
> > Yes, please strongly consider moving your related works into the paper. I believe this will help provide more clarity to the paper and show the gaps you are filling with your work.
> >
> > I am also glad to hear that you have used simulated data to at least show some empirical evaluation of your algorithm.
> >
> > As a result of these revisions, I have decided to increase my rating.

---

### Official Review · Reviewer_9F2D · 2023-07-07

**Soundness:** 2 fair
**Presentation:** 3 good
**Contribution:** 3 good
**Rating:** 5
**Confidence:** 4

**Summary:**

The authors propose a robust parallel knowledge transfer reinforcement learning algorithm for single or multiple source tasks without knowledge on model similarity using the previously defined Tiered Reinforcement Learning framework. The paper remove the limitation on prior knowledge about the task similarity to generalize the framework. The main contribution is three-fold: 1) Establish necessary conditions for lower regret bound, 2) propose robust parallel transfer algorithms for reinforcement learning and its special case, multi-armed bandits, and 3) describe a new source task selection mechanism that guarantees constant regret for larger space of state-action pairs using multiple low-tier tasks.


**Strengths:**

Originality. Removing the assumption on task similarity poses new challenges to guarantee lower regret bounds that are not trivial. The new algorithm to solve the general knowledge transfer require learning whether low-tier and high-tier tasks are similar from observed data and balance between exploration and exploration from the low-tier model at the same time. The lower bounds from simplifying the general case simplify to the prior work under the similarity conditions between low- and high-tier tasks. Therefore, the lower bounds are sound. To extend the approach for the high-tier task to leverage multiple low-tier tasks, the authors describe a method called “trust till failure” that still guarantees lower bounds on regret.

Quality & Clarity. The contribution and novelty is well defined. The definitions and equations are sound.

Significance. The proposed approach generalizes the previous work to a larger class of problems while still maintaining lower bounds on regret minimization.

**Weaknesses:**

Originality & Significance. To remove the limitation of task similarity for generalization is not trivial. However, the motivation for generalization is lacking with few references to potential robotics applications. The paper can benefit from providing an illustrative example or toy problems to motivate new class of problems the proposed method can be applied to.

Quality. I am assuming the authors made a mistake and uploaded the incomplete paper. The related work is supposed to be in the Appendix but there are no Appendices attached to the submitted paper. Without these Appendices, it is difficult to compare how the new method compared to prior work. There are several grammatical errors and the conclusion section does not talk about the limitations of the current approach. The future work is also in Appendix which is not included in the paper.

Clarity. It is not clear how parallel knowledge transfer is any different from meta-learning and/or why it is not a special case of meta-learning. It will be good to discuss when the learning methods overlap or differ under which conditions to better motivate the approach. Specifically, for the use case of multiple low-tier learners with a single high-tier learner.


**Questions:**

Please provide the appendices to complete the evaluation of the work on soundness and its relation to prior work.


**Limitations:**

The authors do not provide limitations of the proposed approach. There are no societal issues related to this work.

---

> ### Author Rebuttal · Authors · 2023-08-09
>
> We thank the reviewer for valuable feedback. Below we address the concerns raised by the reviewer.
>
> ### About the appendix
> We  are afraid the reviewer might have overlooked the **Supplementary Material** part of the submission, where we include the full paper with appendix in a zip file.
>
> If space allowed, we are happy to move part of the related work (currently in Appx A.2 and A.3) and future work (currently in Appex A.5) from the supplementary material to the main paper in the revision for ease of reading and completeness.  We kindly invite the reviewer to check these discussions in the Appendix.
>
> ### Lack of motivation and illustration on toy problems
> Thanks for the suggestion. We will strengthen our motivation by highlighting other potential applications and theoretical gaps in the existing work. Following the suggestion, we have now included some numerical illustration on a toy example which showcases the performance of our proposed algorithm and validates our theory. Please refer to our global response and the attached PDF for more details.
>
> ### Difference with Meta-Learning
>
> In general, meta learning is about learning to learn from **metadata**. Here learning to learn means it learns an algorithm instead of outputting a policy, and metadata refers to the data about data, for example, properties about the algorithm used, learning task itself, etc.
>
> In contrast, in our Tiered RL setting, like transfer RL, we distinguish the importance of tasks and directly use the **data from other source tasks** to accelerate the policy learning in the "high-tier" tasks.
> In another word, meta learning considers to use “high-order” data to learn an algorithm, while we target at using normal data from source tasks to benefit “high-tier” tasks.
>
>
> ### Limitations of the proposed approach
>
> We have discussed some limitations of our results  and open problems in Appedix A.5, for instance, the dependency gap in the lower bound for RL case (see also lines 164-166),  the OVD assumptions. Following the suggestion of the reviewer, we will assemble these limitations in an explicit section in the main paper.

---

> > ### Author Response · Authors · 2023-08-19
> >
> > We thank the reviewer again for the valuable feedback. Given that the discussion period is ending, it would be appreciated if you could review our response and notify us if your concerns are addressed or you have further questions.

---

> > > ### Comment · Reviewer_9F2D · 2023-08-21
> > >
> > > Yes, my concerns were addressed. Thank you.

---

### Official Review · Reviewer_5dMC · 2023-07-09

**Soundness:** 4 excellent
**Presentation:** 3 good
**Contribution:** 3 good
**Rating:** 7
**Confidence:** 3

**Summary:**

This paper studies the tiered reinforcement learning setting, a transfer learning setting where the goal is to transfer information from a low tier task to a higher tier one while learning both to solve both tasks in parallel.
Contrary to prior work the author do not assume that both tasks share the same rewards and dynamics, they show it is still possible to benefit from the low tier task to learn the higher tier one. The author also extend their work by considering multiple low tier tasks and present selection mechanism which can gather information the different tasks.


**Strengths:**

Overall this is a solid paper, while technical the paper is well written and properly organized.
Removing the assumption that the low and high tier sources share the same dynamics and rewards makes this setting much more applicable and interesting.
The setup with multiple low tier tasks is also valuable could have many applications.

**Weaknesses:**

It would have been interesting to add empirical evaluations of the proposed algorithms to understand their actual performance.


**Questions:**

In the multiple source tier setting is there a point where adding more tasks with a low similarity with the high tier tasks can hurt performance?

**Limitations:**

The authors adequately addressed the limitations.

---

> ### Author Rebuttal · Authors · 2023-08-09
>
> We thank the reviewer for the very positive and valuable feedback. Below we address the questions raised by the reviewer.
>
> ### For the experiments
> We conduct some empirical evaluation on toy examples, which well verifies our theory. Please refer to our global response for more details.
>
> ### More tasks could be harmful
> Thanks for the interesting question. One direct observation is that, for the fixed failure rate, we need to adjust the bonus term by a $\log W$, where $W$ is the number of source tasks. Therefore, it could be harmful if one continuously adds source tasks which cannot enlarge the set of transferable states. It might be interesting to study how to select tasks which have more potential to introduce new transferable states, but it is out of the scope of this paper.

---

### Official Review · Reviewer_G1Lf · 2023-07-27

**Soundness:** 3 good
**Presentation:** 3 good
**Contribution:** 3 good
**Rating:** 6
**Confidence:** 3

**Summary:**

The paper extends the work of Huang et al (2022) on Tiered Reinforcement Learning (where the objective is to transfer knowledge from the low tier (risk-tolerant) source task to a high-tier (risk-averse) target task while solving the two two tasks in parallel) by relaxing the assumptions of identical reward and transition functions in the source and target tasks.
The paper first identifies ‘Optimal Value Dominance’ a necessary condition to keep the source algorithm near-optimal while achieving provable benefits for the regret in the high-tier algorithm when Low and High tier tasks are similar.
The paper then proposes algorithms for tiered multiarmed bandit and tiered reinforcement learning with regret lower bounds depending on the similarity between the low- and high-tier tasks, with an improved lower bound for the case when the tasks are the same compared to Huang et al (2022).
Finally, the paper also considers a case when there are multiple low-tier tasks, in which the most similar states to the high-tier task are chosen from any of the low tier tasks for an additional log W factor in regret.

**Strengths:**

Originality

The paper extends the framework (multiple low-tier tasks) of and relaxes some of the assumptions (non-identical reward and transition function among the low- and high-tier tasks) of a previous paper on Tiered RL.

Quality

The paper is mostly well-written, clear and concise. The paper motivates the problem setting well in the introduction, and places it well in the literature. The concepts are introduced and explained in a logical order.

Clarity

The paper has a similar structure to Huang et al (2022). It is mostly clear and easy to follow, although certain parts do feel a bit rushed. (I will mention them in the Weaknesses part.)

Significance

The paper contributes a few findings in Tiered RL that could be useful to the community: it proves a tighter lower bound for the case when the lower and higher tier tasks are the same, and proves lower bounds for the case when the reward and transition functions differ depending on the difference in the gaps. It also introduces to setting when there are multiple low-tier tasks.

**Weaknesses:**

The work in this paper results from a minor relaxation of certain assumption in Huang et al (2022), and a lot of possible extensions and improvements are left for future work.

The RL results depend on a hyperparameter, lambda that needs to be chosen wisely.

Minor typos/grammar/style issues that did not affect the rating of this paper:

Lines 37, 41 and 52: “In [13]” is considered bad style, either write Author et al (year) [13], or better yet just cite the claim.

Lines 62: conceptions -> findings

Lines 67-68, starting with while: that subsentence makes no sense, please rephrase

The sentence on lines 125-127 seems like it belongs to the Frequently Used Notations subsection, not to 2.1

Line 165: no need for ‘actually’

Line 166: ‘leave it to the future work’ -> ‘leave it for future work’

Line 176: remove ‘kind of’ or replace with ‘may [explain]’

Line 177 is -> are, literatures -> papers/works/publications


**Questions:**

What do you mean by ‘back-propagation process of value iteration’ (line 255)? I am familiar with the concepts of backpropagation and value iteration individually, but their combination is rather ambiguous to me.

How would one go about choosing the lambda hyperparameter before beginning to solve the tasks?

**Limitations:**

The authors are upfront about the limitations, such as the unique optimal policy assumption, and mention in the appendix that the avoidance of the lower bound knowledge about the minimal gap would be beneficial. Concerns for negative societal impact in this theoretical paper are not applicable.

---

> ### Author Rebuttal · Authors · 2023-08-09
>
> We thank the reviewer for valuable feedback. We will fix the typos as mentioned. Below we address the main questions raised by the reviewer.
>
> ### “this paper results from a minor relaxation of…”
>
> We humbly disagree to credit our contributions as “minor relaxation” of the assumptions.
>
> First of all, removing the assumption that the low-tier and high-tier tasks share the transition and reward functions is highly non-trivial and of great practical value, which is also noted by Reviewer 5dMC and Reviewer 9F2D . The sharing model assumption in Huang et al (2022) requires very strong prior knowledge (and often unrealistic), which allows simple algorithm design based on PVI. Moreover, without such prior knowledge, the problem becomes much more challenging as we need a carefully designed strategy to identify and avoid negative transfer. We introduce several novel components, including (1) a branching condition to decide transfer or not (noted in our Algs. 1,2), (2) value adjustment to re-ensure overestimation, etc, which are substantially different from PVI in [1].
>
> Secondly, we generalize the results of tiered RL from single source task to multiple different source tasks, which is another substantial extension. This requires the introduction of a novel source task selection mechanism to ensemble information from low-tier tasks , which has not been studied in existing literature and can be of independent interest for transfer learning.
>
> ### The choice of hyperparamter $\lambda$
>
> We want to highlight that we do not treat $\lambda$ as a parameter to be optimized.
> Instead, we are interested in studying the effect of this parameter on the regret.
> Moreover, our main results will not be undermined if the choice of $\lambda$ is not ideal (as long as keep it away from 0), since as we proved, the regret is always at least near optimal and \lambda will only affect the size of transferrable states.
>
> In practice, without prior knowledge about $\max_s d^*_{\text{Lo}}(s)$, one can choose \lambda to be around $\Theta(1/S)$, since for each layer $h$, there exists at least one state such that $d^*_{\text{Lo}}(s_h)\geq 1/S$, or one can choose it to be constant level to avoid large “burn-in” terms.
>
> ### “back-propagation process of value iteration”
> It refers to the backward update $V_h(\cdot) \gets \max_{a_h} r_h(\cdot,a_h) + \mathbb{P}_h V\_{h+1}(\cdot) + (bonus)$, which computes value functions from back to the front. We will modify the wording.
>
> [1] Huang et. al.,  Tiered reinforcement learning: Pessimism in the face of uncertainty and constant regret.

---

> > ### Comment · Reviewer_G1Lf · 2023-08-11
> >
> > Thank you for your response and answering my questions.
> >
> > ``We humbly disagree to credit our contributions as “minor relaxation” of the assumptions.``
> >
> > I would like to emphasize that I did not mean the expression “minor relaxation” as belittling at all: however it can hardly be disputed that the paper indeed relaxes certain assumptions of the work of Huang et al (2022), and extends it in others, hence it is not  an original paper per se in the sense of proposing a novel framework to be investigated for example. However, there is nothing wrong with this, most research is incremental and that’s how it should be too.
> >
> > `We want to highlight that we do not treat $\lambda$ as a parameter to be optimized.`
> >
> > It was clear from the paper that it is not a parameter to be optimized, but the correct way of choosing it was unclear to me. An additional discussion of the role of \lambda akin to your response above (even in the appendix) would make the paper a bit more complete in my opinion.

---

> > > ### Author Response · Authors · 2023-08-11
> > >
> > > Thanks for the quick response and suggestion. We will add more discussion for $\lambda$ in the paper as suggested.

---

### Author Rebuttal · Authors · 2023-08-09

We thank reviewers for their hard work and valuable feedbacks.

## General Remarks on Experiments
Several reviewers (Reviewer 5dMC, Reviewer 9F2D, Reviewer BLAr, Reviewer MBdo) pointed out the lack of empirical evaluations as a main weakness. In our humble opinion, as a theory paper, our results provide a solid understanding on the provable benefits and fundamental limits with robust knowledge transfer in tiered RL. We believe the results are already interesting on their own, both from theoretical and technical perspectives. Note that the majority of the references (e.g., [9, 36]) that we cited in the paper (many also published at ICML/NeurIPS) do not contain any numerical results.

Nevertheless, we agree with the reviewer that some empirical validation can still be valuable.  Note that the general setting of tiered RL without prior knowledge of task similarity and/or with multiple source tasks has not been studied in the literature (which is precisely the gap our work aims to fill in); thus there exists no benchmarks or state-of-the-art baselines in this case.  A reasonable experiment would be to validate the performance of our own algorithms. To this end, we select the most representative Tiered RL algorithm, Alg. 7 in multiple source tasks setting, and evaluate it on a toy tabular MDP task. In the PDF attached in this response, we report the numerical results.

As we can see from Figure 1,
* After the transfer is activated, the regret in the target task will suddenly increase for a while, because the target task has to make some mistakes and learn from it as a result of the model uncertainty. However, because of our algorithm design, the negative transfer will terminate after a very short period.
* If we add more source tasks which can introduce new transferable states, the target task will suffer less regret.

We believe this experiment well verifies our theory prediction and we are happy to add these and more experimental results in the revision if the reviewers find them necessary.

Below we provide details on the experimental setups.

### Construction of Source and Target Tasks
We set $S=A=3$ and $H=5$. We first randomly construct the transition function of the high-tier task $M\_{\text{Hi}}$ (i.e. $\mathbb{P}\_{\text{Hi}}$ are randomly sampled and normalized to make sure their validity).
Then, similarly, we randomly construct the reward function of $M_{\text{Hi}}$ and shift the reward function to ensure $M_{\text{Hi}}$ has unique optimal policy and $\Delta_{\min, \text{Hi}} = 0.1$.

Next, we construct the source tasks by randomly permute the transition matrix of $M_{\text{Hi}}$. In another word, for any $s_h$, we randomly permute $a_1,a_2,a_3$ to $a_1',a_2',a_3'$ and assign $\mathbb{P}\_{h,\text{Lo}}(\cdot|s_h,a_i') \gets \mathbb{P}\_{h,\text{Hi}}(\cdot|s_h,a_i)$ for $i\in[3]$. In this way, the Optimal Value Dominance (OVD) condition is ensured, and we can expect some of $s_h$ are transferable when $\pi^*_{\text{Lo}}(s_h) = \pi^*_{\text{Hi}}(s_h)$.
When the number of source tasks $W > 1$, we repeat the above process and construct $W$ different source tasks.

### Experiments Setting
We adapt StrongEuler in [1] as online learning algorithm to solve source tasks, and use the bonus function in [1] as the bonus function in our Alg. 7.

We evaluate our algorithm when $W = 0, 1, 2, 5$, where $W = 0$ means the high-tier task $M_{\text{Hi}}$ is simply solved by normal online learning method (StrongEuler) without any parallel knowledge transfer.
We choose $\lambda = 0.3 \approx 1/S$ in Alg. 7, and in the MDP instance we test, **among all $S\cdot H=15$ states, for $W=1,2,5$, the number of transferable states would be 6, 9 and 13, respectively**.

We evaluate for $K = 1e7$ iterations, where we start the transfer from $k=5e5$, in order to avoid large "burn-in" terms because of the large uncertainty in source tasks in early stage. Each curve is averaged over 20 runs, and the shadows indicate 96% confidence intervals.


[1] Max Simchowitz and Kevin G Jamieson. Non-asymptotic gap-dependent regret bounds for tabular mdps. (NeurIPS 2019)

---

### Decision · Program_Chairs · 2023-09-21

**Decision:**

Accept (poster)

**Comment:**

The paper studies a specific transfer learning paradigm where knowledge is transferred from a low-tier (source) task to a high-tier (target) task. While all reviewers agree that more solid empirical evaluation would make the paper much stronger, there is consensus that the setting, the algorithmic and theoretical contributions are sufficient to accept the paper.

I would suggest the authors to integrate reviews and rebuttal into the final version of the paper. In particular, provide strong motivations for the setting and clarify the technical challenges and the value of the theoretical results.